# Spatially resolved clonal copy number alterations in benign and malignant tissue

Andrew Erickson[1,15], Mengxiao He[2,15], Emelie Berglund[2,15], Maja Marklund[2], Reza Mirzazadeh[2], Niklas Schultz[3], Linda Kvastad[2], Alma Andersson[2], Ludvig Bergenstråhle[2], Joseph Bergenstråhle[2], Ludvig Larsson[2], Leire Alonso Galicia[2], Alia Shamikh[4,5], Elisa Basmaci[4,5], Teresita Díaz De Ståhl[4,5], Timothy Rajakumar[1], Dimitrios Doultsinos[1], Kim Thrane[2], Andrew L. Ji[6], Paul A. Khavari[6], Firaz Tarish[3], Anna Tanoglidi[7], Jonas Maaskola[2], Richard Colling[1,8], Tuomas Mirtti[9,10,11], Freddie C. Hamdy[1,12], Dan J. Woodcock[1,13], Thomas Helleday[3,14], Ian G. Mills[1], Alastair D. Lamb[1,12,16 ✉] & Joakim Lundeberg[2,16 ✉]

Defining the transition from benign to malignant tissue is fundamental to improving early diagnosis of cancer[1]. Here we use a systematic approach to study spatial genome integrity in situ and describe previously unidentified clonal relationships. We used spatially resolved transcriptomics[2] to infer spatial copy number variations in >120,000 regions across multiple organs, in benign and malignant tissues. We demonstrate that genome-wide copy number variation reveals distinct clonal patterns within tumours and in nearby benign tissue using an organ-wide approach focused on the prostate. Our results suggest a model for how genomic instability arises in histologically benign tissue that may represent early events in cancer evolution. We highlight the power of capturing the molecular and spatial continuums in a tissue context and challenge the rationale for treatment paradigms, including focal therapy.

Mutations can be either inherited or acquired (somatic). Inherited genomic polymorphisms are readily identifiable as these are present in all cells, whereas post-developmental somatic mutations are usually present in only a small fraction of cells[3]. To obtain spatial information about these rarer non-heritable genetic events, studies have commonly used laser-capture microdissection to retrieve histologically defined (or biomarker-defined) tissue regions or even single cells[1,4,5]. These studies have an inherent bias as only a limited number of spatial regions or single cells per tissue section can be collected and examined. The possibility to perform spatial genome analysis without being confined by histological boundaries would therefore provide an important contribution to delineating the clonal architecture in tumours and co-existing benign tissue.

## Inferred copy number variation predicts clonal hierarchies

Spatially resolved transcriptomics has emerged as a tool for genome-wide analysis of gene expression to explore tissues in an unsupervised manner[6]. In this study, we infer genome-wide copy number variations (CNVs) from spatially resolved mRNA profiles in situ (Fig. 1a). Gene expression has previously been used to infer CNVs in single cells, successfully identifying regions of chromosomal gain and loss[7]. Here we expand into a spatial modality, generating CNV calls in each spatial region represented

by barcoded spots. First, using unsupervised clustering methods, we sought corroboration that inferred CNV data (obtained using inferCNV[7]) could mirror DNA-based phylogenies, constructed using simultaneously extracted RNA and DNA from single cells[8] (Extended Data Fig. 1a). Next, we attempted to recapitulate published DNA-based phylogenies in prostate cancer using RNA from the same samples[9–11] (Extended Data Fig. 1b,c) and identified similarity between automated clone calling and published phylogenies. To ensure that inferCNV[7] could robustly capture sufficient and accurate CNV information for individual spots from a multifocal tumour model and enable us to deduce clonal relationships between cells, we designed an in silico system to synthesize a tissue containing multiple clones determined by stochastic copy number mutations in a single artificial chromosome. Using a probabilistic method to generate gene expression from such mutations, we then interrogated the expression data using spatial inferred CNVs (siCNVs), while blind to the underlying 'ground-truth' copy number status, and successfully recapitulated both the copy number status and the clonal groupings (Extended Data Fig. 2a–c).

## Organ-wide clonal landscape in the prostate

Next, we used a cross-section of an entire prostate organ to explore the siCNV landscape of a commonly multifocal malignancy[12]. The

[1]Nuffield Department of Surgical Sciences, University of Oxford, Oxford, UK. [2]Department of Gene Technology, KTH Royal Institute of Technology, Science for Life Laboratory, Solna, Sweden. [3]Science for Life Laboratory, Department of Oncology-Pathology, Karolinska Institutet, Solna, Sweden. [4]Department of Oncology-Pathology, Karolinska Institutet, Stockholm, Sweden. [5]Department of Clinical Pathology and Cytology, Karolinska University Hospital, Stockholm, Sweden. [6]Program in Epithelial Biology, Stanford University School of Medicine, Stanford, CA, USA. [7]Department of Clinical Pathology, University Uppsala Hospital, Uppsala, Sweden. [8]Department of Cellular Pathology, Oxford University Hospitals NHS Foundation Trust, Oxford, UK. [9]Department of Pathology, University of Helsinki & Helsinki University Hospital, Helsinki, Finland. [10]Research Program in Systems Oncology, Faculty of Medicine, University of Helsinki, Helsinki, Finland. [11]iCAN–Digital Precision Cancer Medicine Flagship, Helsinki, Finland. [12]Department of Urology, Oxford University Hospitals NHS Foundation Trust, Oxford, UK. [13]Big Data Institute, University of Oxford, Oxford, UK. [14]Weston Park Cancer Centre, Department of Oncology and Metabolism, University of Sheffield, Sheffield, UK. [15]These authors contributed equally: Andrew Erickson, Mengxiao He, Emelie Berglund. [16]These authors jointly supervised this work: Alastair D. Lamb, Joakim Lundeberg. ✉e-mail: alastair.lamb@nds.ox.ac.uk; joakim.lundeberg@scilifelab.se

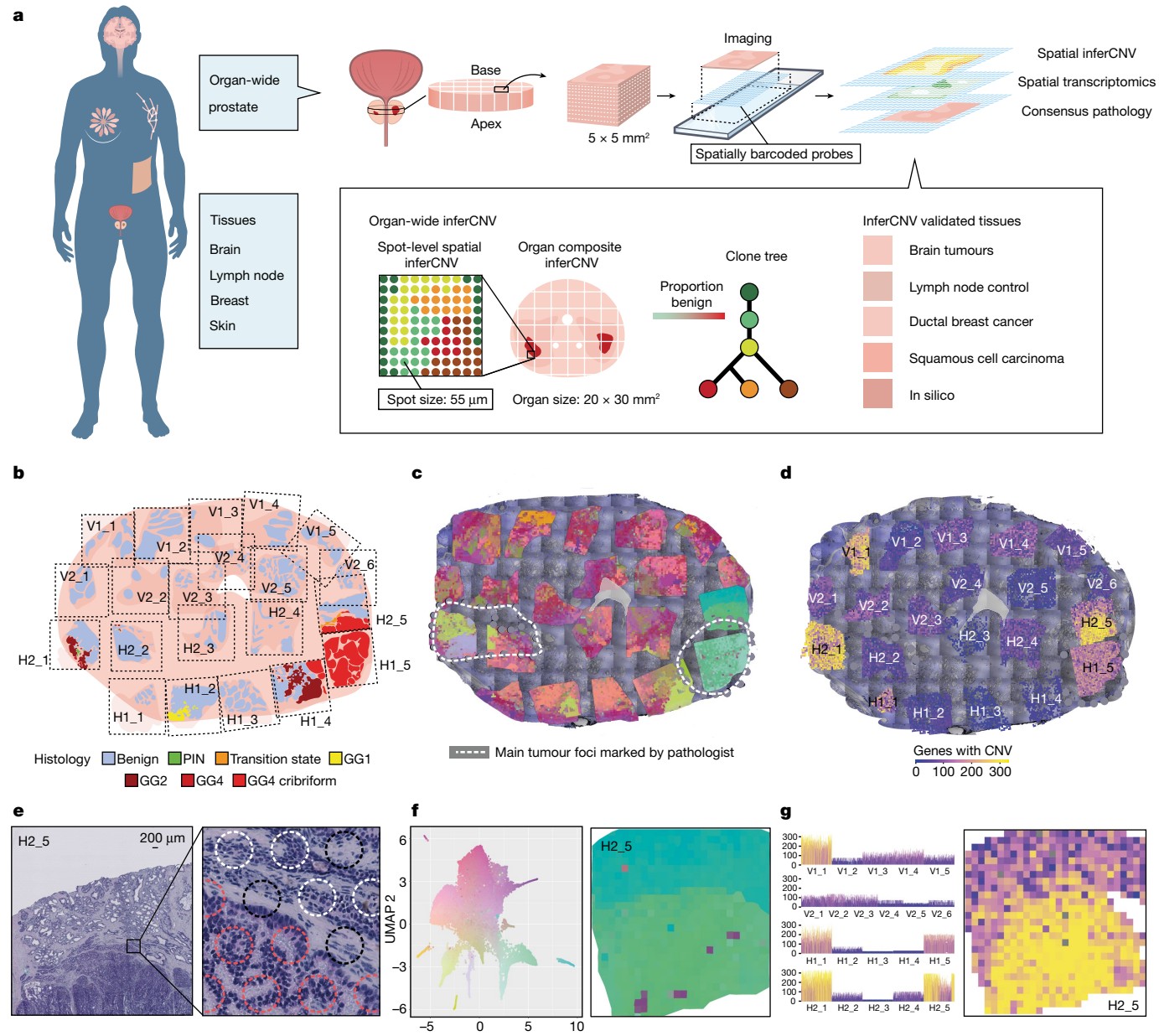

**Fig. 1 | Organ-wide spatial determination of transcript and CNV status.**
**a**, For organ-wide assessment, axial segments of the prostate were divided into $5 \times 5$ mm² blocks for spatial transcriptomic analysis with spatially barcoded probes. The resulting spatial gene expression profile was accompanied by an inferred copy number profile supported by spot-by-spot consensus pathology calls. Copy number features were used to detect clonal groups and instruct phylogenetic tree construction. Tissue-specific analyses of multiple phenotypes were performed. **b**, Histology status for each organ-wide section. Black dashed lines represent the area covered by the spatial transcriptomics array surface. GG, International Society of Urological Pathology (ISUP) Gleason 'grade group';

PIN, prostatic intra-epithelial neoplasia. **c**, Spatial distribution of gene expression (see **f**). **d**, Spatial distribution of summed copy number events (see **g**). **e**, Representative spot-level consensus pathology for section H2_5. Red circles indicate spots with ≥50% cancer cells, white circles indicate spots with ≥50% benign epithelium and black circles indicate spots with ≤50% of a single cell type. The diameter of the circles represents 55 µm. **f**, UMAP principal-component analysis of GEFs with a representative close-up for section H2_5. **g**, Total copy number events for each section with a representative close-up for section H2_5.

specimen was obtained by open radical prostatectomy from a patient with prostate cancer, and an axial section was taken from the mid-gland. The axial section was subdivided into cubes (Fig. 1a,b), and corresponding tissue sections were histologically graded using the Gleason grading system[13], identifying extensive intratumoral heterogeneity (ITH) in the context of surrounding benign tissue (Fig. 1b,e). We obtained organ-wide transcriptional information from 21 cubes (tissue sections) and >21,000 barcoded regions (100-µm-diameter spots) with a mean of

3,500 expressed genes detected per barcoded spot[2]. We then analysed the transcriptional data using factorized negative binomial regression (Extended Data Fig. 3a). This provided an unsupervised view of gene expression factors (GEFs)[14] over the cross-section of the prostate (Fig. 1c). Twenty-five factors showed overlap between histology and GEFs representing tumour, hyperplasia and benign epithelia annotated by the factor marker genes, as previously reported[14] (Fig. 1f). Next, we undertook an siCNV analysis to provide an overall landscape of

genome integrity (Fig. 1d), identifying certain regions with increased CNV activity (V1_1, H2_1, H1_1, H1_5 and H2_5; Fig. 1g) while the majority of the tissue area appeared to be copy number neutral. These initial results suggested that siCNVs could identify tissue regions, at organ scale, with inferred genomic variability, distinct from morphology or expression analysis.

To increase the fidelity of our analysis of variable siCNV regions, we took advantage of smaller 55-μm-diameter barcoded spots (Visium, 10x Genomics), reducing the number of cells to approximately 5–10 per spot, to perform a more detailed interrogation of seven key sections of siCNV activity. Two pathologists independently annotated each spot to provide consensus pathology and histology scoring (Fig. 1e). We first validated the increased precision of this higher-resolution platform using the synthetic tissue method (Extended Data Fig. 2d,e). We next obtained data from approximately 30,000 spots using factorized negative binomial regression, resulting in 24 spatially distinct GEFs (Extended Data Fig. 3b). We then examined clonal evolution patterns across the investigated tissue using siCNVs. Having established an association between GEFs and certain regions of interest (Fig. 1c,f), we wanted to determine the degree of clonal copy number heterogeneity in these regions. After designating all histologically benign spots as a reference set (Extended Data Fig. 3c), it was immediately apparent that, while certain GEFs displayed a fairly homogenous inferred genotype (for example, GEFs 7, 14 and 22; Extended Data Fig. 3d), others were notably heterogeneous (for example, GEF 10; Extended Data Fig. 3e).

Prompted by the realization that certain regions annotated as histologically benign displayed copy number heterogeneity (Fig. 1d)[15–18], we refined the reference set to spots that were both histologically benign (outside the regions of interest) and lacked any siCNV (Extended Data Fig. 4). This constituted a 'pure benign' reference set for all subsequent siCNV analyses, unique to each patient. In cancer-wide inferred genotypes (Fig. 2a–e), there was evidence of clonally distributed copy number heterogeneity within areas of spatially homogeneous Gleason patterns (Fig. 2a,d,e). We constructed a phylogenetic tree to describe sequential clonal events versus independently arising cancer clones (Fig. 2b). Two cancer clones (clones A and B) lacked key truncal events, including loss of regions on chromosomes 16q and 8p, that were otherwise ubiquitous across all cancer clones (Fig. 2a,b). These clones were spatially restricted to section H1_2 containing a region of low-grade Gleason grade group 1 (discussed later). The majority of clonally related spots were located around the largest focus of Gleason grade group 4 disease with a notable pattern of truncal and branching events (clones H, I, J and K). We therefore focused on this dominant region of cancer (spanning sections H1_4, H1_5 and H2_5), to establish a first view of the interplay between spatial architecture and clonal dynamics (remaining sections in Extended Data Fig. 5a,b).

To construct clone trees, we assumed that (1) groups of cells containing identical copy number profiles were more likely to be related than to have arisen by chance and (2) somatic copy number events were acquired sequentially over time (the more numerous the events, the more distinct the clone). We cannot conclusively rule out the possibility that smaller clones may represent clone cell mixtures due to the inherent size of the Visium spots. However, using this approach, we observed a common ancestral clone (clone H; Fig. 2b) containing truncal events including copy number loss on chromosomes 6q and 16q and copy number gain on chromosomes 12q and 16q. These events were clearly located in two tissue regions: an area of Gleason grade group 2 on the medial side of the main tumour focus (section H1_4) and a region described as 'transition state' by consensus pathology at the upper mid-edge (section H2_5). These conserved siCNV features in distinct spatial locations are noteworthy. A possible explanation is that clone H represents a linear sequence of branching morphology in the prostatic glandular system[1] and that further somatic events took place, giving rise to clones I, J and K and forming a high-grade tumour focus (Fig. 2b), which pushed apart the branching histology owing to

an aggressive expansile phenotype. We thus have a spatial imprint of these events in prostate tissue. We also propose that some CNVs may be of particular pathological significance (Extended Data Fig. 4d) based on spatial molecular phylogeny. Our analysis therefore provides insight into processes of tumour clonal evolution, identifying discriminating events by spot-level CNV calling in a spatial context.

## Somatic clones cross histological boundaries

Given this discovery of a discordance between cellular phenotype and inferred genotype, we then undertook a detailed interrogation of section H2_1 in the left peripheral zone of the prostate (Figs. 1c and 2c) containing roughly equal proportions of cancer and benign tissue. We profiled the copy number status of every spot in this section and ordered these spots by hierarchical clustering into 'clones' A to G on the basis of defined levels of cluster separation (Fig. 3a,b). Spatially, we observed that these data-driven clone clusters were located in groups, broadly correlating with histological subtype, but with some important distinctions (Fig. 3c,d). We observed that many CNVs had already occurred in benign tissue (clone C; Fig. 3a–d), most notably on chromosomes 8 and 10, which has been well described in aggressive prostate cancer, including the oncogene *MYC* and tumour-suppressor gene *PTEN*[19–21], but also several other copy number gains and losses. Spatially, this clone constituted a region of exclusively benign acinar cells branching off a duct lined by largely copy number-neutral cells in nearby clones A and B (Fig. 3d). The unobserved ancestor to clone C gave rise to a further unobserved clone followed by cancer-containing clones E, F and G. Whereas clone G was made up exclusively of Gleason grade group 2 cancer cells, clones E and F were mixed, with up to 25% benign cells (Fig. 3d). The presence of somatic events in histologically benign cells highlights that these clone groups traverse histological boundaries.

To validate that this inferred copy number status was truly representative of underlying genotype, we used fluorescence in situ hybridization (FISH) probes to target two specific genes of discriminatory interest, *MYC* and *PTEN*, encompassed in the notable chromosomal changes in benign tissue clone C as well as high-grade tumour clones, but absent in low-grade disease. This confirmed that, whereas the status of both genes was diploid in normal benign tissue (clone A), *MYC* amplification and *PTEN* loss were evident in altered benign (clone C) as well as tumour (clone F) clones (Fig. 3e and Extended Data Fig. 6). Going forward, we propose that other homogenous inferCNV calls are accurate, on the basis of the evidence provided by these two selected loci. This evidence suggests that somatic events, creating a mosaic of branching clones during ductal morphogenesis, are present even in histologically benign disease. It therefore follows that an understanding of this somatic mosaicism could distinguish which regions of benign glandular tissue may give rise to lethal cancer and which will not.

We recognize that a limitation of using siCNVs is that this approach does not capture mutations such as single-nucleotide variants (SNVs) or other copy number-neutral events, which could add value in discriminating clonal groupings. We therefore undertook an analysis of transcribed (exonic) single-nucleotide polymorphisms (SNPs) using cb_sniffer[22]. Analyses of the ratios of clonal variant allele fractions for both specific events with high-coverage SNPs (exemplified by chr8:143580183 and chr8:99892049; Extended Data Fig. 7) supported copy number events on the same allele, in line with shared ancestry (Fig. 3b).

Having established the clonal subgroups in this heterogeneous section of prostate tissue, we used differential expression analysis to investigate potential functional alterations unique to these cellular groups. Focusing on clone C of altered benign cells, we observed upregulation of *MYC* activity (Extended Data Fig. 8c) as well as pathways responsible for phenotypic versatility[23] (Extended Data Fig. 8b) when compared with diploid benign cells (clone A). Furthermore,

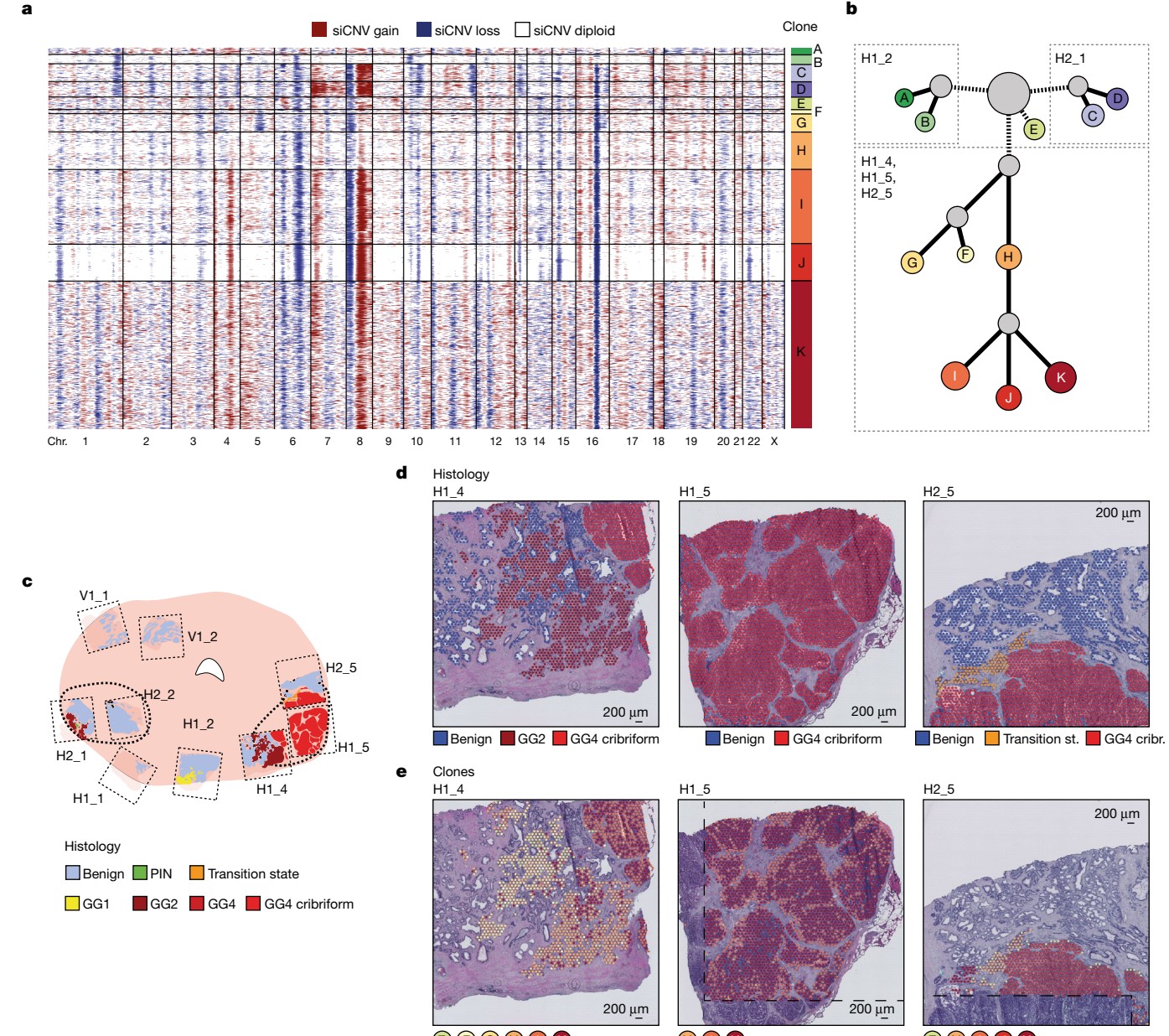

**Fig. 2 | Specific somatic copy number alterations in all cancer organ-wide analysis. a**, Genome-wide derived analysis (siCNVs) for all Visium spots harbouring tumour from prostate patient 1. Clonal groupings of spots (with approximately 10–15 cells each) were determined by hierarchical clustering. Chr., chromosome. **b**, Phylogenetic clone tree of the tumour clones from **a**, with grey clones representing unobserved, inferred common ancestors. Clone circle area is proportional to the number of spots and branch length was determined by weighted quantity of CNVs (both on a logarithmic scale). siCNV changes for each clone are available in Supplementary Table 1. **c**, Representation of all tissue sections from prostate patient 1. Thicker black lines denote original boundaries annotated by initial clinical pathology. **d**, Consensus epithelial histological annotations for sections H1_4, H1_5 and H2_5, corresponding to the right tumour focus. **e**, Spatial visualization of tumour clones (from **a**). The dashed lines mark areas where no spatial transcriptomics data were obtained owing to these regions being outside of barcoded array surfaces.

there was downregulation of conventional androgen receptor (AR) target genes (for example, *KLK2*, *KLK3*, *FKBP5* and *NKX3-1*), raising the hypothesis of a reduced (or altered) dependence on AR regulation in these cells[24]. We also investigated the distinction between clone C and clones containing histologically transformed cells (clones E–G). We observed reduced *MSMB* and increased *GDF15* expression in both groups (Extended Data Fig. 8a,d), which are normally thought to be pathognomonic of malignantly transformed cells[21,25]. When analysing differentially expressed genes found in only altered benign cells, we observed an enrichment for genes associated with oxidative phosphorylation and mitochondrial energy metabolism as well as protein

stabilization (Supplementary Table 5), in line with cells trying to cope with extrinsic and intrinsic stress.

We considered the place of branching morphogenesis in the sequential acquisition of transformative events in a predominantly benign section of the prostate (section H2_1 as well as section H2_2; Extended Data Fig. 9). Here we noted that such events seemed to occur during the development of prostatic ducts and acinar branches, with changes occurring at key branching points, and the altered genotype was passed on to daughter cells lining the ducts and glands of associated branches. Interestingly, not all cells in such branches displayed the same cellular structure, raising important questions as to why epithelial glands

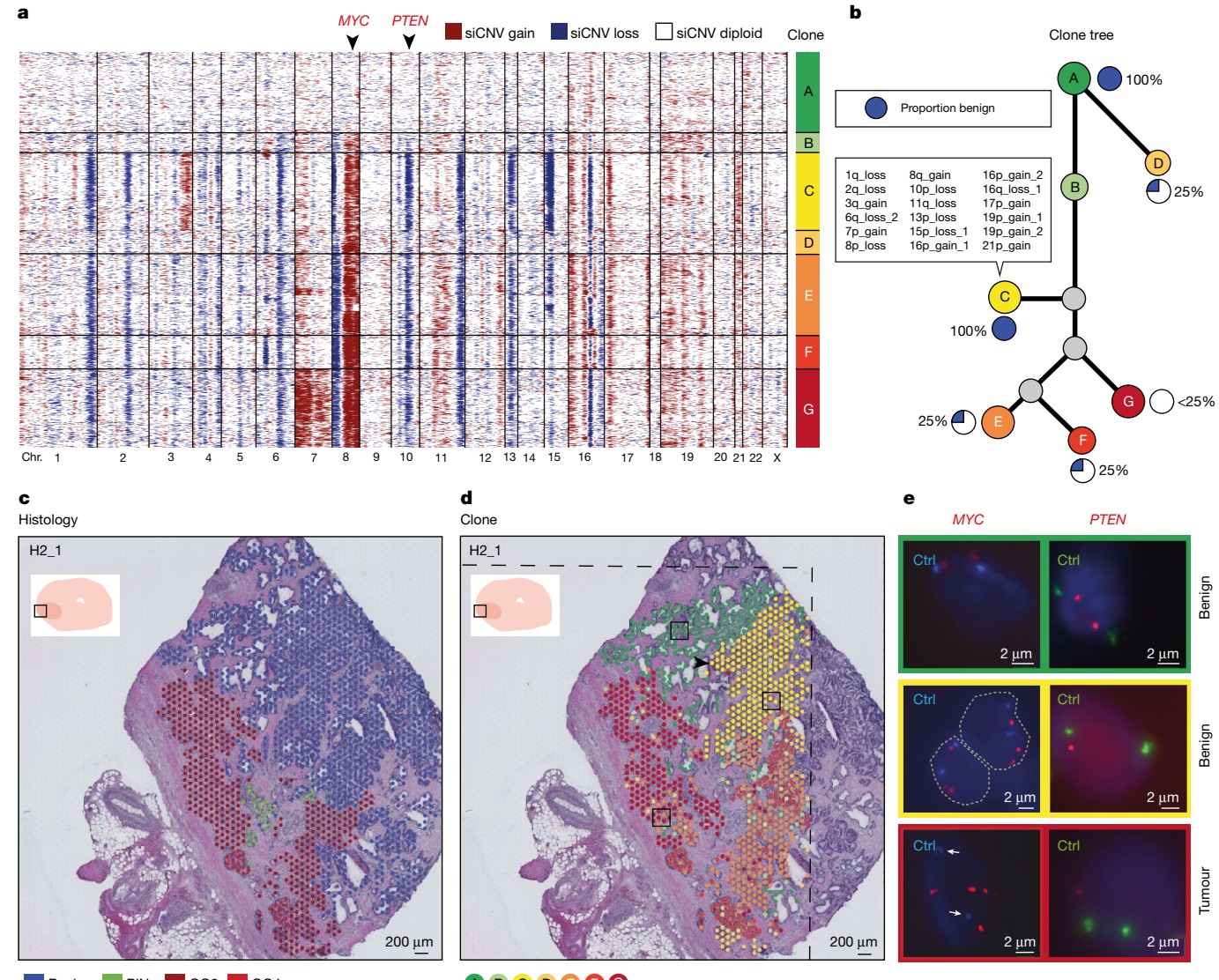

**Fig. 3 | Somatic events in both cancer and benign prostate epithelium.**
**a**, Genome-wide derived CNV analysis (siCNVs) for each barcoded high-resolution spatial transcriptomic spot from section H2_1, which contained a mixture of tumour and benign epithelia (red, gain; blue, loss). Clonal groupings of spots (approximately 10–15 cells each) were determined by hierarchical clustering. On the basis of SNV analysis (Extended Data Fig. 7), clone A probably represents a polyclonal population of diploid cells. **b**, Phylogenetic clone tree of all clones from **a**. The proportion of benign epithelial cells in each clone was as indicated. Specific CNV locations unique to clone C are listed (summarized by the number of the chromosome where the event was located, p/q arm and gain/loss; the remainder of siCNV changes are given in Supplementary Table 2). **c**, Spatial visualization of the histopathological status of each spot. Each spot

with seemingly identical inferred genotypes might display divergent histological phenotypes.

In view of the above findings, we considered that analysis of the inferred genotype of low-grade cancer might reveal important differences from that of high-grade cancer. Section H1_2 contained a region of Gleason grade group 1 prostate cancer (Extended Data Fig. 5d). As noted previously, there were two clones (Fig. 2a, clones A and B) that lacked key changes on both chromosomes 8 and 16, with little in common with other cancer-bearing clones (Fig. 2c). A spot-wise re-analysis of section H1_2 (including benign spots) showed that these two clones, now labelled F and G, were spatially grouped as two approximately equal halves of this region of Gleason grade group 1 cancer (Extended

was assessed by two pathologists for consensus annotation, with only spots with >50% cellularity included. **d**, Spatial visualization of the clone status of each spot. Clonal groupings cross histological boundaries. The branching point of the prostatic duct (arrowhead) represents a possible site of somatic events arising in clone C (see also Extended Data Fig. 6). The dashed lines mark areas where no spatial transcriptomics data were obtained owing to the region being outside of barcoded array surfaces. **e**, FISH validation of two siCNVs: *MYC*, from chromosome 8q, and *PTEN*, from chromosome 10p (arrowheads in **a**). Control probes (Ctrl) targeted centromeres for chromosomes 8 and 10, respectively. All FISH panels depict single cells with one exception, where dashed grey lines highlight the nucleus borders and the presence of two cells. White arrows indicate the location of centromere controls.

Data Fig. 5c,d). This is evidence that low-grade prostate cancer is indeed fundamentally distinct from high-grade disease[26] and raises the hypothesis that such cancer cannot become higher grade because it lacks essential somatic events.

## Clonal heterogeneity in multiple tissues

To corroborate our findings, we first performed validation through an additional 37,000 spots from a cross-section of a further prostatectomy that confirmed the spatial continuum of benign clones in proximity to cancer with shared truncal events. We also confirmed the high degree of ITH of siCNV clones within prostate tumour loci (Extended Data Fig. 10)

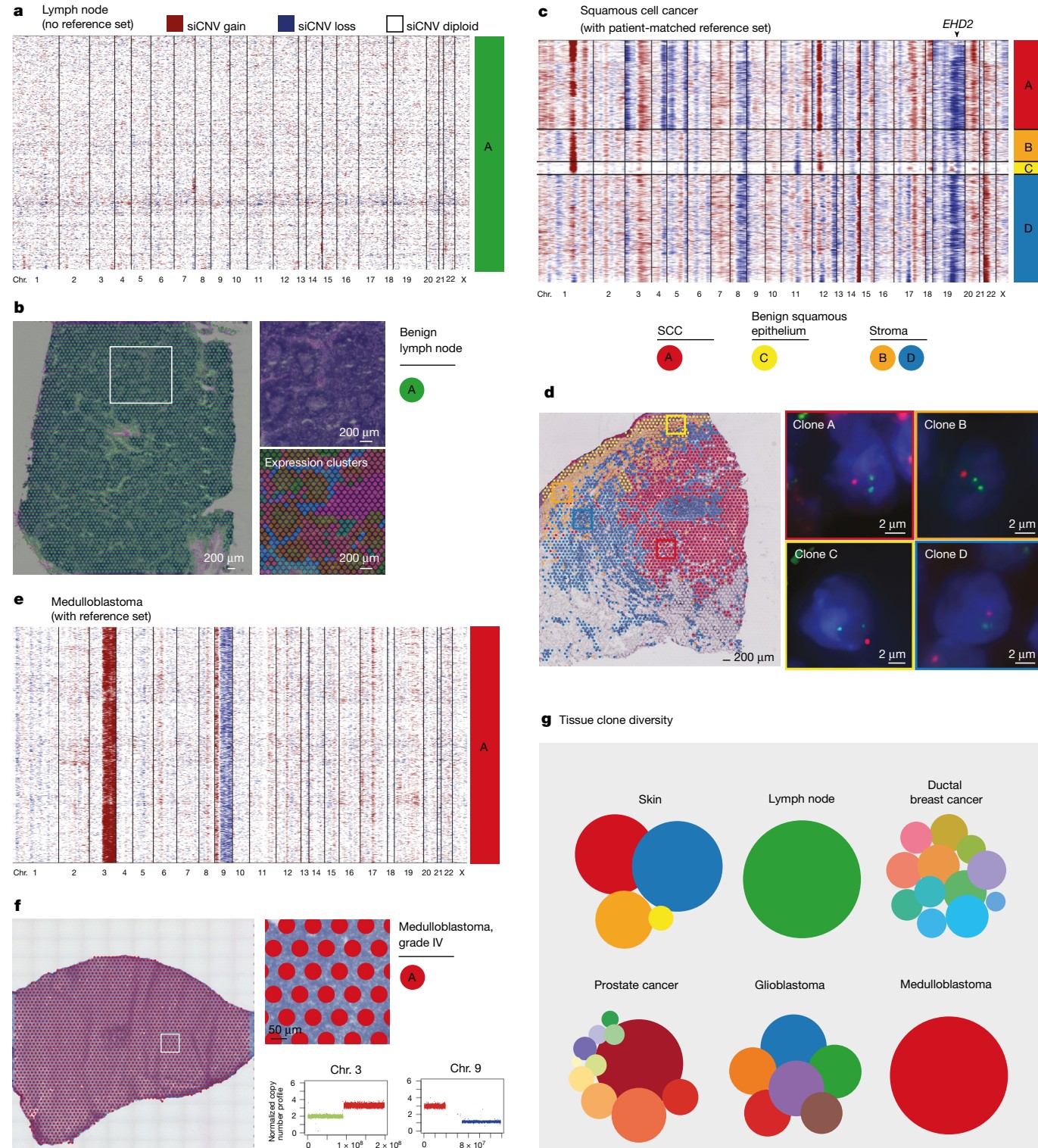

**Fig. 4 | Somatic copy number alterations in cancer and benign histologies.**
**a,b**, Benign lymph node with distinct histological features and gene expression heterogeneity (**b**) harbouring no detected copy number alterations (**a**). Gene expression clusters were determined by UMAP analysis in **b. c,d**, Skin containing SCC (clone A, red) as well as benign squamous epithelium (clone C, yellow). A subset of somatic events visualized in cancer clone A are also detected in adjacent benign epithelial clone C. **d**, FISH validation of the siCNV encompassing *EHD2* on chromosome 19q (arrowhead in **c**). **e,f**, Monoclonal childhood medulloblastoma. siCNVs on chromosomes 3 and 9 (**e**) were corroborated by copy number calls from WGS (**f**, lower right). **g**, Clone distribution for each tissue type. Circle area corresponds to the number of spots per clone. Results from ductal breast cancer and glioblastoma are available in Extended Data Fig. 11.

and the presence of key somatic events in benign prostate glands. We then generalized our findings in multiple organs (Fig. 4 and Extended Data Fig. 11). First, we analysed a benign lymph node displaying distinct gene expression clusters for different histological entities (such as germinal centres), and siCNV analysis provided, as expected, a copy number-neutral profile for the entire tissue section (Fig. 4a,b). This

provided further validation that siCNV clones are distinct from gene expression. We next analysed skin tissue containing both benign squamous epithelia and squamous cell carcinoma (SCC). For this, we obtained a patient-matched benign reference set of RNA-sequenced single skin cells with confirmation from adjacent sections of benign histology[27]. siCNV analysis identified four clones within the tissue, one of which corresponded to SCC, containing several copy number events. Notably, two key events (partial chromosome 1 and 12 gain) were shared with another nearby clone composed entirely of histologically benign tissue (Fig. 4c,d). Additional validation of siCNV signals was obtained by DNA FISH with three probes, for chromosome 1q gain (*CKS1B*), chromosome 8q loss (*MYC*) and chromosome 19q loss (*EHD2*), of consecutive sections of the SCC sample. We found that siCNV analysis correctly predicted CNV status in 91% ($n = 11/12$) of spatial clonal regions (Fig. 4d, Extended Data Fig. 12 and additional data hosted on Mendeley (https://doi.org/10.17632/svw96g68dv.1)). This substantiates our finding of siCNV clones traversing histological boundaries for an additional tumour type. To contrast these observations, we performed analysis of a Sonic hedgehog (SHH)-driven paediatric medulloblastoma (Fig. 4e,f) with sex- and age-matched samples. The results showed a uniformly homogeneous spatial inferCNV clone type throughout the tumour with key expected genetic alterations such as 3q gain (encompassing *PIK3CA*) and a 9q deletion (encompassing *PTCH1*) as well as a short gain on 9p. These homogenous findings were validated by whole-genome sequencing (WGS) of the tumour, in which distinct CNV calls were found for the three altered chromosomal regions identified by our siCNV analysis (Extended Data Fig. 13). We further analysed two additional tumour types without reference sets: ductal breast cancer and an adult glioblastoma (Extended Data Fig. 11). Here we confirmed a multifaceted spatial siCNV tumour landscape with multiple co-existing clone types in tumour tissue of histologically similar appearance. For example, in ductal breast cancer (Extended Data Fig. 11k,l), we observed two distinct clone types (C and F), separated by stroma, with little or no CNV overlap. In the glioblastoma tissue, we similarly identified five clone types that had sharp spatial demarcations separating the siCNV clones, despite being histologically similar (Extended Data Fig. 11m,n). Overall, the clonal appearances of ITH were clear as was the overlap with tumour morphology.

The tissue clone diversity over the five investigated tissue types was notably variable, with genomes ranging from homogenous to highly heterogeneous in both tumours and benign tissue (Fig. 4g). We therefore believe that combining siCNV information with spatial gene expression patterns, which provide some functional understanding, and cell type mapping (using single-cell RNA-seq (scRNA-seq)) could enable targeted treatment options for individual clones, 'benign' or tumour, that would not be easily attainable by any other means. Such targeted approaches could include a more intelligent rationale for focal therapy or, for systemic therapy, could facilitate the identification of such clones by 'liquid biopsy'.

## Discussion

We show that spatial transcriptomic data across multiple cancer types can robustly be used to infer CNV, as validated by FISH and WGS. Specifically, we performed an in-depth spatial analysis of the prostate organ that generated an unprecedented atlas of up to 50,000 tissue domains in a single patient and 120,000 tissue domains across ten patients. For these domains, we inferred genome-wide information in each spot, which facilitated data-driven clone generation in a tissue-wide fashion at high resolution. Notably, the spatial information allowed us to identify small clonal units not evident from morphology, which would therefore be overlooked by histologically guided laser microdissection or even random sampling of single cells. We go on to show that, in some tumour types, particularly in prostate, glioma and breast cancers, CNV analysis identifies distinct clonal patterns within tumours, in line with

a recent spatial genome methodology that has also shown granularity in the study of multiclonality of tumours[28].

Focusing on prostate cancer, the patterns, as defined by the conservation of CNVs across morphological entities, indicate hitherto unappreciated molecular relationships between histologically benign and cancerous regions. It is known that CNVs occur early in tumorigenesis[21]. We propose that CNVs can precede tumorigenesis and are a feature of glandular morphogenesis, with propagation of particular variants traversing disease pathology. It seems that clonal status alone and the copy number alterations described here retained in heritable clonal lineages at cell division are insufficient to deliver immediate phenotypic transformation. We believe that our work generates interesting hypotheses regarding epigenetic determinism[29] and the environmental effect with, for example, the stromal niche or cross-talk between neighbouring clones. Furthermore, questions remain about the timing of events and how long is needed for morphological transformation to occur. Expression analysis of altered benign clones identified changes consistent with enhanced phenotypic versatility, suggesting that these cells may represent an intermediate state between benign and malignant cells—metabolically active as they try to survive the mutational burden they have acquired, before phenotypic transformation. In summary, this study shows that CNVs in regions of the genome that encode certain cancer drivers (for example, *MYC* and *PTEN*) are truly early events, occurring in tissue regions currently unknown to and therefore ignored by pathologists (Extended Data Fig. 4d). This is important given that the risk stratification delivered by pathologists dictates to a large degree treatment decisions and subsequent clinical outcome.

Our study therefore provides an unbiased avenue to interrogate genomic integrity, adding to the armamentarium of cancer molecular pathology. Our findings provide a basis for improved early detection of clinically important cancers, targeted focal and systemic therapy, and improved patient outcomes for ubiquitous malignancies such as prostate cancer. Overall, our study raises important biological questions about cancer evolution, somatic mosaicism and tissue development.

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

## Methods

### Ethics declaration

The study was performed according to the Declaration of Helsinki, Basel Declaration and Good Clinical Practice. The study was approved by the Regional Ethical Review Board (REPN) Uppsala, Sweden, before study initiation (Dnr 2011/066/2, Landstinget Västmanland, S. Stenius) and by the Regional Ethical Review Board (EPN), Stockholm, Sweden (DNR 2018/3-31, M. Nistér). All patients were provided with full and adequate verbal and written information about the study before their participation. Written informed consent was obtained from all participating individuals before enrolment in the study.

### Tissue specimens

Whole prostates were obtained by open radical prostatectomy at Västerås Hospital. Prostate patient 1 was 82 years old, and prostate patient 2 was 63 years old. Both had reported Gleason scores of 4+3 (ISUP grade group 3) at initial biopsy, and the prostatectomy pathology was ISUP grade group 4 for patient 1 and ISUP grade group 3 for patient 2. Each prostate was divided into half by a horizontal cut, and the upper part (closest to the patient's head) was used and cut on a stepped 5-mm mould to obtain a 5-mm-high cylinder. Next, stripes were cut from the cylinder, and each stripe was cut into smaller cubes (total of 21 for patient 1 and 28 for patient 2). All tissue cubes were fresh-frozen in liquid nitrogen and stored at −80 °C until embedding for cryosectioning. The childhood brain tumours were collected and provided by the Swedish Childhood Tumour Biobank and stored at −80 °C until embedding for cryosectioning.

### Datasets

Human SCC and case-matched dissociated normal skin cells (reference set) were obtained from a published dataset[27]. The human lymph node, human adult glioblastoma multiforme (tumour grade IV) and human breast cancer (ductal carcinoma in situ, lobular carcinoma in situ, invasive carcinoma) datasets were provided by 10x Genomics (https://support.10xgenomics.com/spatial-gene-expression/datasets).

### Spatial transcriptomics (1k arrays)

For prostate (patient 1), all 21 tissue cubes were cryosectioned into 10-μm sections from the bottom (two sections per cube) for spatial transcriptomics analysis. The sections were mounted onto spatially barcoded microarray slides. The protocol described in refs. [2,30] was used to prepare all mounted sections with a few modifications. Fixation was performed for 10 min at room temperature, and samples were permeabilized using exonuclease I buffer for 30 min at 37 °C and 0.1× pepsin (pH 1) for 10 min at 37 °C. The material was processed into libraries as described in ref. [31] and sequenced on an Illumina NovaSeq instrument using paired-end 300-bp reads.

### Spatial transcriptomics (10x Genomics Visium)

The Visium Spatial Tissue Optimization Slide & Reagent kit (10x Genomics) was used to optimize permeabilization conditions for the tissue sections. One 10-μm section from each patient was processed according to the manufacturer's instructions. Spatially barcoded cDNA from every tissue section was generated using the Visium Spatial Gene Expression Slide & Reagent kit (10x Genomics). Tissue sections from prostate patient 1 were fixed according to the manufacturer's instructions, and permeabilization was performed for 8 min. Sections from prostate patient 2 were fixed for 10 min using acetone at −20 °C and permeabilized for 15 min. Childhood brain tumour sections of 12 μm were permeabilized for 30 min. Libraries for all tissue sections were generated following the 10x Genomics Visium library preparation protocol and sequenced on Illumina sequencing instruments.

### Spatial transcriptomics data processing

For 1k arrays, FASTQ files were processed using ST Pipeline v.1.5.1 software[32]. Transcripts were mapped with STAR[33] to the GRCh38.79 human reference genome. Mapped reads were counted using the HTseq count tool[34]. Spatial barcodes were demultiplexed using an implementation of TagGD UMI filtering[35] carried out to remove duplicated reads. A mean of 3,582 unique genes and 10,734 unique transcripts was obtained per spot after removing spots with fewer than 100 genes or transcripts.

For 10x Visium arrays, specifics regarding data processing before data analysis after demultiplexing of FASTQ files have been described elsewhere for the human SCC specimen[27] and datasets provided by 10x Genomics (https://support.10xgenomics.com/spatial-gene-expression/datasets). For the childhood brain tumour, read 2 was trimmed to remove both the TSO adaptor sequence and poly(A) homopolymers using Cutadapt[36]. Trimmed fastq files were then run through Space Ranger (version 1.0.0, 10x Genomics) where reads were mapped to the human reference genome (GRCh38, release 93). The raw sequencing reads for the prostate samples were directly processed using Space Ranger (version 1.0.0 for prostate 1 and version 1.2.1 for prostate 2; 10x Genomics) and mapped using the same human reference genome as above. A mean of 2,334 of the 2,104 unique genes and 10,221 of the 5,711 unique transcripts was obtained per spot after removing spots with fewer than 100 genes or transcripts for patient 1 or patient 2.

### Factorized negative binomial regression of prostate samples

GEF analysis was performed as previously described[14]. In all analyses, we factorized the data into $T$ = 25, 24 and 20 GEFs (1k, Visium patient 1 and Visium patient 2) and ran the optimization for 5,000 iterations. Convergence was assessed by tracking the loss (negative unnormalized log posterior), which had plateaued by 5,000 iterations for all analyses. Spots were annotated on the basis of their section to control for sample-wise batch effects.

### Processing and visualization of non-prostate samples

Data processing and visualization were carried out using the Seurat (version 3.2.2)[37] and STUtility (version 0.1.0)[38] R packages. UMI counts were filtered using the InputFromTable function, and genes were removed if they were present in fewer than five spots or had a total UMI count below 100. All spots containing fewer than 500 UMI counts were also removed. Counts were normalized using SCTransform, and dimensionality reduction was performed using principal-component analysis. The top 20 principal components were used for all samples except the childhood brain tumour, where 10 components were used. Expression-based clustering was performed by constructing a shared nearest neighbour (SNN) graph through FindNeighbors using previously established components and clusters identified through FindClusters. The resolution parameter was set to 0.8 for all samples except the childhood brain tumour, for which 0.2 was used. Finally, a two-dimensional UMAP embedding was constructed from the previously established top principal components for each tissue type. For the human lymph node specimen, differentially expressed genes for each cluster were determined using the FindAllMarkers function, only testing genes detected in at least 25% of the spots in either of the two populations, that is, cluster or background.

### Paediatric tumour DNA sequencing and data analysis

Libraries for WGS were prepared using Illumina TruSeq PCR-free reagents. WGS samples were sequenced using 2 × 150-bp paired-end reads, on a HiSeqX v2.5 (patient 1) or NovaSeq 6000 (patients 2 and 3) instrument (Illumina). DNA sequence data were processed with Sarek, following the GATK best-practice recommendations[39], on UPPMAX Clusters at Uppsala University (https://www.uppmax.uu.se/resources/systems/the-bianca-cluster/). In brief, the steps run were quality control

of the FASTQ files using FastQC (https://www.bioinformatics.babra-ham.ac.uk/projects/fastqc/), alignment of short reads to the human reference genome sequence (GRCh38/hg38) using bwa-mem with the ALT-aware option turned on[40], sorting of reads and marking of PCR duplicates with GATK MarkDuplicates and base quality score recalibration and joint realignment of reads around indels using GATK tools (https://github.com/broadinstitute/gatk). Tumour CNV profiles were generated using Control-FREEC[41]. The matched normal sample was used to call somatic CNVs.

## DNA FISH

An optimal cutting temperature (OCT)-embedded block of fresh-frozen prostate sample was sectioned at 5-μm thickness, and several consecutive sections were mounted on positively charged microscope slides (VWR) and placed at −80 °C until processing. Sections were fixed with methanol and acetic acid (3:1 ratio) for 15 min at room temperature, washed in 1× PBS and briefly air dried, followed by haematoxylin and eosin staining and imaging. DNA FISH probes targeting *MYC*/8cent (Cytocell, MPD28000), *PTEN*/10cent (Cytocell, MPD15000), *CKS1B*/1cent (Cytocell, LPH039) or *EHD2*/19cent (Cytocell, LPS047) were added (10–15 μl) on top of the tissue sections, and sections were sandwiched with 18 × 18 coverslips and sealed with rubber glue (BioNordika, PCN009). Slides were placed on a hot plate for exactly 6 min at 76 °C to denature DNA molecules and immediately placed inside an incubator with 100% humidity for overnight incubation at 37 °C. Coverslips were gently removed, and slides were washed in a ceramic jar containing prewarmed 0.4× SSC for 3 min at 72 °C, transferred to 2× SSC with 0.05 Tween-20 for 2 min at room temperature and then quickly washed in 2× SSC and nuclease-free water. To reduce the autofluorescence background, we applied quenching probes (Thermo Fisher Scientific, R37630) to the top of sections, incubated sections for 5 min at room temperature and washed in 1× PBS. Nuclei were then counterstained with DAPI and slides were mounted using mounting medium (Thermo Fisher Scientific, S36936). Microscopy images were acquired using a ×100/1.45-NA objective mounted on an Eclipse upright microscope system (Nikon) controlled by NIS Elements. We collected multiple image stacks per sample, each consisting of 30–40 focal planes spaced 0.3 μm apart.

## Pathologist workflow: spot-level annotation for prostate patient 1

All Visium spots were annotated on a spot-by-spot basis using Loupe Browser version 5.0 (10x Genomics) for the Visium sections by two uropathologists (R.C. and T.M.). Using a cell-type specific coverage threshold of >50%, the pathologists annotated spots by histological class or as 'exclude' (for example, for mixed coverage, when array regions did not cover tissue such as lumens or if a scanning/sectioning artefact rendered it impossible to determine a histological class). The annotations were cleaned, unified and visualized in Loupe Browser for review. Next, a consensus workflow was applied wherein the pathologists were asked to determine a final annotation class if there were discrepancies between benign or cancerous luminal epithelial cells. If there were discrepancies between luminal classes and stroma, A.E. performed a review and reclassification, such that if over 50% of cells of one class could be identified the spot was marked as the corresponding class. If there was uncertainty, the spot was marked as 'mixed' and excluded from downstream analysis. The final consensus annotation dataset consisted of a total *n* of 23,282 spots. We defined low-grade prostate cancers as spots with Gleason grade group 1 and high-grade cancer as spots containing Gleason pattern 4. Final confirmation of benign annotations in regions of tissue harbouring inferred CNVs (Fig. 3, clone C) was performed by assessing digital images of p63/AMACR staining from consecutive tissue sections, with detection of the presence or absence of basal cells by p63 positivity (thus indicating whether the region of interest was benign or tumour). High-resolution images of staining results can be found in the Mendeley repository.

## Pathologist workflow: annotation for prostate patient 2

Prostatic luminal epithelial cells were annotated for 15 Visium sections from prostate 2 for the presence of tumour histology. Luminal epithelial spots from benign tissue sections were analysed for selection of a benign reference set. Tumour histology was confirmed in sections H3_1, H2_1, H2_2 and H3_6 using Loupe Brower.

## Data preprocessing for inferring spatial CNVs

To systematically interrogate the data, we developed an R package called SpatialInferCNV (https://github.com/aerickso/SpatialInferCNV). Additional analyses were performed using a series of R packages (tidyverse, Seurat, infercnv and hdf5r) and Python and BASH scripts as follows. Histological annotations were imported from the final consensus annotation files for all sections, and barcodes were appended with their section identifier. Next, the annotations were filtered for a given feature of interest. Files output from the Cell Ranger pipeline (filtered_feature_bc_matrix.h5) were imported, and barcodes were appended with their corresponding section name. The count files were then filtered for only those within the analysis of interest. The count files further underwent a quality-control filter[8] wherein spots containing 500 or fewer counts were removed. The annotations file and counts file were joined for each section, and the resulting files were then all combined into a final matrix that was output (.tsv file) for downstream analysis with inferCNV[7]. The barcodes for only spots that passed the annotation and quality-control filters were merged again with the annotations, and these were separately exported (.tsv file) for further inferCNV[7] analysis. Lastly, a genomic positions file was created following the instructions at https://github.com/broadinstitute/inferCNV/wiki/instructions-create-genome-position-file.

## Selection of benign references

Inputs to inferCNV[7] can include a reference set of UMI-barcoded objects, to improve precise inference of genomic copy number events in the observed population. We first performed an unsupervised analysis of only the benign luminal epithelial reference cells (parameter for inferCNV object: ref_group_names = NULL; parameters for run: cutoff = 0.1, cluster_by_groups = FALSE, denoise = TRUE). Using the denoised outputs, we identified by visual inspection a subgroup of all benign spots that harboured few to no inferred CNVs (Extended Data Fig. 4). The associated dendrogram file (with the cluster structure and each barcode therein) was then further analysed for node selection.

## InferCNV parameters

For unsupervised siCNV analysis, we included the following parameter for the function CreateInfercnvObject(): chr_exclude = c("chrM"). For the run() function, we used the following parameter values: cutoff = 0.1, num_threads = 10, cluster_by_groups = FALSE, denoise = TRUE, HMM = FALSE. A reference set was used for all analyses, with the exceptions of defining the reference set or if a suitable reference set was not available (Fig. 4 and Extended Data Fig. 11).

In supervised siCNV analysis (to call inferCNV[7] hidden Markov model (HMM) functions), inferCNV[7] was run as follows. The node identity file was used in place of the annotation file. The following inferCNV run parameters were used: cutoff = 0.1, num_threads = 10, cluster_by_groups = TRUE, denoise = TRUE, HMM = TRUE.

For the global visualization of siCNV events in Fig. 1, we analysed spatial transcriptomics (1k arrays) data with inferCNV[7] for all 21 sections in a global analysis without a reference set. We performed the analysis such that each individual spatial transcriptomics spot was run with the following inferCNV run() parameters: cutoff = 0.1, num_threads = 10, cluster_by_groups = FALSE, denoise=TRUE, HMM=TRUE, analysis_mode="cells," HMM_report_by="cell". To spatially visualize global siCNV profiles across an entire prostate, we then determined the number of

individual genes detected to harbour an inferred copy number gain or loss. To reduce background noise in the visualization, the resultant HMM calls were thresholded for the number of gene-level siCNV events present in at least 35% of all spots across the entire dataset and in at least 45% of the spots of a given section. These thresholds were selected after detailed interrogation of thresholds ranging from 10–90% in 5% increments with positive-control, neutral and negative-control sections for visual consistency.

#### Clone selection

The dendrogram tree with numerical node identities was visualized, nodes were extracted and the specific barcodes (Visium spots) were digitally selected and assigned a clone identity. All members of a given analysis were merged, and a .csv file containing the clone identity and barcode was output for each Visium section.

#### Clone visualization

Loupe Browser version 5.0 (10x Genomics) was used to spatially visualize resultant clones from clone selection. For the manuscript, if a clone was present in <10 spatial spots (from 1k arrays or Visium) in a given section, it was not visualized.

#### Manual algorithmic tree building from preprocessed inferCNV data

**Clone tree consensus siCNV event calling.** Both HMM siCNVs (from files infercnv.17_HMM_predHMMi6.hmm_mode-samples.png and 17_HMM_predHMMi6.hmm_mode-samples.pred_cnv_regions.dat) and manual interpretation of denoised outputs (from file infercnv.21_denoised.png) were used to identify putative subclonal CNVs. These were then merged in a final consensus set, in which events were listed for each clone for building clone trees (Supplementary Tables 1 and 2). In brief, trees were constructed by identifying where CNVs were shared across the clusters identified above as, under the assumption that a CNV cannot be reversed once it occurs, this indicates that the cells in these clusters share a common ancestry. We therefore used this logic to identify ancestral relationships among clusters and build the clone tree. As our clone trees identify clones as related groups of cells (as opposed to clones being simply related mutations, an approach commonly taken in bulk-sequenced studies), where clones were present in subtrees that were not spatially proximate, we marked this uncertainty with dotted lines between common ancestors.

**Clone trees: branch lengths.** To semiquantitatively depict the 'evolutionary distance' between subclones, we determined the branch lengths by taking the logarithm (base 2) of the number of additional CNVs in the descendant clone and adding an arbitrary value to ensure that branches were always visible even with few CNV differences. The formula is given as $b_k = 100\log_2(|Z_{descendent}| - |Z_{parent}|) + 300$, where $b_k$ is the length of branch $k$ in pixels.

**Clone trees: clone diameters.** We scaled the size of each circle denoting a clone by the proportion of spots in the sample that was assigned to a clone using the formula $d_l = 10\log_2(s_l)$, where $d_l$ denotes the circle diameter in pixels and $s_l$ is the number of spots that correspond to a clone.

#### Maximum-parsimony clone trees

To validate our manual clone trees, we additionally computed maximum-parsimony clone trees following the instructions provided at https://cran.r-project.org/web/packages/phangorn/vignettes/Ancestral.html#parsimony-reconstructions (Extended Data Fig. 14). We used gene-level HMM copy number inferences (from file 17_HMM_predHMMi6.hmm_mode-samples.pred_cnv_genes.dat) as a 'user-defined input' matrix to the R package phangorn. All genes were included; if a clone did not have an inferred CNV event predicted, the matrix information for the gene in that clone was set to diploid.

#### siCNV parameters (Fig. 4)

Patient-matched scRNA-seq data from dissociated normal skin cells were analysed for selection (previously described) of a benign reference set. This reference set was then used as a reference control for all spatial transcriptomics spots in section T28. Node selection was performed (previously described). One pathologist (R.C.) annotated the resultant clones with the percentage of spots for each clone that harboured stroma, tumour epithelia or non-invasive epithelia (Supplementary Table 6). For siCNV analysis of the childhood brain tumour, patients 2 and 3 were selected as reference samples for patient 1. The selected reference samples appeared to have few to no inferred CNV gains and losses, as shown in Extended Data Fig. 13.

#### RNA versus DNA phylogeny analysis of previously published single-cell data

DNA and RNA data, co-extracted from single tumour cells, were obtained from publicly available data repositories[8]. Genomic and transcriptomic libraries were aligned to GRCh38.79. DNA-based CNV profiles were analysed and clustered with GINKGO (https://github.com/robertaboukhalil/ginkgo)[42]. RNA profiles were analysed with inferCNV[7], without a reference set, using default parameters. Tanglegrams of hierarchical clustering of both DNA-based copy number profiles and RNA-based inferred copy number profiles were then analysed with the R package Dendextend[43].

#### RNA versus DNA phylogenies from published prostate data

RNA data were obtained for patient A21 (refs. [9,44]), patient 499 (ref. [10]) and cases 6, 7 and 8 (ref. [11]). For patients A21 and 499, only a subset of all specimens had transcript data available. For cases 6, 7 and 8, only RNA microarray data were available, precluding their analysis by inferCNV[7]. The transcriptomes were aligned to GRCh38.79, and RNA counts were obtained. These were then processed into inferCNV[7] objects and run with standard inferCNV settings, without a reference set. Dendrograms from the inferCNV[7] outputs were visualized using R.

#### Synthetic data: generative process

To evaluate our application of the computational method inferCNV[7] to spatial transcriptomics data, we designed a generative process that resulted in an in silico spatial transcriptomics experiment of a tissue with a known—and spatially structured—clonotype population. In short, we constructed a spatial domain (representing a tissue region) in which we placed a set of virtual cells with a common genome structure and then let these cells populate the tissue region by simulating growth. In the process, at every time point cells can move, generate offspring, die or stay stagnant. The generative process above is implemented in Python code and available as a CLI application that can be accessed at GitHub (https://github.com/almaan/growmeatissue). The GitHub repository also contains more extensive documentation and examples of how to use the code; the exact parameters (defined in a TOML design file) used to produce the data presented here are included in Supplementary Data 1.

#### Synthetic data: evaluation

The process described above was used to generate a set of synthetic data incorporating a single chromosome, from which the obtained spatial gene expression data together with associated annotations were entered as input to siCNV (these data can be found in the Mendeley repository). The synthetic data were analysed according to the same procedure as previously outlined for the real data, providing as output information regarding the clonal population as determined from the inferred genomic state. To compare the results with the ground truth, we focused exclusively on the set of cells not being used as a reference (non-benign). InferCNV[7] assigns a state (either 3 or 6 depending on which HMM approach is used) to every gene in each clone; we

converted these states into a categorization that was more suitable for comparison according to the following scheme, given as 'spatial inferCNV state': new category: 1, deletion; 2, deletion; 3, neutral; 4, amplification; 5, amplification; 6, amplification. For the ground-truth data, we computed the average copy number of all cells assigned to each spot and rounded this value to the nearest integer. We considered a gene (within a clone) as deleted if the rounded average copy number within the given clone was less than 1, amplified if it was higher than 1 and neutral if it was equal to 1. Having cast the two datasets (real and synthetic) into comparable formats, we then computed the accuracy (within each clone) as the number of equal gene annotations (deletion, neutral, amplification) between the ground truth and the inferred results (from siCNV analysis).

### SNV analysis of Visium spatial transcriptomics data

To call SNVs from the data, we ran the cb_sniffer pipeline (https://github.com/sridnona/cb_sniffer) as published in ref. [22]. We identified all variants from 1000 Genomes[45] within any gene with an inferCNV[7] HMM-predicted alteration (5.4 million variants from 3,324 genes) in clones from patient 1, section H2_1 (Fig. 3). This output a total of 13,447,918 reads mapping to SNV loci, which corresponded to 573,781 unique candidate SNV loci detected in any spot. Of these, 51,945 SNVs had at least one read in one clone spot for each clone. We calculated clonal variant allele fractions for each variant within each clone by assessing the ratio of reference to alternative allele reads detected within spots assigned to a specific clone. Spot percentage was determined by calculating the total number of spots within a given clone that had a detected read that covered a candidate SNV locus divided by the total number of spots assigned to the given clone.

### Differential gene expression analysis

To analyse differentially expressed genes, we used the Seurat R package (version 4.0.5) and imported Space Ranger output files, after which the data were normalized and scaled using the default Seurat NormalizeData() and ScaleData() functions. Differential gene expression analyses were performed comparing groups using the FindMarkers() function with the following parameter: test.use = wilcox. For gene set enrichment analysis (GSEA), the msigdbr R package (version 7.4.1) was used to download the hallmark gene set from the Molecular Signatures Database. Genes that remained following filtering according to quality-control threshold criteria[46] ($\log_2$(fold change) ≥ 0.25, group percent threshold ≥ 0.1 and adjusted $P$ value ≤ 0.01) were passed through for GSEA. The plotEnrichment() function from the fgsea R package (version 1.16.0) was used to create GSEA enrichment plots.

### Statistics and reproducibility

All differential expression analysis was performed using gene markers found by two-sided Wilcoxon rank-sum test used by default in the Seurat FindAllMarkers function.

All spatial transcriptomics experiments, including histology, of prostate samples were performed in technical replicates of two and a biological replicate in the form of an additional whole prostate. All samples and analyses confirmed the original findings. In addition, technical repeats of data analysis (siCNV) were also re-run to confirm analysis results. Single-molecule FISH and spatial transcriptomics experiments on other tissues were not repeated.

### Reporting summary

Further information on research design is available in the Nature Research Reporting Summary linked to this article.

### Data availability

Sequence data for the prostate samples have been deposited at the European Genome-phenome Archive (EGA; www.ebi.ac.uk/ega/), which is hosted by the European Bioinformatics Institute (EBI), under accession number EGAS00001006124. The data are available under Data Use Conditions (DUO) and are limited to not-for-profit use as well as health/medical/biomedical purposes. Access is granted if the above criteria are fulfilled and local institutional review board/ethical review board approvals are provided. Raw FASTQ files for the childhood brain tumour samples are available through a materials transfer agreement with M. Nistér (monica.nister@ki.se), in line with GDPR regulations. Count matrices, high-resolution histological images and additional material are available on Mendeley Data (https://doi.org/10.17632/svw96g68dv.1). Public data used for comparison of phylograms were obtained from the European Nucleotide Archive (http://www.ebi.ac.uk/ena), under accession numbers ERP022266 (RNA-seq) and ERP022267 (WGS), as well as from the EGA, under accession numbers EGAS00001001659 and EGAS00001000942. Public patient-specific benign cutaneous scRNA-seq data were obtained from the Gene Expression Omnibus (GSE144236). Public spatial transcriptomics data used in the study were all obtained from 10x Genomics. Human lymph node (https://www.10xgenomics.com/resources/datasets/human-lymph-node-1-standard-1-1-0), breast cancer (https://www.10xgenomics.com/resources/datasets/human-breast-cancer-block-a-section-1-1-standard-1-1-0) and glioblastoma (https://www.10xgenomics.com/resources/datasets/human-glioblastoma-whole-transcriptome-analysis-1-standard-1-2-0) data are all available as dataset resources.

### Code availability

Details of the spatial transcriptomics analysis pipeline can be found at https://github.com/jfnavarro/st_pipeline. The factor analysis software (STD) is available under GNU General Public License v3 at https://github.com/SpatialTranscriptomicsResearch/std-nb. The SpatialInferCNV package along with documentation is available at https://github.com/aerickso/SpatialInferCNV. An archived permanent repository of SpatialInferCNV is available using https://doi.org/10.6084/m9.figshare.19666317.v1. The code as well as documentation for generating synthetic data is available at https://github.com/almaan/growmeatissue.

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

**Acknowledgements** This project has received funding from the European Research Council (ERC) under the European Union's Horizon 2020 research and innovation programme (grant agreement no. 101021019). The study was also supported by the Swedish Cancer Society, Swedish Foundation for Strategic Research, AstraZeneca and Science for Life Laboratory. We also acknowledge the Swedish Childhood Tumour Biobank, supported by the Swedish Childhood Cancer Fund, for access and handling of patient biobank material/sequencing data and the Swedish Childhood Cancer Fund. We would like to thank the National Genomics Infrastructure, Sweden, for providing infrastructure support. We thank A. Mollbrink, X.M. Abalo, M. Nistér and P. Lundin for helpful assistance and discussions. A.D.L. was supported by a Cancer Research UK Clinician Scientist Fellowship award (C57899/A25812) that also funded A.E. F.C., R.C. and A.D.L. have received support from the Oxford National Institute for Health Research (NIHR) Biomedical Research Centre Surgical Innovation and Evaluation Theme. I.G.M. is grateful to the John Black Charitable Foundation for support, and D.J.W. is grateful to the Cancer Research UK Oxford Centre. Computation used the Oxford Biomedical Research Computing facility, a joint development between the Wellcome Centre for Human Genetics and the Big Data Institute supported by Health Data Research UK and the NIHR Oxford Biomedical Research Centre. The views expressed are those of the author(s) and not necessarily those of the NHS, the NIHR or the Department of Health. We appreciate the kind assistance of A. Ståhl, R.A. Novoa and K. Rieger with additional histology assessment of specimens in this study. We appreciate the kind assistance of S. Tuomisto and W. Yin for their technical assistance and feedback on code development. We also thank S. Figiel for assistance with spatial transcriptomics technical developments and feedback on data analysis and visualization.

**Author contributions** E.Berglund, M.H., M.M., R.M., L.A.G. and N.S. performed the experiments; A.E., L.B., L.K., J.B., L.L., D.J.W., T.R., K.T., A.L.J., M.M. and T.D.D.S. analysed the data; L.B. and J.M. developed the factorization approach; A.A. and A.E. designed and implemented the method for synthetic data generation; F.T. and T.H. provided tissue; R.C., T.M., A.T. and A.S. undertook pathology analysis; E.Basmici, D.D., A.L.J., P.A.K., F.T., I.G.M., F.C.H. and D.J.W. aided interpretation of study data; E.Berglund, A.E., M.H., D.J.W., A.D.L. and J.L. drafted the manuscript; all authors read and approved the final manuscript. A.D.L. and J.L. supervised and managed the study. M.M., R.M. and N.S. contributed equally to this work.

**Funding** Open access funding provided by Royal Institute of Technology.

**Competing interests** M.H., M.M., R.M., L.K., A.A., L.L., L.A.G., K.T. and J.L. are scientific consultants to 10x Genomics, Inc.

**Additional information**
**Correspondence and requests for materials** should be addressed to Alastair D. Lamb or Joakim Lundeberg.

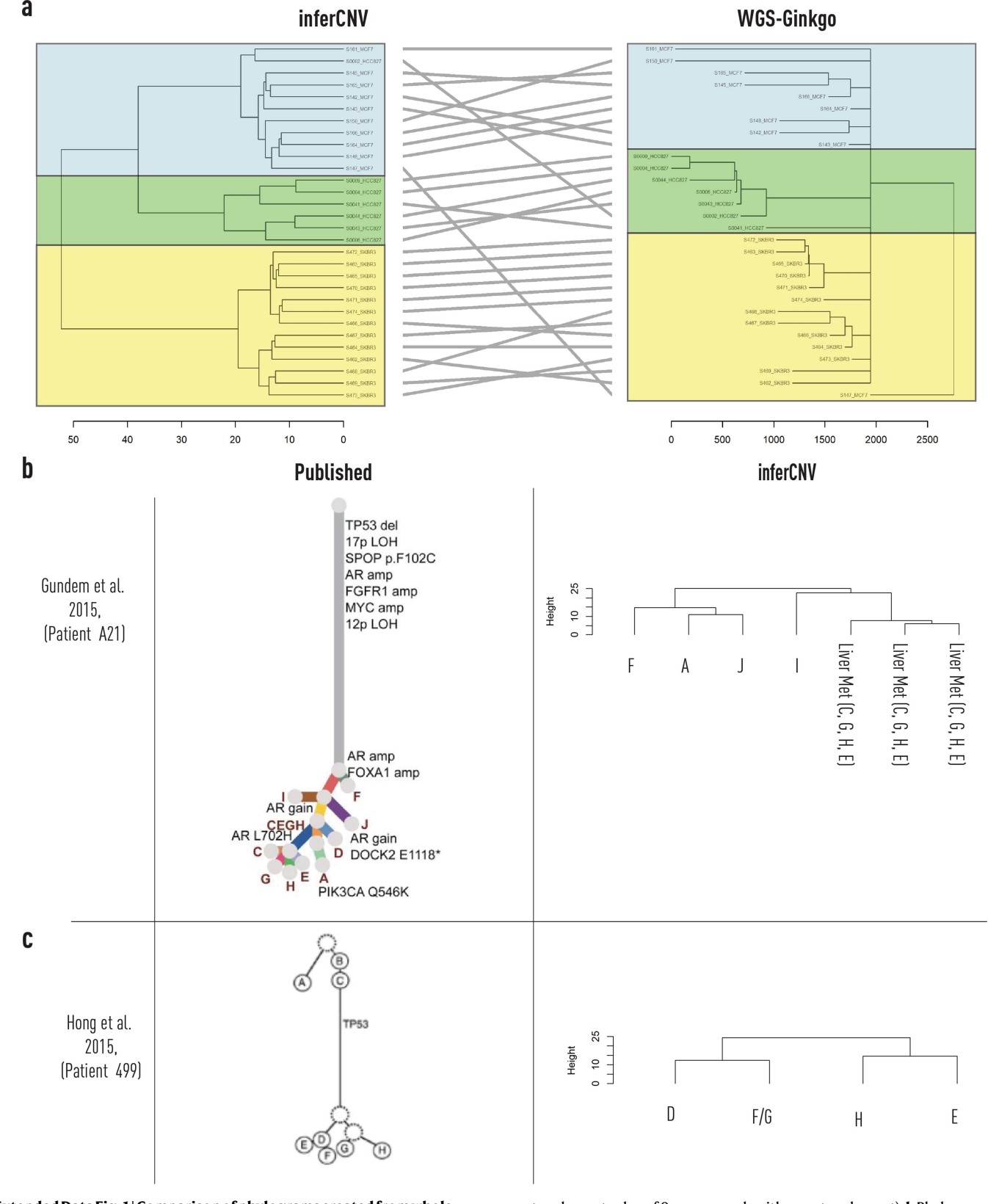

**Extended Data Fig. 1 | Comparison of phylograms created from whole genome sequencing CNVs and iCNV's. a**, Comparison of single tumour cells with co-isolated DNA and RNA (Han et al., Genome Res 2018). Colours correspond to individual cell lines (yellow: SKBR3, green: HCC827, and light blue: MCF7). Entanglement of the phylograms was 0.11 (an entanglement value of 1 corresponds with full entanglement of two phylograms, whereas an entanglement value of 0 corresponds with no entanglement). **b** Phylogeny from patient A21, as published and reproduced from Gundem et al., Nature, 2015. Transcript data were available only for a subset of specimens. **c**, Phylogeny from patient 499, as published and reproduced from Hong et al., Nat. Comms, 2015. Transcript data available for a subset of specimens (used to reproduce phylogenetic tree by inferCNV).

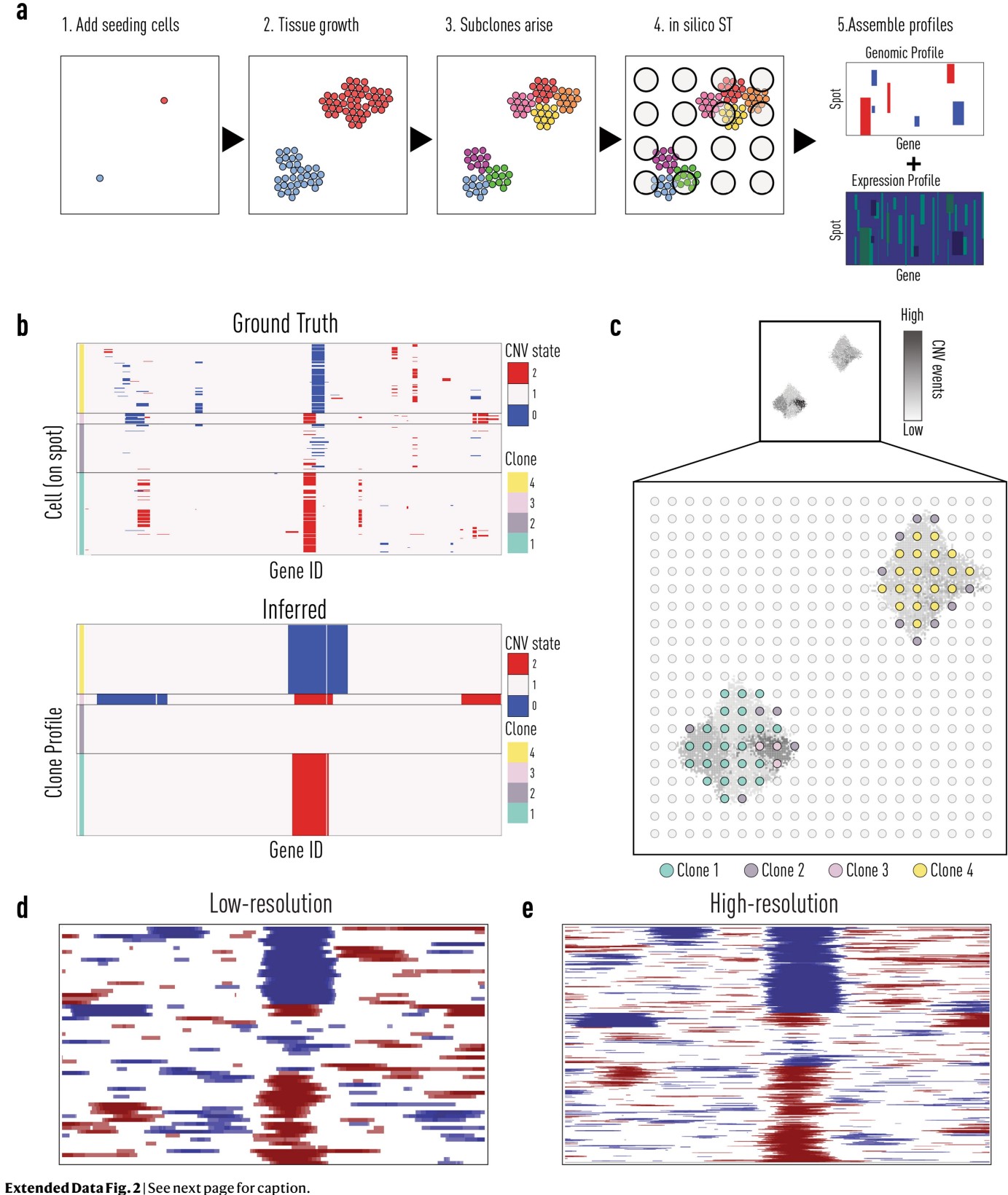

**Extended Data Fig. 2** | See next page for caption.

**Extended Data Fig. 2 | Generating and running inferCNV on synthetic data.**
**a**, Schematic overview of the generative process used to produce artificial spatial data. 1) First a set of seeding cells (red and blue circles) are placed in a defined tissue domain (square), every seeding cell hosts one unique copy number event. 2) The cells are allowed to "grow" within the tissue domain until the number of cells in the domain exceeds a predetermined number. 3) Mutations in the genome occur stochastically during growth and as a result, subpopulations (indicated by colour) of cells with similar genomic profiles arise. 4) Unoccupied space in the tissue domain is filled with benign cells (no copy number variations), spatial capture locations are placed in a grid over the grown tissue and transcripts are "captured" from the cells overlying each spot. 5) Synthetic spatial expression data is produced together with associated ground truth genomic data (both on spot and cell level). **b**, Results from applying siCNV (bottom) to a set of synthetic data together with ground truth information (top), only cells residing at spots being annotated as non-benign are shown. Blue indicates a deletion event while red indicates an amplification event. The ground truth shows the genomic profiles for all cells contributing to the spots assigned to a given clone. Comparing the inferred state with the ground truth on a clone 19 level, the average accuracy across genes was 0.90 (standard deviation 0.10) **c**, Spatial organization of the synthetic data analysed in (b), with thumbnail of the complete cell population in the artificial tissue, each pixel corresponding to a cell. The cells' intensity levels are proportional to their total number of associated copy number events. Circles represent the spots used to "capture" transcripts. Spots are coloured by their inferred clone identity. Note how Clone 2, predicted to have zero copy number events, is found along the borders of both foci, where there's a mixture of benign and non-benign cells. **d**, siCNV outputs from simulated synthetic data of spots simulating ST 1k array (low-resolution) with 100 µm spot diameter and centre-to-centre distance of 200 µm. **e**, Visium (high-resolution). High resolution spots were 0.55x size of low resolution and had 5x more spots per area. The synthetic ground truth data were identical for both.

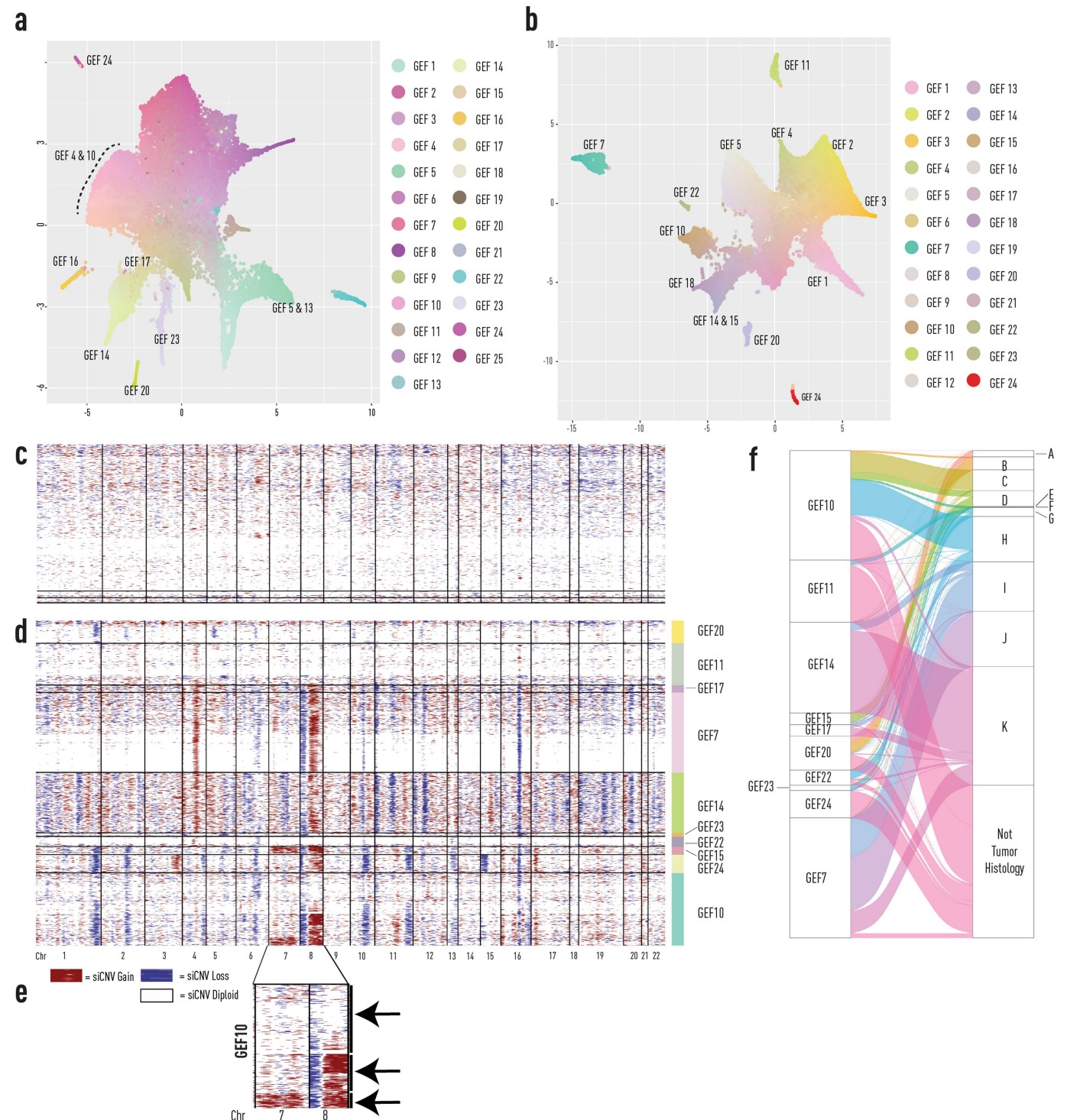

**Extended Data Fig. 3 | GEF directed inferCNV analysis from prostate patient 1 (high resolution Visium analysis). a**, UMAP summary of GEFs from 1k spatial transcriptomics experiments of prostate samples from patient 1. **b**, UMAP summary of GEFs from high resolution Visium experiments of prostate samples from patient 1. Top marker genes for each GEF are available in Supplementary Table 3, 4. **c**, Benign GEFs from b (high resolution) were used as a reference set for analysis of **d**, Tumour GEFS from b (high resolution). **e**, Snapshot of inferCNV profiles for chr 7 and 8 from GEF10. GEF inferCNV heterogeneity is highlighted by 3 subclones: the first harbouring no changes to chr 7 and 8, the second having a deletion and amplification in chr 8, and the last having alterations in both chr 7 and chr 8. While further subclustering of GEF10 spots using gene expression factors improved GEF to clone concordance, GEF to clone heterogeneity remained. **f**, Tumor GEFs distribution by siCNV clones (Fig. 2). GEF = Gene Expression Factor, chr = Chromosome, siCNV = spatial inferCNV.

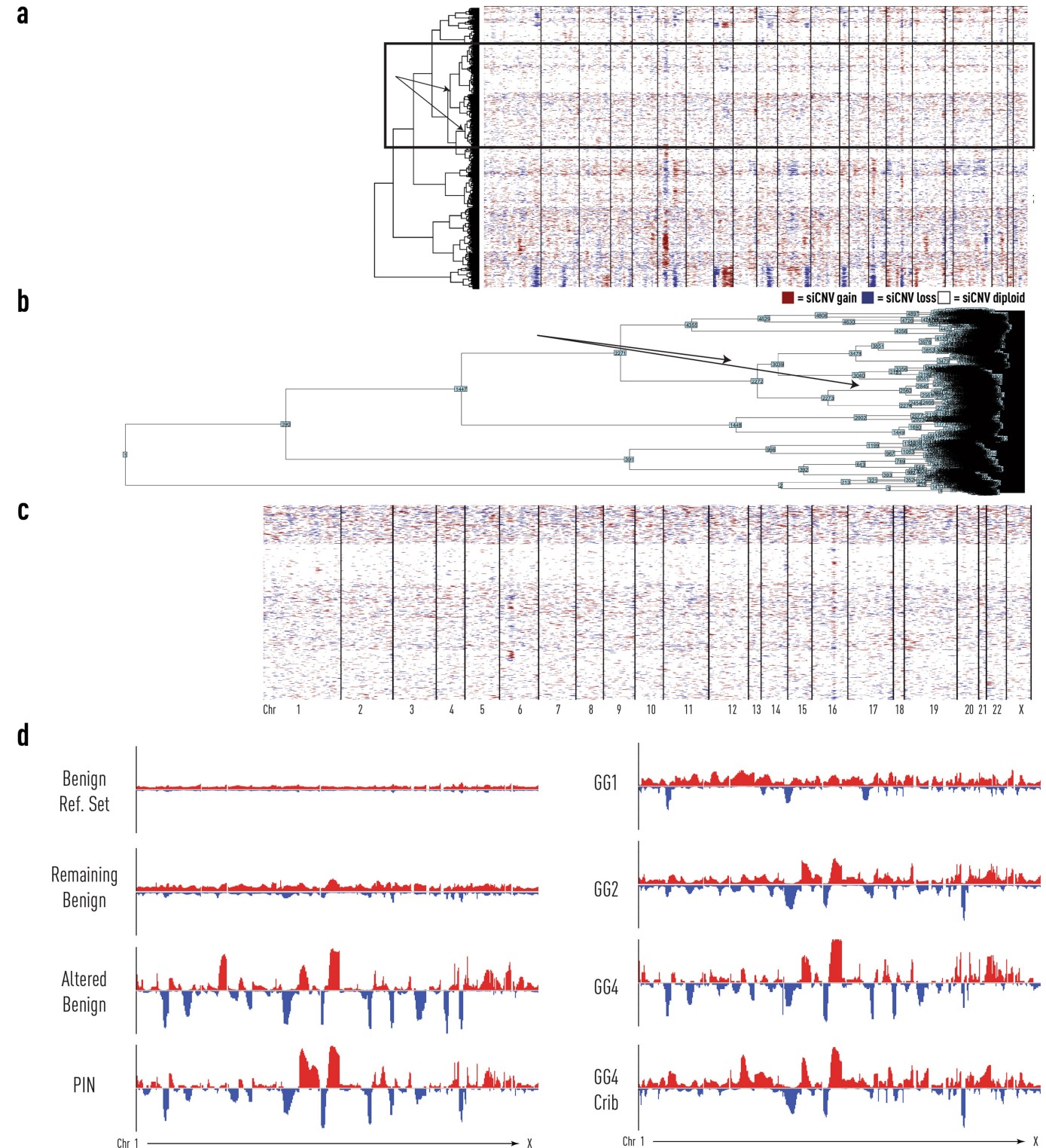

**a**

= siCNV gain = siCNV loss = siCNV diploid

**b**

**c**

Chr 1 2 3 4 5 6 7 8 9 10 11 12 13 14 15 16 17 18 19 20 21 22 X

**d**

Benign Ref. Set

Remaining Benign

Altered Benign

PIN

GG1

GG2

GG4

GG4 Crib

Chr 1 → X

Chr 1 → X

**Extended Data Fig. 4 | Identification of a histologically benign reference set from prostate patient 1. a**, Visual selection of benign epithelial spots harbouring the least amount of inferred copy number variations, as outlined by the black box bounding box. Arrows identify dendrogram nodes corresponding to barcoded spots within the box. **b**, InferCNV output of the dendrogram nodes with numerical identifiers for selection corresponding to Panel a. **c**, Finalized benign reference set from analysis of epithelial cells in prostate patient 1, section H2_1 (Fig. 3). **d**, Global spatial inferCNV profiles of the selected benign reference set from panel a, the remainder of the benign not included in the reference set, altered benign (Clone C, Fig. 3), and the other Visium spots with luminal epithelial annotations (PIN, GG1, GG2, GG4, GG4 Cribriform).

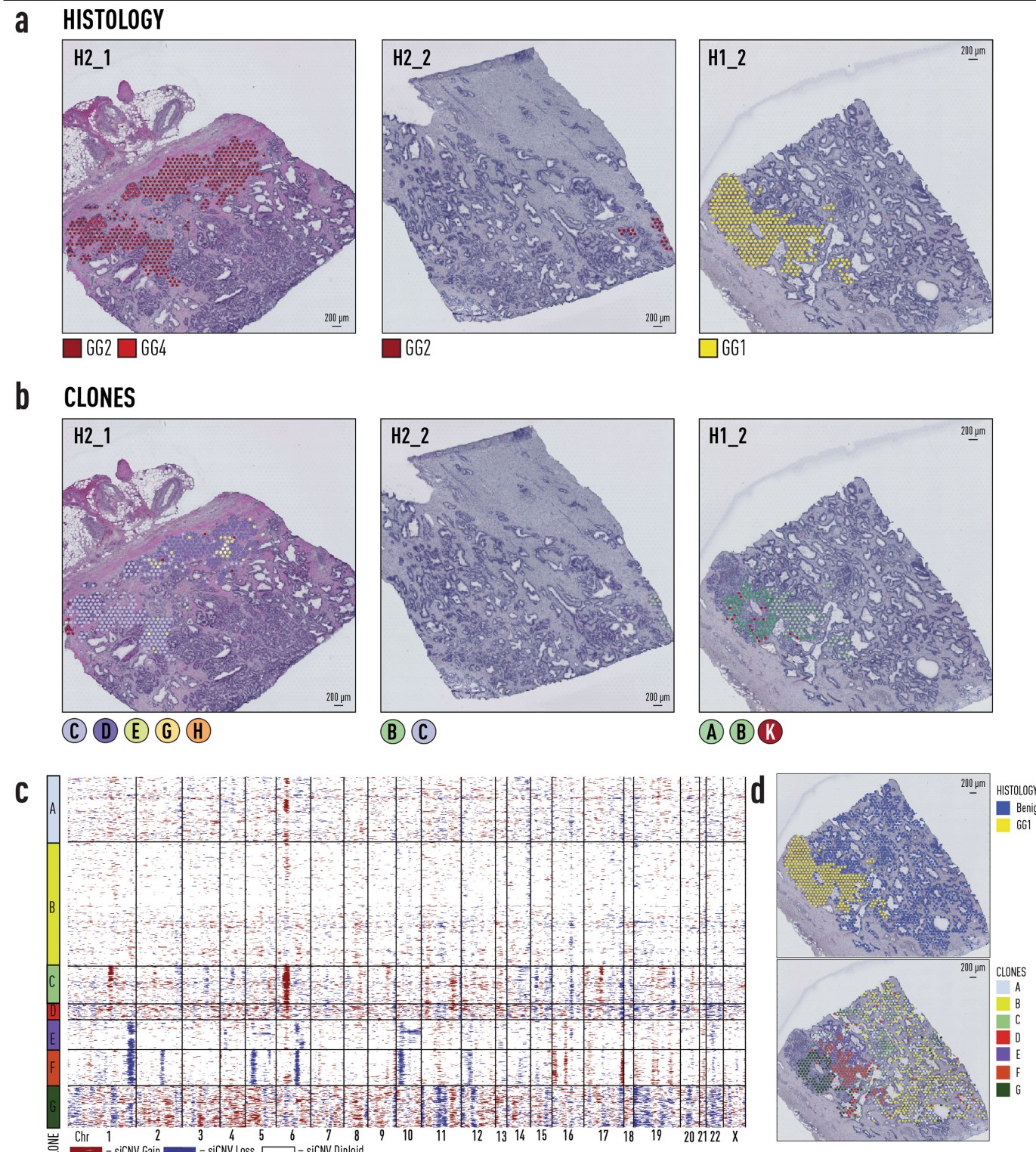

**Extended Data Fig. 5 | Histology and clones (from Fig. 2a) for prostate patient 1. a**, Consensus pathology annotations for tumour spots from sections H2_1, H2_2, and H1_2. **b**, Clonal groupings of spots (approx. 10-15 cells each) determined by hierarchical clustering. **c**, Distinct siCNV profile of GG1 tumour focus from organscale prostate patient 1. siCNV profiling of epithelial Visium spots from section H1_2. **d**, Spot level histology and siCNV clone calls. GG = ISUP Gleason 'Grade Group', siCNV = spatial inferCNV.

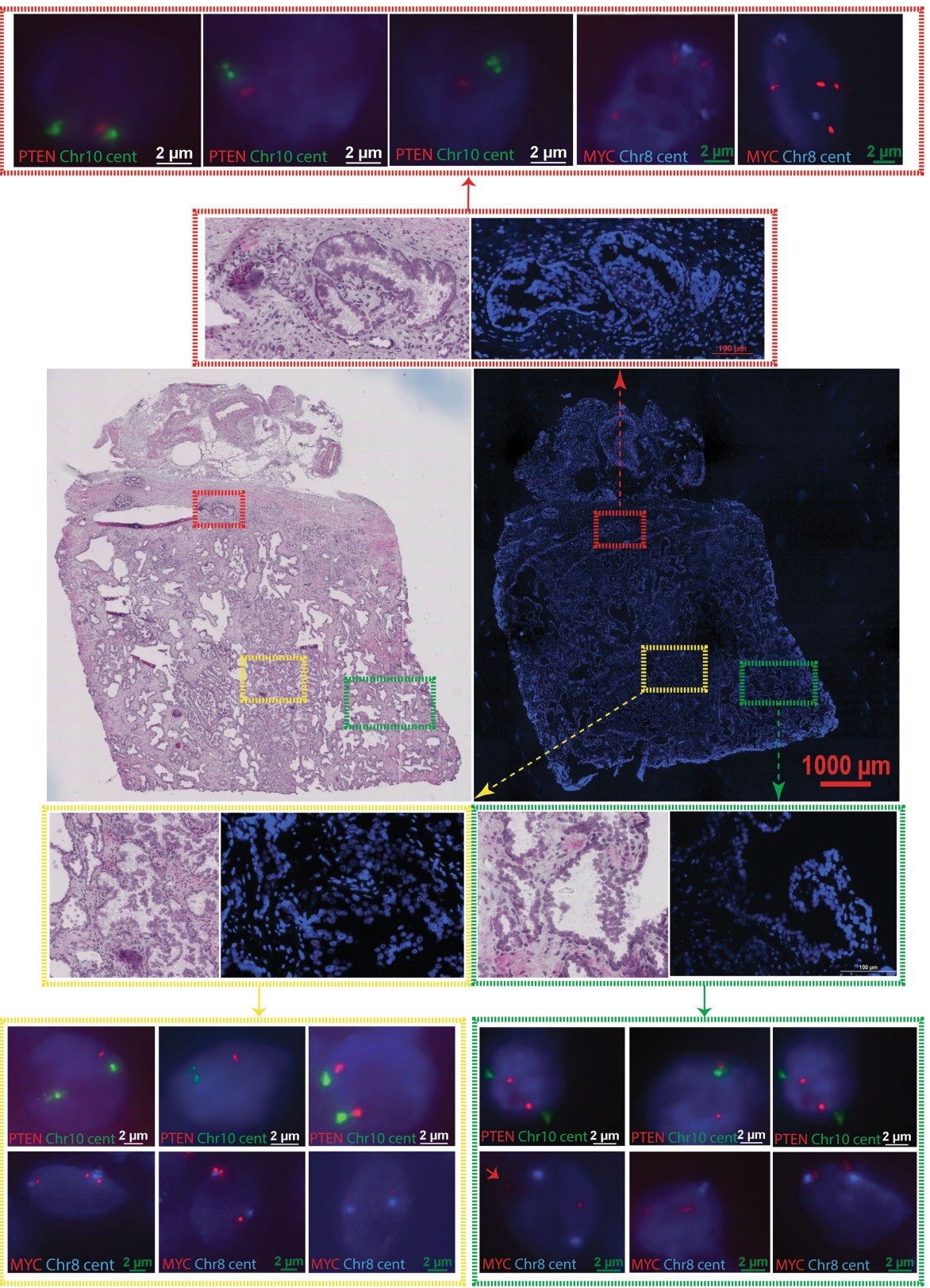

**Extended Data Fig. 6 | DNA FISH targeting MYC and PTEN loci.**
Representative images from fresh frozen prostate tissue sections obtained from patient H2.1 labelled with Cytocell MYC/8cent and PTEN/10cen probes. Three consecutive sections were used for H&E staining and FISH. Control probes labelled chromosome 8 and 10 centromeres in (**green** & **aqua**) respectively, and MYC and PTEN shown in (**red**). Nuclei counterstained with DAPI (**dark blue**).

**a**

| Gene EEF1D, chr8:143580183 | | | | | | |
|---|---|---|---|---|---|---|
| Clone | Ref_count | Alt_count | Total_CB | ClonalVAF | SpotPercentage | siCNV |
| A | 463 | 501 | 964 | 0.52 | 90.2 | Diploid |
| B | 201 | 115 | 316 | 0.36 | 94.6 | AMP |
| C | 1668 | 931 | 2599 | 0.36 | 99.1 | AMP |
| D | 210 | 97 | 307 | 0.32 | 94.0 | AMP |
| E | 1217 | 722 | 1939 | 0.37 | 99.6 | AMP |
| F | 1149 | 426 | 1575 | 0.27 | 100.0 | AMP |
| G | 1066 | 462 | 1528 | 0.3 | 97.4 | AMP |

**b**

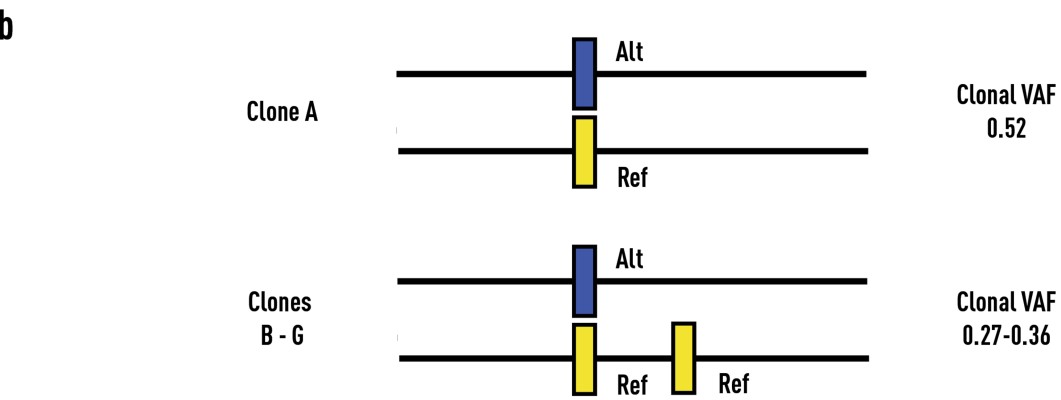

**c**

| Gene COX6C, chr8:99892049 | | | | | | |
|---|---|---|---|---|---|---|
| Clone | Ref_count | Alt_count | Total_CB | ClonalVAF | SpotPercentage | siCNV |
| A | 234 | 270 | 504 | 0.54 | 79.56 | Diploid |
| B | 75 | 126 | 201 | 0.63 | 90.91 | AMP |
| C | 1009 | 1885 | 2894 | 0.65 | 100 | AMP |
| D | 73 | 125 | 198 | 0.63 | 76.12 | AMP |
| E | 612 | 1489 | 2101 | 0.71 | 96.46 | AMP |
| F | 313 | 988 | 1301 | 0.76 | 100 | AMP |
| G | 436 | 1138 | 1574 | 0.72 | 95.22 | AMP |

**d**

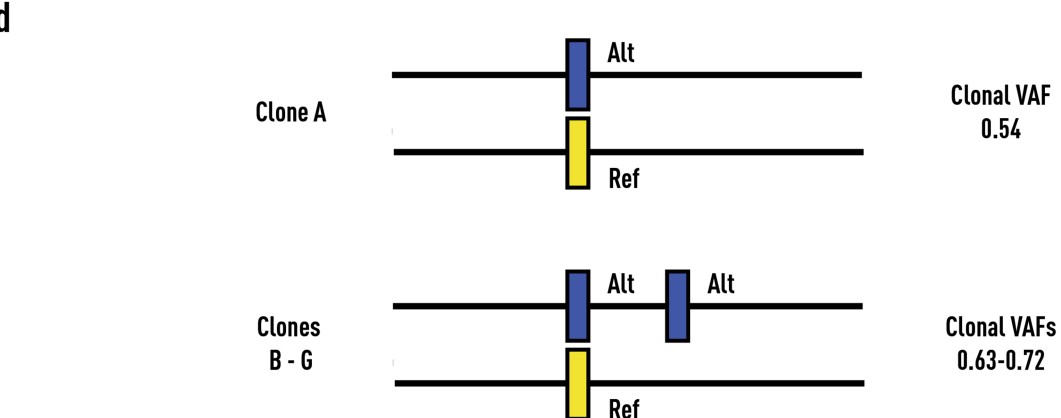

**Extended Data Fig. 7 | Single nucleotide variant analysis of spatial transcriptomic data from prostate patient 1, section H2_1. a**, Summary table of alt and reference read data from clones A-G (Fig. 3) of *EEF1D*. **b**, Cartoon diagram demonstrates how clone B-G, harbor copy number gain of the same allele as evidenced by the decreased variant allele fraction (VAF). **c**, Summary table of alt and reference read data from clones A-G (Fig. 3) of COX6C. **d**, Cartoon diagram demonstrates how clone B-G, harbor copy number gain of the same allele as evidenced by the increase VAF.

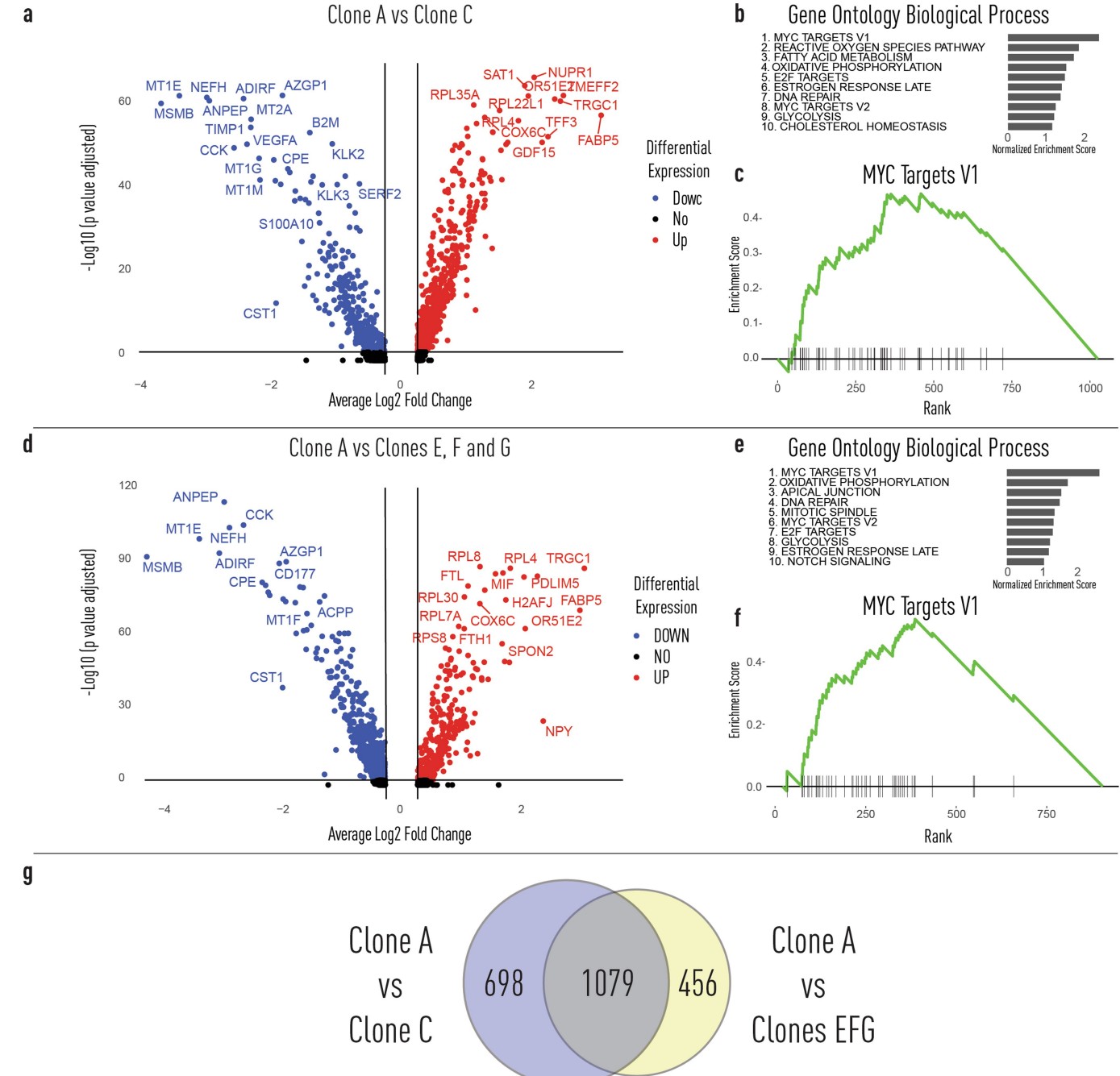

**Extended Data Fig. 8 | Differential gene expression analyses benign, altered benign and tumour clones. a**, Differentially expressed genes from benign clone A and altered clone C. Using a two-sided Wilcoxon Rank-Sum test. **b**, Top 10 pathways identified by geneset enrichment analyses (GSEA) from clone A vs clone C. **c**, Top ranked enrichment pathway from GSEA. **d**, Differentially expressed genes from benign clone A and tumour clones E, F and G. Using a two-sided Wilcoxon Rank-Sum test. **e**, Top 10 pathways identified by GSEA from clone A vs clones E, F and G. **f**, Top ranked enrichment pathway from GSEA. **g**, Venn-Diagram of genes from differential gene expression analyses identified only in benign clone A vs altered benign clone C analysis (left), benign clone A vs tumour clones E, F and G (right).

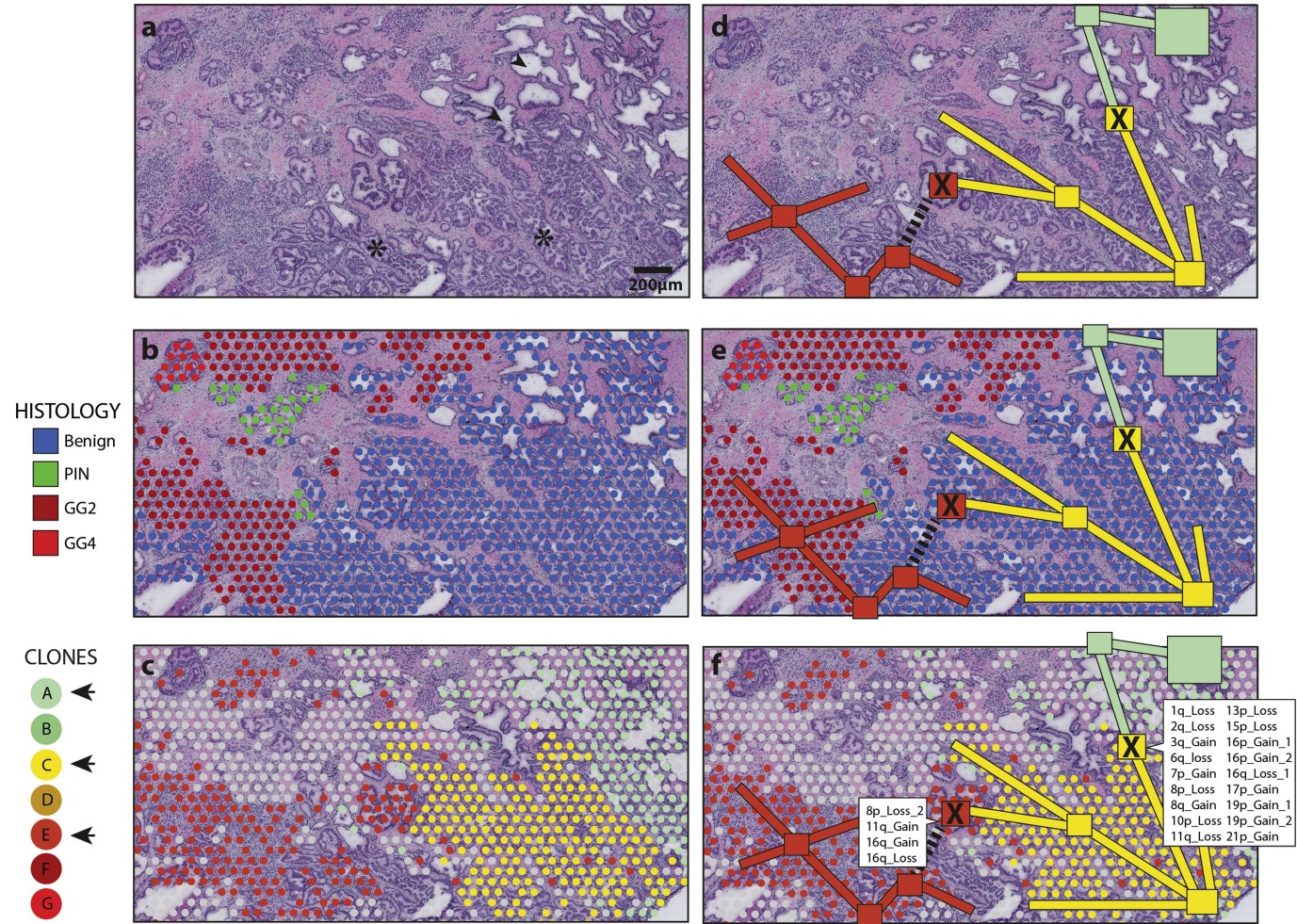

**HISTOLOGY**

- Benign
- PIN
- GG2
- GG4

**CLONES**

- A ←
- B
- C ←
- D
- E ←
- F
- G

1q_Loss 13p_Loss
2q_Loss 15p_Loss
3q_Gain 16p_Gain_1
6q_loss 16p_Gain_2
7p_Gain 16q_Loss_1
8p_Loss 17p_Gain
8q_Gain 19p_Gain_1
10p_Loss 19p_Gain_2
11q_Loss 21p_Gain

8p_Loss_2
11q_Gain
16q_Gain
16q_Loss

**Extended Data Fig. 9 | Branching morphogenesis and somatic mosaicism in prostate epithelium. a**, Close up histology of Section H2_1 demonstrating clear ductal (e.g. arrow heads) and acinar (e.g. stars) branching patterns. **b**, Overlayed spot-level histology. **c**, Overlayed clone groupings (from Fig. 3).

**d-f**, Possible arrangement of clonal expansion during branching morphogenesis with key mutational events (marked with X, siCNV events from Fig. 3) passed on to downstream branches. Dotted line represents presumed branch/duct not visible in two-dimensional plane.

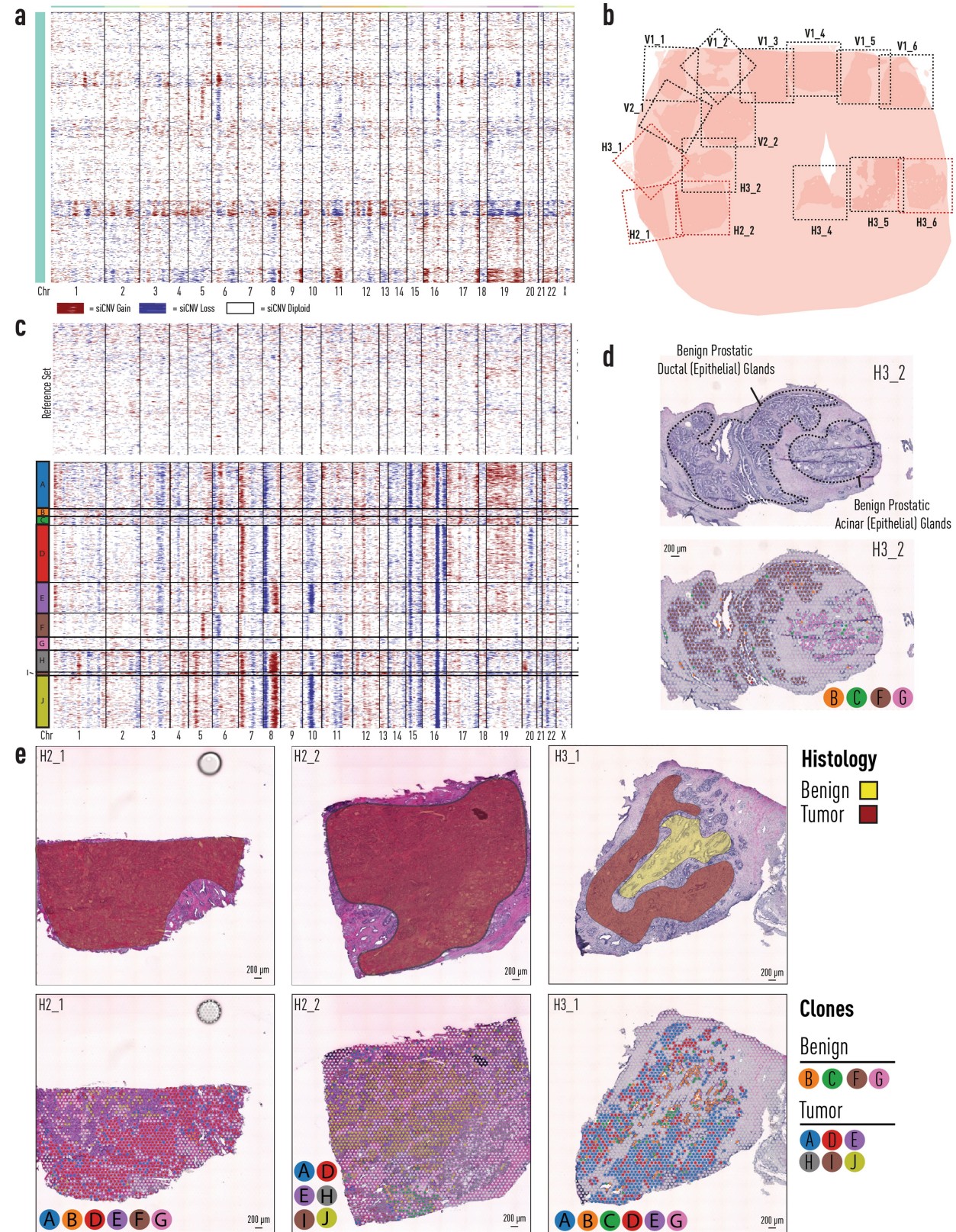

**Extended Data Fig. 10 | Organscale prostate patient 2. a**, Spatial inferCNV (siCNV) profiles of histologically benign prostatic epithelial cells from 11 sections from prostate patient 2. **b**, Reference overview of 15 sections available for analysis: sections H2_1, H2_2, H3_1, and H3_6 harbour tumour (marked with red dotted lines). Black dotted lines represent the area covered by spatial transcriptomics array surface. **c**, Analysis of tumour foci in sections H3_1, H2_1 and H2_2. Analysis includes section H3_2, a non-tumour bearing section which included spatially co-localized benign spots harbouring inferred CNV alterations from panel a. **d**, Spatial histology and clone distribution in section H3_2 (no-tumour). Benign ductal histology (Clone F) harbours distinct inferred CNVs (chr5 amplification, chr6 deletion), not harboured in neighbouring benign acinar glands (Clone G). **e**, Section histology (transparent red indicates tumour, and transparent yellow denotes benign) and clones from tumour-bearing sections H3_1, H2_1, and H2_2. CNV = Copy-Number Variant, chr = chromosome.

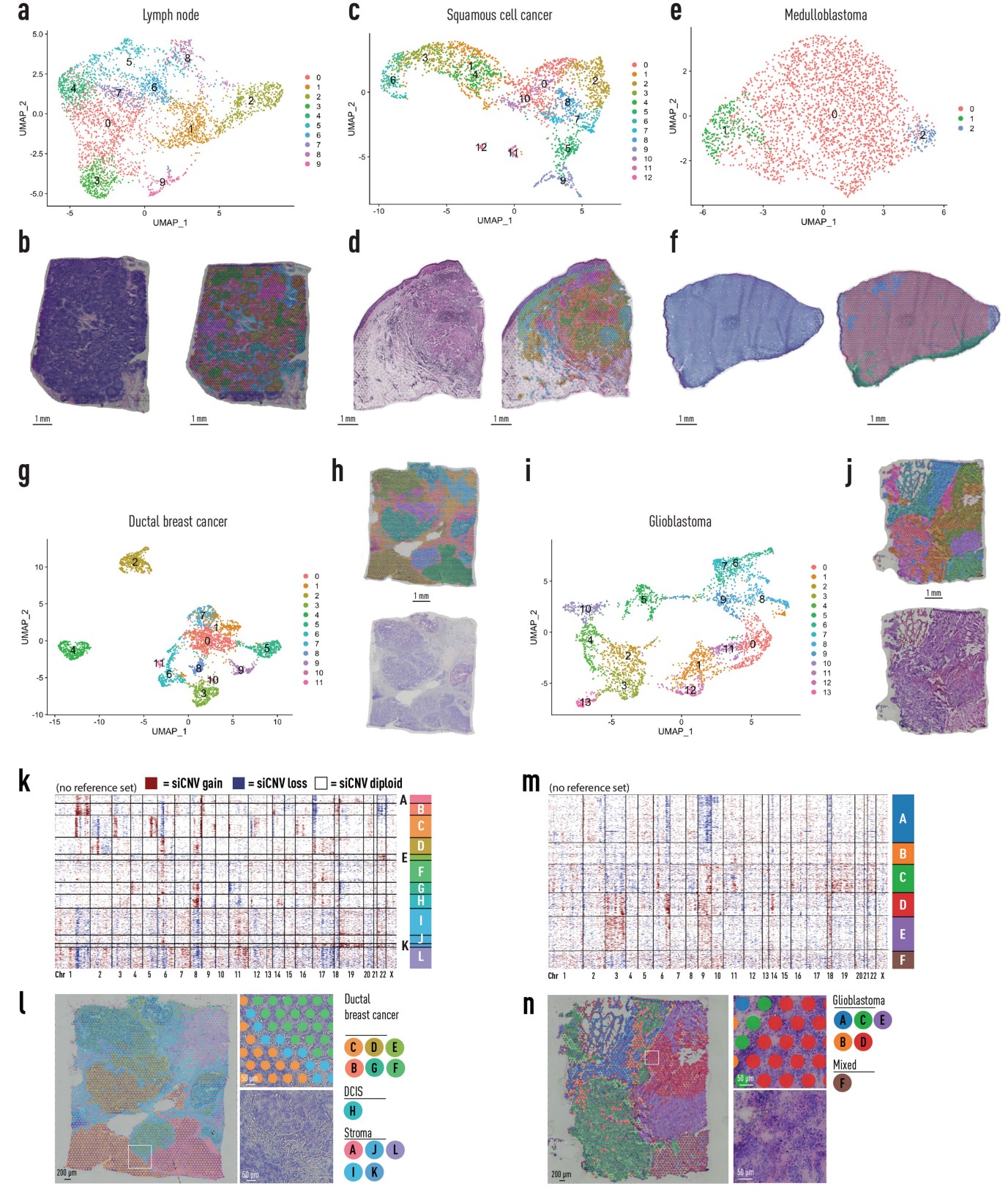

**Extended Data Fig. 11 | Spatial transcriptomics and siCNV analysis of multiple sample types.** a, c, e, g, i, Transcript UMAPs of all spots labelled by cluster from human lymph node (a), human squamous cell carcinoma (c), malignant childhood brain tumour diagnosed as medulloblastoma (e) human invasive ductal breast carcinoma (g), malignant childhood brain tumour diagnosed as medulloblastoma SHH grade IV (i). b, d, f, h, j, H&E stain and unbiased cluster spots visualized spatially on tissue from human lymph node (b), human squamous cell carcinoma (d), childhood medulloblastoma (f), human invasive ductal breast carcinoma (h), human glioblastoma multiforme (j). **k-n**, somatic copy number alterations in breast tissue containing ductal breast cancer and DCIS (k, l) and brain tissue containing glioblastoma (m, n). While some of the samples did not have an annotated benign reference set, interestingly, unsupervised siCNV could still segment different histological clones. However, the lack of a reference set did reduce the ability to identify specific inferred CNVs.

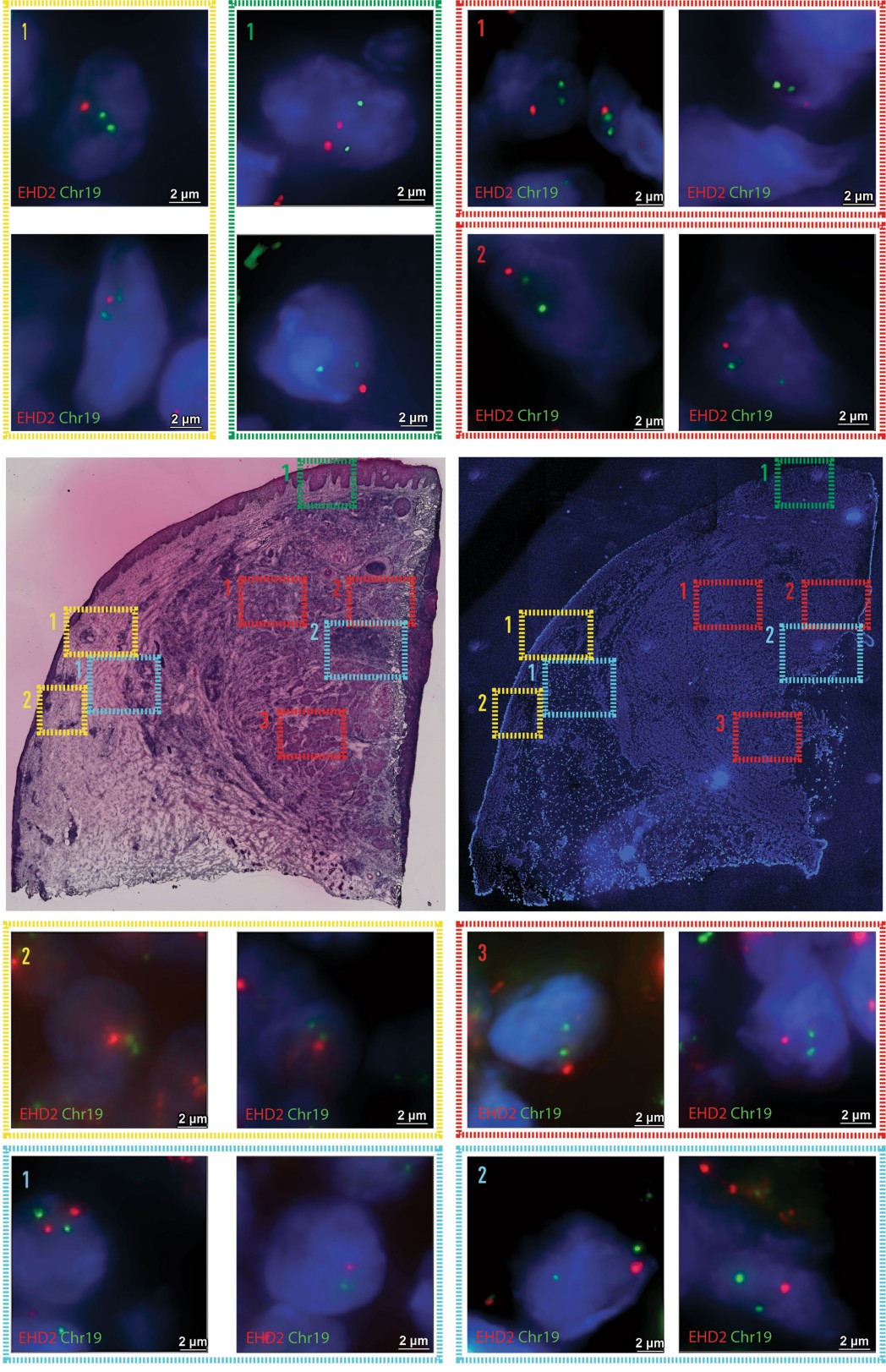

**Extended Data Fig. 12 | DNA FISH targeting EHD2 locus.** Representative images from fresh frozen squamous cell carcinoma tissue sections labelled with Cytocell 19q13/19p13 probe. Consecutive sections were used for H&E, DAPI staining and FISH. Control probe labelled the 19p13.2 region of chromosome 19 in **green**, and EHD2 is shown in **red**. Nuclei counterstained with DAPI (**dark blue**). Yellow dashed rectangles mark the clonal group B position 1 and 2.

Red dashed rectangles mark the clonal group A positions 1, 2, and 3. Blue dashed rectangles mark the clonal group D positions 1, and 2. Green rectangle marks the clonal group C position 1. Predicted deletions of EHD2 gene are shown in clones A, B, C, and D. Note that the clonal groups C, and D show deletions of EHD2 gene as well as diploid cells.

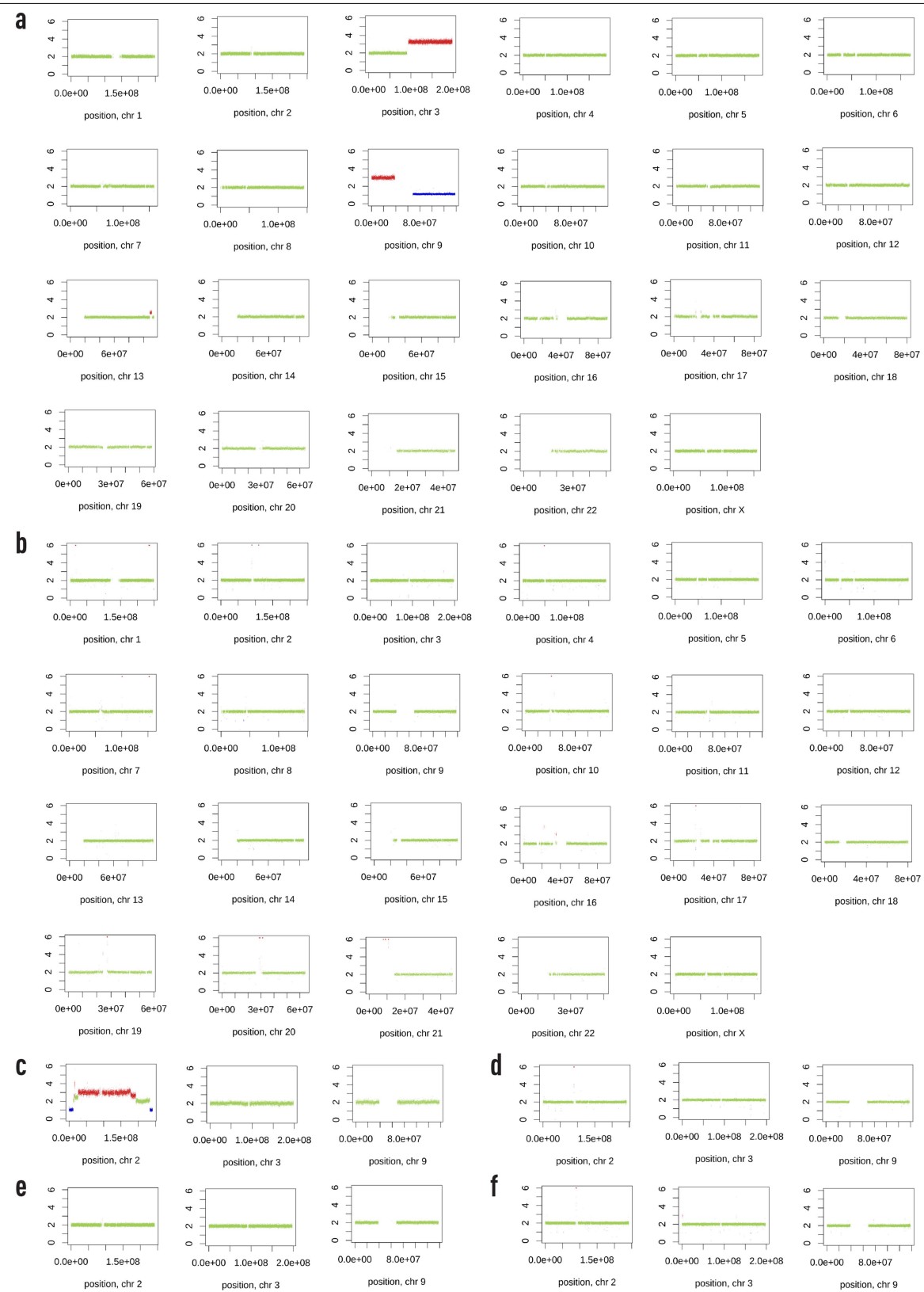

**Extended Data Fig. 13 | Whole-genome sequencing-based copy number profiles for paediatric brain tumour patients. a**, Somatic WGS CNV profile of patient 1 diagnosed with medulloblastoma (grade IV, desmoplastic/nodular, SHH-activated) with **b**, match normal blood. **c**, Somatic WGS CNV profile of Chr 2, 3 and 9 of patient 2 diagnosed with medulloblastoma (grade IV, classic morphology, SHH-activated) with **d**, match normal blood. Notably inferCNV analysis on Visium data did not show any genomic variability in chr 2 but since Visium and WGS data were generated from different locations of each tumour,

we speculate that the observed WGS CNV patterns in patient 2 could be due to the inherent spatial heterogeneity of DNA copy number alterations observed by others when sampling multiple sites of medulloblastoma tumours. **e**, Somatic WGS CNV profile of Chr 2, 3 and 9 of patient 3 diagnosed with CNS embryonal tumour (grade IV, multi-layered rosettes, NOS) with **d**, match normal blood. No CNV was detected by WGS in the chromosomes not displayed. WGS = Whole-genome sequencing. Chr = Chromosome. SHH = Sonic hedgehog. CNS = Central nervous system. NOS = Not otherwise specified.

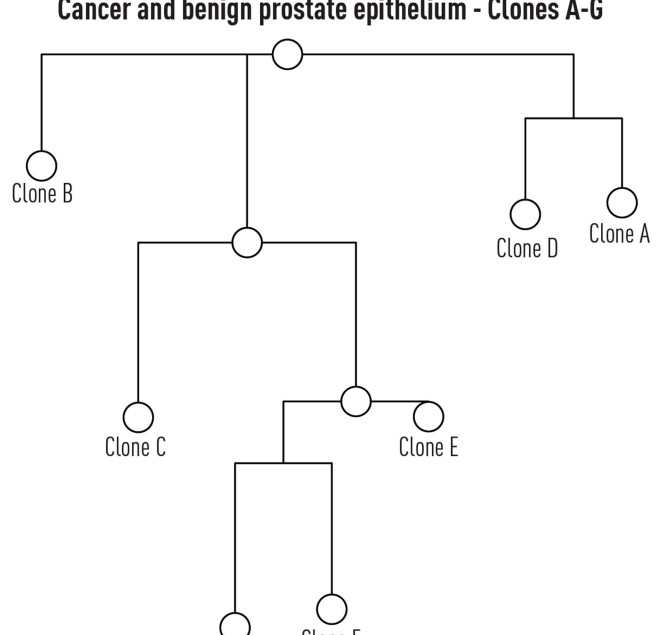

**a** All cancer organ-wide analysis - Clones F-K

Clone F

Clone G

Clone H

Clone I

Clone J

Clone K

**b** Cancer and benign prostate epithelium - Clones A-G

Clone B

Clone D

Clone A

Clone C

Clone E

Clone G

Clone F

**Extended Data Fig. 14 | Maximum parsimony reconstructions of prostate cancer clone trees. a**, Maximum parsimony tree for clones F-K from spatially proximate tumour bearing sections from sections H1_4, H1_5, and H2_5 from prostate cancer patient 1 (Fig. 2). **b**, Maximum parsimony tree for prostate cancer and benign epithelial clones A-G from sections H2_1 from prostate cancer patient 1 (Fig. 3). Input data to construct both trees were derived from gene-level siCNV hidden markov model data.

# Reporting Summary

## Statistics

For all statistical analyses, confirm that the following items are present in the figure legend, table legend, main text, or Methods section.

| n/a | Confirmed | |
|---|---|---|
| ☐ | ☒ | The exact sample size (*n*) for each experimental group/condition, given as a discrete number and unit of measurement |
| ☐ | ☒ | A statement on whether measurements were taken from distinct samples or whether the same sample was measured repeatedly |
| ☐ | ☒ | The statistical test(s) used AND whether they are one- or two-sided *Only common tests should be described solely by name; describe more complex techniques in the Methods section.* |
| ☐ | ☒ | A description of all covariates tested |
| ☐ | ☒ | A description of any assumptions or corrections, such as tests of normality and adjustment for multiple comparisons |
| ☐ | ☒ | A full description of the statistical parameters including central tendency (e.g. means) or other basic estimates (e.g. regression coefficient) AND variation (e.g. standard deviation) or associated estimates of uncertainty (e.g. confidence intervals) |
| ☐ | ☒ | For null hypothesis testing, the test statistic (e.g. $F$, $t$, $r$) with confidence intervals, effect sizes, degrees of freedom and $P$ value noted *Give P values as exact values whenever suitable.* |
| ☐ | ☒ | For Bayesian analysis, information on the choice of priors and Markov chain Monte Carlo settings |
| ☐ | ☒ | For hierarchical and complex designs, identification of the appropriate level for tests and full reporting of outcomes |
| ☒ | ☐ | Estimates of effect sizes (e.g. Cohen's *d*, Pearson's *r*), indicating how they were calculated |

*Our web collection on statistics for biologists contains articles on many of the points above.*

## Software and code

Policy information about availability of computer code

Data collection | Histology images were captured using the Metafer Slide Scanning Platform (Metasystems). Raw images were stitched together with the VSlide Software (Metasystems).

Sequencing of spatial transcriptomics libraries were performed on illumina instruments using their proprietary platform and demultiplexed using DRAGEN.

| Data analysis | The manuscript used publicly available, open source R and Python libraries/packages as described in the methods text. Two new libraries were developed for this manuscript, and are made available via Github (https://github.com/aerickso/SpatialInferCNV and https://github.com/almaan/growmeatissue).<br><br>Software and package version used during analysis:<br>Python (3.6.0)<br>R (4.1.3) with packages:<br>- Seurat (3.2.2)<br>- STUtility (0.1.0)<br>- SCTransform (0.3.3)<br>- tidyverse (1.3.1)<br>- infercnv (1.10.0)<br>- hdf5r (1.3.5)<br>- phangorn (2.8.1)<br>- Dendextend (1.15.2)<br>- msigdbr (7.4.1)<br>- fgsea (1.16.0) |
| --- | --- |

For manuscripts utilizing custom algorithms or software that are central to the research but not yet described in published literature, software must be made available to editors and reviewers. We strongly encourage code deposition in a community repository (e.g. GitHub). See the Nature Portfolio guidelines for submitting code & software for further information.

# Data

Policy information about availability of data

All manuscripts must include a data availability statement. This statement should provide the following information, where applicable:
- Accession codes, unique identifiers, or web links for publicly available datasets
- A description of any restrictions on data availability
- For clinical datasets or third party data, please ensure that the statement adheres to our policy

Count matrices, high-resolution histological images and additional material, are available on Mendeley Data (https://doi.org/10.17632/svw96g68dv.1).

Raw fastq for the prostate samples are available on request and is deposited to European Genome-Phenome Archive (EGA, www.ebi.ac.uk/ega/), which is hosted by the European Bioinformatics Institute (EBI) under the study: ID EGAS00001006124. The data are available under Data Use Conditions (DUO) and are limited to non-for-profit use as well as health/medical/biomedical purposes. Access is granted if the above is fulfilled and local institutional review board/ethical review board approvals are provided.

Raw fastq files for the childhood brain tumour samples are available through a Materials Transfer Agreement with Monica Nister (monica.nister@ki.se), in line with GDPR regulations.

Public data used for comparison of phylograms were obtained from European Nucleotide Archive (ENA; http://www.ebi.ac.uk/ena), accession numbers ERP022266 (RNA-seq) and ERP022267 (WGS) as well as from European Genome-phenome Archive (EGA; https://www.ebi.ac.uk/ega/), accession number EGAS00001001659 and EGAS00001000942. Public patient-specific benign cutaneous scRNAseq data were obtained from GEO (GSE144236). Public spatial transcriptomics data used in the study were all obtained from 10x genomics. Human lymph node (https://www.10xgenomics.com/resources/datasets/human-lymph-node-1-standard-1-1-0), breast cancer (https://www.10xgenomics.com/resources/datasets/human-breast-cancer-block-a-section-1-1-standard-1-1-0) and glioblastoma (https://www.10xgenomics.com/resources/datasets/human-glioblastoma-whole-transcriptome-analysis-1-standard-1-2-0) are all available as dataset resources.

# Field-specific reporting

Please select the one below that is the best fit for your research. If you are not sure, read the appropriate sections before making your selection.

☒ Life sciences   ☐ Behavioural & social sciences   ☐ Ecological, evolutionary & environmental sciences

For a reference copy of the document with all sections, see nature.com/documents/nr-reporting-summary-flat.pdf

# Life sciences study design

All studies must disclose on these points even when the disclosure is negative.

| Sample size | This was a biological study and not a clinical trial and therefore we did not undertake a power calculation for the number of patients. For prostate, sample size of two was used as this is what was provided by the urologist. As a exploratory study of prostate cancer heterogeneity and showcase of spatial inferCNV this sample size was deemed sufficient. |
| --- | --- |
| Data exclusions | All patients analysed were included in the data presented. There were no excluded subjects. |
| Replication | All spatial transcriptomics experiments, including histology, of prostate samples were performed in technical replicates of two and a biological replicate in the form of an additional whole prostate. All samples and analyses confirmed the original findings. In addition, technical repeats of data analyses (spatial inferred CNV) was also re-run to confirm analysis results. smFISH and spatial transcriptomics experiments on other tissues were not repeated. |

| Randomization | This was a biological study and not a clinical trial and therefore we did randomize. |
|---|---|
| Blinding | This was a biological study and not a clinical trial therefore we did not blind. All samples processed contained prior known cancer tumors. |

# Reporting for specific materials, systems and methods

We require information from authors about some types of materials, experimental systems and methods used in many studies. Here, indicate whether each material, system or method listed is relevant to your study. If you are not sure if a list item applies to your research, read the appropriate section before selecting a response.

### Materials & experimental systems

| n/a | Involved in the study |
|---|---|
| ☒ ☐ | Antibodies |
| ☒ ☐ | Eukaryotic cell lines |
| ☒ ☐ | Palaeontology and archaeology |
| ☒ ☐ | Animals and other organisms |
| ☐ ☒ | Human research participants |
| ☒ ☐ | Clinical data |
| ☒ ☐ | Dual use research of concern |

### Methods

| n/a | Involved in the study |
|---|---|
| ☒ ☐ | ChIP-seq |
| ☒ ☐ | Flow cytometry |
| ☒ ☐ | MRI-based neuroimaging |

## Human research participants

Policy information about studies involving human research participants

| Population characteristics | Prostate patient 1 and 2 were both male at the age of 82 and 63 years old respectively. Both were diagnosed with prostate cancer and had radical prostatectomy performed. No genotyping was performed on the patients. |
|---|---|
| Recruitment | Candidate subjects diagnosed with primary prostate cancer whom were to undergo radical prostatectomy were identified and randomly selected by one of the study pathologists (AT). The two human subjects were provided with full and adequate verbal and written information about the study before their participation. Written informed consent was obtained from all participating subjects before enrolment in the study. |
| Ethics oversight | The study was performed according to the Declaration of Helsinki, Basel Declaration and Good Clinical Practice. The study was approved by the Regional Ethical Review Board (REPN) Uppsala, Sweden before study initiation (Dnr 2011/066/2, Landstinget Västmanland, Sari Stenius), Regional Ethical Review Board (EPN), Stockholm, Sweden (DNR 2018/3-31, Monica Nister). |

Note that full information on the approval of the study protocol must also be provided in the manuscript.

