## [Peer Review File · Nature]

Manuscript Title: Spatially resolved clonal copy number alterations in benign and malignant tissue.

Reviewer Comments & Author Rebuttals

Reviewer Reports on the Initial Version:

Referees' comments:

Referee #1 (Remarks to the Author):

Erickson, Berglund et al present a study of the spatial distribution of genome-wide copy number variation in many regions from both benign and malignant tissue, in a total of ~120,000 spots. The main organ sampled is the prostate (n=2, ~88,000 spots), but regions from a lymph node, skin, medulloblastoma, adult glioblastoma and ductal breast cancer are also subjected to spatial transcriptomics. Based on inferred copy number variants from the transcriptomic data, regions with CNVs are then assigned to clones and ordered into phylogenies. In prostate and skin, these clones are focused on regions with histological abnormalities, but are also present in benign cells. Together, this data shows the presence of genetically abnormal clones that seem to act as precursors to full-blown cancer, or that arise in parallel.

While I think the data and the observations made here are of interest and importance to the field, I believe the analysis of the data could be expanded to provide more biological insights. In addition, I have some methodological concerns with the correctness of the phylogenies derived from the CNV profiles. I have detailed my comments below.

1. I believe the title of the paper promises more than is achieved in the paper. While CNVs are an important category of somatic mutations, they are far from the only one. Single nucleotide variants, insertions, deletions and structural variants each play crucial roles in somatic evolution. None of these have been (or can be adequately) derived from the transcriptomic data. Therefore, I would suggest rephrasing the "landscape of clonal somatic mutations" from the title to reflect this limitation.

2. The observation that clones carrying CNVs are comprised of benign cells adds to the research emerging over the past few years that many "normal" cells carry genomic alterations that have been associated with cancers, such as driver mutations, sometimes even representing precursor lesions. Since these studies rely on DNA sequencing rather than transcriptomics, they cannot shed light on whether these premalignant cells exhibit a different behaviour compared to their unmutated counterparts. Rather than simply showing the existence of benign clones with CNVs, the transcriptomic data used here should allow for an assessment of functional differences between benign clones with and without CNVs, or between those belonging to different clones with different sets of CNVs. Do benign cells with CNVs already show expressional features of malignant cells? Presumably, given that some clones differ by only a few CNVs, would it be possible to directly assess the effect of those CNVs on the transcriptome (besides the obvious dosage effect on the genomic region affected by the CNV)? These are just a few ideas, but I believe there are interesting biological angles to the data that have remained unexplored.

3. While it is clear to me that some GEFs are associated with certain clones, I currently find their biological relevance somewhat unclear and wonder whether a more in-depth look at these GEFs will reveal interesting biology (in a similar way as highlighted in the previous point). Is the difference in expression captured by the GEFs solely due to the CNVs that are carried by those clones?

4. The observation that both benign and malignant cells can be part of the same clone is interesting but begs the question of why some of these cells remain benign with identical CNV landscapes. Can the authors speculate on this?

5. I have some serious concerns regarding the construction of the phylogenies from the iCNV data.

- Extended Data Fig. 1a: While the samples in the right-hand phylogeny (derived from DNA) line up in the same way as the groups in the left-hand side (derived from RNA), the tree topology from the DNA-based phylogeny allows many more configurations due to the large multifurcation in the centre. This suggests that the RNA-based iCNV method groups samples into clones that do not seem to be fully supported by the DNA, contrary to what is said in the text. Instead of the current x-axis (which is unclear to me as to what it represents), a phylogeny is conventionally displayed with the axis beginning at the root of the tree and branch lengths reflecting the number of genetic events, such as the number of inferred acquired CNVs. It would be pertinent to show this corroboration as it underpins the methodology.

- Extended Data Fig. 1b/c: in a similar way to the previous point, the iCNV tree topologies in Extended Data Fig. 1b/c do not seem to mirror the ones from the published data correctly, but it is difficult to judge if the dendrogram is based on hierarchical clustering rather than a proper phylogeny. In b, one would expect the liver mets to coalesce with clone A before coalescing with clone I (etc.). Similar discrepancies are present in panel c.

- The way in which the phylogenies in Figures 2 and 3 are constructed from the iCNV data is unclear to me and seems erroneous. For instance, all clones in Fig. 2 besides clone A share "6q_loss_2", but this apparent shared event is completely absent from the tree topology. There are

- The methods section does not list a specific algorithm for tree building, which leads me to believe they were constructed manually. To sort out the discrepancies, it would be imperative to algorithmically build trees. Given the assumptions of shared CNVs likely denoting shared ancestry and CNVs generally not being lost once acquired, a maximum parsimony framework seems appropriate. If the data supports more than one phylogeny, it would be good to provide alternative solutions.

- Within the phylogenies, it would be good to annotate branches with the CNVs that occurred on them.

6. To see CNVs observed in different clones are due to shared ancestry rather than multiple, independent acquisitions of the same CNV, could the authors assess whether it is the same allele that is amplified/lost across all clones with those CNVs from SNPs in the transcriptomic data? In case there is a CNV violating the topology of the phylogeny, this might be proof of an independent acquisition.

7. In Fig. 4 c/d, there seem to be many CNVs in stromal cells, some of which seem to be shared with the squamous cell cancer, such as loss of 8q (in clones A, B and D), gain of 1q (A and B) and loss of 19q (A, D, possibly B). Intuitively, given that all these cells seem to share some CNVs, it could indicate all derive from a single cell that carried those CNVs. However, this seems to suggest a rather large clonal expansion traversing cell types. Is it likely these represent artefactual calls?

Some minor points:

8. The introductory paragraph lacks references.

9. I believe using subheadings would increase the readability of the manuscript.

10. Fig. 2a: Why do the cells assigned to clone J appear to have a CN profile much less noisy than cells belonging to other clones? Is the denoising specific to each clone? This seems to be the case for clone C in Fig. 4c as well.

11. On page 5, line 12: 16q is listed to be both gained and lost, which I think is an error.

12. I cannot find any clinical information on the patients, such as the age of the men whose prostates were sampled. Are the large clones in prostate due to advanced age of the men, at which point such benign clonal expansions might be expected?

Referee #2 (Remarks to the Author):

The broad goals of this study is to elucidate the development of cancer from benign tissue, improving our understanding of disease progression and enabling earlier diagnosis of cancer. Specifically, the authors have used spatial transcriptomics to infer genome-wide copy-number variations (CNV) in situ. Using the 10X Genomics Visium system, the authors comprehensively analyzed a full width axial section of two prostate specimens in addition to several other tumor types. The authors convincingly demonstrate their ability to resolve in situ transcriptomic data that represents highly localized subclonal populations within the prostate. Using their spatially defined transcriptomic data, the authors infer somatic copy number alterations as others have done with single cell data. The authors then used these inferred CNAs (iCNAs) to construct detailed phylogenies of tumor development with profound spatial resolution.

One of the key findings of the manuscript is the discovery of 'benign' subpopulations of prostate cancer cells that nonetheless carry deleterious oncogenic iCNAs. Figure 3 demonstrates this finding for MYC amplification and PTEN loss in both 'benign' and tumor tissues. Additional strengths of this manuscript include the robust computational validation of iCNAs comparing with WGS and FISH data, and the demonstration of the complementary roles of transcriptomics, histopathology and iCNAs in resolving unique aspects of tumor heterogeneity. The use of multiple tumor types to support utility of the iCNA method in detecting alterations in benign tissue is also noteworthy. The study is well done and represents an important advance, although previous studies have documented oncogenic mutations in benign tissue. (E.g. PNAS 1996 doi: 10.1073/pnas.93.24.14025, Science 2015 doi: 10.1126/science.aaa6806, and Science 2019 doi: 10.1126/science.aaw0726.

There are a few questions that need the attention of the authors as follows:

1. One of the most striking findings of this study is that copy number alterations are evident in "benign" tissue as defined by two pathologists. Utilization of clinical stage markers e.g. AMACR/HMW cytokeratin staining could be helpful in further validating the 'benign' nature of these regions.
2. The authors' speculation on page 7 line 21-23 that low-grade prostate cancer is fundamentally distinct from higher grade cancer, as opposed to being in an evolutionary continuum is overly strong considering that they present evidence from only two patients at a single point in time. Additionally, a more nuanced discussion of the translational relevance e.g. targeted treatment for subclones (p9 line 11-14), and contextualization in light of how this analysis might be implemented when most men undergo TURPs rather than open resection, are warranted.

Referee #3 (Remarks to the Author):

I have reviewed the manuscript presented by Erickson and colleagues in which they present the application of inferred CNV profiles at 55um spatial resolution in various normal, benign and tumorous tissues. The authors present results of an algorithm which infers the CNV profiles from ST/Visium data, which is demonstrated on a synthetic in silico dataset, demonstrate its application to decipher heterogeneity in nearly a full planar section of a resected prostate, followed by demonstrating it on a number of other tissues.

The application of spatial resolved transcriptomics (SRT) to decipher spatial genomic copy numbers accurately is exciting and much needed in the field. The results presented in the manuscript suggest that the authors can indeed do this.

However, I feel that the focus of the paper lacks clarity and this in turns questions the novelty of

tis paper. On the one hand, the paper reads like a methods paper, but the details of the computational methods are lacking so it becomes difficult to compare their approach to e.g. Kueckelhaus et al 2020 (<https://doi.org/10.1101/2020.10.20.346544>) who have also inferred CNVs from SRT data. The authors also do not compare their tool to a similar published tool called STARCH (<https://doi.org/10.1088/1478-3975/abbe99>). On the other hand, they describe the clonal CNV heterogeneity in multiple tumour types (including an impressive full slice of prostate tissue), however do not go into details of the biology nor extensively validate their ST/Visium findings using non-spatial single cell sequencing nor provide extensive comparisons between the inferred CNV and transcriptional landscape.

Never-the-less, the paper is a good start to what could be a very good piece of work. I hope that with the following points I can describe some of my specific concerns.

Major

[1] Focus of the paper. Is this a methods paper, or a biology paper? The title and abstract point to the biology of the prostate cross section, but the start and end of the paper focus on the methodological aspect. Given that the concept of calling CNVs from ST/Visium data has already been described and demonstrated by others, I would recommend focusing more on the biological results.

[2] Novelty of inferring CNVs. Inference of spatial CNV profiles has already been described by Kueckelhaus et al 2020 (<https://doi.org/10.1101/2020.10.20.346544>) where they used inferCNV to infer CNVs, Lu et al 2021 (<https://doi.org/10.1186/s13058-021-01451-6>) where they use LCM and bulk WGS+RNAseq. I believe that the only published computational tool that infers CNVs from ST/Visium data is STARCH, presented by Elyanow and colleagues (<https://doi.org/10.1088/1478-3975/abbe99>) in 2021 with initial commits to their GitHub repo made in May 2020. Given the concept of applying scRNAseq CNV inferences tools to ST data has already been described and that a method for inferring CNVs from ST data exists, I believe that the authors should try to go to the same lengths as Elyanow and colleagues in demonstration and benchmarking of their computational approach. Elyanow also present pretty much the same 1D-to-2D synthetic in silico tissue benchmark. It would suggest that the comparison include performance of inferCNV as described by Kueckelhaus et al, similarly using CopyKat (Gao et al <https://doi.org/10.1038/s41587-020-00795-2>), and also perhaps considering also the simulated spatial data from scRNAseq as demonstrated in STARCH.

[3] Validation of the inferred CNV profiles. While the authors do indeed validate the MYC and PTEN CNAs via FISH, and validate the CNV calling on matched scRNAseq+scDNAseq data, they do not extensively validate their results from spatial data. I believe that validation via single cell DNA sequencing of at least one heterogeneous section of the prostate tissue should be provided. If this is not possible, then the authors should try to source a similarly heterogeneous piece of tissue to profile using Visium and scDNAseq.

[4] Integration of scRNAseq.

[4.1] The authors seems to be missing in depth description of the consistency and differences of gene expression profiles with the inferred CNV patterns. One would assume that these should be similar, but the extent of this is not described.

[4.2] The "normal" tissue for the CNV inferences should ideally be the cell type that is the cell of origin of the tumour, however this basic analysis is ignored.

[4.3] The authors do not consider that spots could contain a mixture of tumour and normal cells, e.g. TILs. This is a concept that a number of co-authors should be familiar with. It would be nice to see if the authors can also consider the tumour cell content of spots in the their model using supervised decomposition methods (for samples with matched Visium and scRNAseq data) or de novo deconvolution methods (if such tools are accurate). Alternative, the authors could try to segment DAPI stained slides and perhaps use this to model the number of cells within a spot to assist with deconvolution. At the very least this should be demonstrated on one the heterogeneous

prostate cancer slide H1_4.

Minor

[5] The manuscript would benefit from subheadings that separate out results sub sections and the discussion

[6] The use of the term "inferCNV" in the manuscript is inappropriate given that there is another method called inferCNV... even more so considering that the authors also refer specifically to this tool in their manuscript.

[7] CNV as a term is synonymous with polymorphic variants, but the term CNA (CN-alteration or CN-aberration) is more common for cancer genomics. Alternatively, the authors can use the term "somatic CNVs".

[8] The choice of T for the GEF analysis was not rationalised. Was this an informed choice based on a factorisation metric such as silhouette or FN? Or was this based of other statistics such as AIC, BIC, or the "elbow method"?

[9] Similar to point [8], the authors do not actually show that the GEF modelling is converging at 5000 iterations. It is very likely that it is, but this should be stated.

[10] The colour pallet for clusters is hard to follow at times. E.g. in figure 2 and 3, the authors use a large variety of red colours for the various CNV clusters – these are very difficult to see in subsequent panels, e.g. I cannot tell the different between the colour for clone E and F in Figure 3d.

[11] The results of the prostate tumour sample could implicate a poly-clonal seeding of tumors. Did the authors check to see if this patient carries a predisposition allele to prostate tumors or cancer?

[12] The authors could provide more insights into application of their method to their own large datasets, e.g. <https://doi.org/10.1038/s41551-020-0578-x>.

[13] Can authors say something about ERG on chr21 (seems amplified in clone C)?

[14] The method for inferring CNVs doesn't seem well described. Or am I mistaken and the authors only use the existing "inferCNV" tool to inferCNVs?

[15] Github repo link doesn't work.

[16] GEF CNV profiles. In Ext Fig 3 the authors shows that the GEF harbour what looks like different CNV clones. However, I find this analysis lacking. It only compares the authors approach of GEFs and not more established methods such as clustering via Leiden/Louvain+SNN. Also, the authors could further look into sub-clustering of gene expression of e.g. GEF10 to see if these can indeed be separated at gene expression level. Likewise, the authors do not elaborate on how different GEF17 and GEF7 are w.r.t. transcriptional profile given their similarity at the CNV level.

Line specific issues [Page 1, Line 2 = P1L2]

[P2L2] Page 2 Line 2 (P2L2). Should this rather be "polymorphisms" that are inherited rather than "alterations"?

[P2L3] How sure are we that somatic mutations are only in a small fraction of cells? Wouldn't a somatic mutation in the early stages of development be present in many cells? Uniparental disomy? Perhaps the wording should be re-evaluated by the authors.

[P2L5] Somatic alterations can also happen in normal tissue.

[P2L6] Perhaps add some references?

[P2L11] Is spatial transcriptomics a "genome-wide" methodology? By extensions, I would find it hard to say that a gene-expression microarray is also "genome-wide" – genomes contain mainly more features than genes, including regulatory elements, repeats, centromeres, telomeres, etc.

[P2L14] Missing space in between "infer" and "CNV"

[P2L16] This is hard to follow. "inferCNV" is not actually introduced as a new method. Also see pints [P2L14] and [6].

[P2L18] The use of "successful" is questionable without statistical support. While the reconstructions seem OK, it is not perfect, e.g. Ext fig 1e has clone "E" in the wrong clade. Support with statistics such as RF score, Matching Split Distance, etc. The R package TreeDist

(<https://github.com/ms609/TreeDist/>) implements a few useful metrics. This should also be compared to some of the other existing tools – see point [2].

[P3L5-6] The authors should mention that the samples was resected from a prostate tumour patient.

[P3L11] Erroneous space in between 21 and 000.

[P3L14] “GEFs” is not a well established term, so perhaps a citation would be useful.

[P3L19-20] The authors refer to these interesting regions in 3 different ways in the text. On P3L19-20 they are regions with “increased iCNV activity”, on P4L3 they are “seven key regions”, and on P4L20 they are described as “regions of interest”. This should be consistent.

[P3L21] I believed the authors want to say that “iCNVs could be used to distinguish this regions,...”.

[P4L5] Erroneous space in between 30 and 000.

[P4L7-8] Would it flow better if the authors describe the pathology annotation of spots when the Visium experiment is first described, e.g. at the end of P4L3.

[P4L8] Perhaps “investigating clonal relationships” could be more precisely termed as clonal evolution patterns?

[P4L19-20] Perhaps this could be better illustrated using a Sankey Flow plot of GEFs to CNV clusters.

[P4L21] Do you mean a “phylogenetic tree”? Did the authors “construct” or “compute” this tree?

[P6L8-9] Can the authors relate these CNVs to oncogenes and tumour suppressor genes that have been described (e.g. PTEN and MYC).

[P7L14] Should “low grade cancer” rather be “benign tumour”, or “low grade tumour”? In some models, such as the Vogelstein CRC progression model, it is only a late loss of TP53 that is the “carcinogenic” transformation from the adenoma (which would already be harbouring a loss of chr18 with SMAD2/4).

[P10L5] Is this consistent with P7L14?

[P10L6-8] Haven’t there been studies of normal oesophageal tissue which show the same thing?

Figures

[Fig1f] Is it worrying that the tumour associated GEFs are not distinct from the “normal” GEFs?

[Fig2a etc] Scale is not consistent with “CNV state” in Extd Fig2.

[Fig4g] missing prostate.

[Ext Fig1a] the dendrogram can be flipped to align better between iCNV and WGS-Ginko. E.g. the green and blue parts of the dendrogram for iCNV can be flipped with no change in interpretation.

[Ext Fig2b] Why is the normal state set at “1” and not “2” for diploid? Does 0 indicate loss of 1 copy or 2 copy?

[Ext Fig2d,e] Perhaps using a seriation algorithm would make the 2 panels more comparable.

[Fig 3e] Low resolution. Does the centromere control of chr8 duplicate in clone C, but get lost in clone G? Is this expected? Figure 3A doesn’t seem to imply that there should be centromere amplification in chr8.

Author Rebuttals to Initial Comments:

Editor Comment (Michelle Trenkmann, Senior Editor;)

Michelle Trenkmann, Senior Editor, Biology, London

Education: Advanced degree in Biochemistry, University of Leipzig; PhD in Molecular Biology, University of Zurich; postdoctoral work, University College Dublin.

Areas of responsibility include: genetics, genomics and molecular evolution.

michelle.trenkmann@nature.com*

Your manuscript entitled "The spatial landscape of clonal somatic mutations in benign and malignant tissue" has now been seen by 3 referees, whose comments are attached below. While they find your work of potential interest, as do we, they have raised important concerns that in our view need to be addressed before we can consider publication in Nature.

We note that the referees have set quite a high bar for the revisions but we do agree with them that a rewrite and refocusing (as per reviewers #1 and #3) as well as validation of findings (all referees) and making better use of the data (that is, explore them better for more in-depth biological insight; referees #1 and #3) will be required for further consideration in Nature.

Should further experimental data and or analyses allow you to address these criticisms, we would be happy to consider a revised manuscript (unless something similar has been accepted at Nature or appeared elsewhere in the meantime). However, please bear in mind that we will be reluctant to approach the referees again in the absence of major revisions.

Thank you for these editorial comments. As above, we have addressed the reviewers comments by performing new experiments and undertaking additional data analysis and we are delighted to describe these major revisions in detail below.

All references to page and lines are from the manuscript file containing tracked changes.

Referee #1: somatic evolution

Referee #2: prostate cancer genomics

Referee #3: spatial transcriptomics

Referee #1 (Remarks to the Author):

Erickson, Berglund et al present a study of the spatial distribution of genome-wide copy number variation in many regions from both benign and malignant tissue, in a total of ~120,000 spots. The main organ sampled is the prostate (n=2, ~88,000 spots), but regions from a lymph node, skin, medulloblastoma, adult glioblastoma and ductal breast cancer are also subjected to spatial transcriptomics. Based on inferred copy number variants from the transcriptomic data, regions with CNVs are then assigned to clones and ordered into phylogenies. In prostate and skin, these clones are focused on regions with histological abnormalities, but are also present in benign cells. Together, this data shows the presence of genetically abnormal clones that seem to act as precursors to full-blown cancer, or that arise in parallel.

While I think the data and the observations made here are of interest and importance to the field, I believe the analysis of the data could be expanded to provide more biological insights. In addition, I have some methodological concerns with the correctness of the phylogenies derived from the CNV profiles. I have detailed my comments below.

We thank the reviewer for their engagement with our manuscript and for providing such a detailed critique. We have endeavoured to respond to each comment in turn including additional analyses and changes to the text where indicated. Original reviewer comments are in black with our responses in blue. Where we mention pages & lines we are specifically referencing the tracked changes version of the manuscript.

1. I believe the title of the paper promises more than is achieved in the paper. While CNVs are an important category of somatic mutations, they are far from the only one. Single nucleotide variants, insertions, deletions and structural variants each play crucial roles in somatic evolution. None of these have been (or can be adequately) derived from the transcriptomic data. Therefore, I would suggest rephrasing the “landscape of clonal somatic mutations” from the title to reflect this limitation.

We acknowledge that the reference to “somatic mutations” was overly broad given the original focus on copy number variants. As prompted by the reviewer, we have actually now included a single nucleotide variant analysis of spatial transcriptomics data to complement the presented CNV data as prior studies have shown that it is possible to identify SNVs from single cell RNA sequencing data (Schnepp et al, doi.org/10.1093/hmg/ddz207; Liu et al, doi.org/10.1186/s13059-019-1863-4). Our SNV analysis is based on the work by Petti et al, doi.org/10.1038/s41467-019-11591-1 demonstrating the possibility to identify mutations in single cell RNAseq data (please see Reviewer 1, comment 6) . We have therefore left the title unchanged, with the inclusion of the new SNV analysis into the revised manuscript (please see page 7, lines 16-20), but would, of course, still be happy to amend the title if requested by the reviewer/editorial team.

2. The observation that clones carrying CNVs are comprised of benign cells adds to the research emerging over the past few years that many “normal” cells carry genomic alterations that have been associated with cancers, such as driver mutations, sometimes even representing precursor lesions. Since these studies rely on DNA sequencing rather than transcriptomics, they cannot shed light on whether these premalignant cells exhibit a different behaviour compared to their unmutated counterparts. Rather than simply showing the existence of benign clones with CNVs, the transcriptomic data used here should allow for an assessment of functional differences between benign clones with and without CNVs, or between those belonging to different clones with different sets of CNVs. Do benign cells with CNVs already show expressional features of malignant cells? Presumably, given that some clones differ by only a few CNVs, would it be possible to directly assess the effect of those CNVs on the transcriptome (besides the obvious dosage effect on the genomic region affected by the CNV)? These are just a few ideas, but I believe there are interesting biological angles to the data that have remained unexplored.

As requested by the reviewer we have now undertaken an assessment of gene expression in order to assess the functional differences between benign (clone A), altered benign (clone C) and tumour clones (clones E, F and G). We performed differential gene expression of the clones presented in Figure 3, identifying differentially expressed genes (DEG) as outlined below.

In particular, we observed that the gene expression profile of Clone C (benign but copy number altered cells) displays a broadly similar expression status when compared to the histological transformed clones E, F and G, including for example altered expression of genes normally thought to be specifically linked to malignant transformation (e.g. reduced expression of MSMB, increased expression of GDF15).

We compared individually each of altered benign Clone C, and tumour clones E-G with normal benign clone A and observed that the top hallmark GSEA Gene Ontology was MYC signaling.

We include some additional biological observations centred on similarities and differences in undertaking DEG analysis for ‘altered benign’ Clone C and ‘malignant’ Clones E-G compared to ‘normal’ diploid Clone A. We note that malignant-only genes point towards a settled phenotype that has overcome intrinsic and extrinsic stresses. By contrast, ‘altered benign’ only genes point towards a high energy, high protein folding phenotype that is trying to overcome intrinsic and extrinsic stresses. Altered benign display reduced AR signaling (using a published AR activity signature; Bluemner et al; doi: 10.1016/j.jccell.2017.09.003). We believe that Clone C may be a pre-cancerous state that, if it survives extrinsic and intrinsic stresses, may become cancerous.

We have added a whole new subsection on the above in the results (page 7, lines 21-22 and page 8, lines 1-13), a subsection in the methods (page 33, lines 10-20), as well as in the discussion (page 12, lines 8-11). We have added a new Extended Data Figure 7 and Supplementary Table 5 to the manuscript as well as the individual DEG analyses (including raw DEG lists and GO terms) to Mendeley for others to further analyze the data.

In terms of assessing direct effects, we feel it is important to note that our ability to assess the effect of specific CNVs on the transcriptome is limited by the fact that the CNV calls have been generated from the same data. We are therefore keen to avoid a logical fallacy on this point. We have therefore limited our observations on this to CNV-associated DEGs, identifying potential ‘drivers’ of phenotype such as c-Myc and others (listed in supplementary material in Mendeley, example below).

Clone C (Altered Benign) vs Clone A (Benign Benign)
 Up = Higher in Clone C (Altered Benign), Down = Higher in Clone A (Benign Benign)

Gene	p_val	avg_log2FC	pct.1	pct.2	p_val_adj	Clone	ENSMBLID	state	chr	start	end	iCNVLabel	
NUPR1	1.18E-73	2.039929	1	0.951	3.96E-68	Clone_C	ENSG00000176046		4	chr16	28097979	28211920	Clone_C_Amp
SAT1	1.14E-70	1.896096	1	0.942	3.83E-66	NA	NA	NA	NA	NA	NA	NA	
AZGP1	2.42E-68	-1.82458	0.995	0.996	8.13E-64	NA	NA	NA	NA	NA	NA	NA	
TMEFF2	2.58E-68	2.488912	0.986	0.458	8.65E-64	NA	NA	NA	NA	NA	NA	NA	
MTIE	2.86E-68	-3.40734	0.447	0.969	9.61E-64	Clone_C	ENSG00000169715		2	chr16	56688584	56699497	Clone_C_Del
RPL22L1	3.27E-68	1.949705	0.995	0.542	1.19E-63	Clone_C	ENSG00000163584		4	chr3	1_7E+08	1_7E+08	Clone_C_Amp
NEFH	7.42E-68	-2.98554	0.226	0.942	2.49E-63	NA	NA	NA	NA	NA	NA	NA	
ADIRF	1.37E-67	-2.42183	0.954	0.987	4.60E-63	Clone_C	ENSG000000148671		2	chr10	86958599	86963258	Clone_C_Del
OR51E2	1.76E-67	2.356214	0.963	0.307	5.92E-63	NA	NA	NA	NA	NA	NA	NA	
ANPEP	4.66E-67	-2.94763	0.152	0.924	1.50E-62	NA	NA	NA	NA	NA	NA	NA	
TRGC1	5.60E-67	2.442872	0.995	0.747	1.88E-62	Clone_C	ENSG000000211689		4	chr7	38239580	38249572	Clone_C_Amp
MSMB	1.92E-66	-3.68793	1	1	6.45E-62	NA	NA	NA	NA	NA	NA	NA	
RPL35A	4.55E-66	1.112294	1	0.996	1.53E-61	Clone_C	ENSG00000182899		4	chr3	1.98E+08	1.98E+08	Clone_C_Amp
RPL4	9.08E-65	1.508934	1	0.996	3.05E-60	NA	NA	NA	NA	NA	NA	NA	
FABP5	1.27E-63	3.068543	0.972	0.529	4.27E-59	Clone_C	ENSG00000164687		4	chr8	80967810	81112068	Clone_C_Amp
COX6C	4.20E-63	1.281236	1	0.951	1.41E-58	Clone_C	ENSG00000164919		4	chr8	99013266	99877580	Clone_C_Amp
MT2A	1.14E-62	-2.30556	0.935	0.991	3.84E-58	Clone_C	ENSG00000125148		2	chr16	56466836	56520087	Clone_C_Del
TRPM4	2.54E-62	1.797932	0.995	0.836	8.52E-58	NA	NA	NA	NA	NA	NA	NA	
RPL17	1.19E-61	1.155741	1	0.96	4.00E-57	NA	NA	NA	NA	NA	NA	NA	
TIMP1	9.60E-61	-2.31054	0.553	0.951	3.22E-56	NA	NA	NA	NA	NA	NA	NA	
RPL7A	1.65E-60	1.005751	1	0.996	5.53E-56	NA	NA	NA	NA	NA	NA	NA	
MIF	1.25E-59	1.403189	0.995	0.947	4.19E-55	NA	NA	NA	NA	NA	NA	NA	
PUPP1	1.47E-59	1.406218	1	0.96	4.92E-55	NA	NA	NA	NA	NA	NA	NA	
B2M	1.74E-59	-1.40413	1	0.996	5.84E-55	Clone_C	ENSG000000166710		2	chr15	44562696	44663678	Clone_C_Del
ACTG1	1.39E-58	1.020255	1	0.982	4.66E-54	NA	NA	NA	NA	NA	NA	NA	
TF3	1.62E-58	2.252432	0.977	0.609	5.42E-54	Clone_C	ENSG000000160180		4	chr21	42199689	42297244	Clone_C_Amp

3. While it is clear to me that some GEFs are associated with certain clones, I currently find their biological relevance somewhat unclear and wonder whether a more in-depth look at these GEFs will reveal interesting biology (in a similar way as highlighted in the previous point). Is the difference in expression captured by the GEFs solely due to the CNVs that are carried by those clones?

Thank you for highlighting this. GEFs represent transcriptional programs in a spot, or groups of spots, and are identified by a factorisation approach (as reported previously Berglund et al, doi: 10.1038/s41467-018-04724-5). They are broadly linked to histological landmarks but the presence of multiple GEFs in tumor regions, represents expected prostate tumor heterogeneity. Interestingly the GEF analysis was a starting point for us in trying to delineate clonal boundaries. In the previous version of the manuscript we demonstrated the spatial relationship of GEFs to siCNVs (Ext Data Fig 3). This visualization highlights CNV heterogeneity within the GEFs (see particularly, the single GEF 10 with multiple inferred CNV clones Extended Data Figure 3e, reproduced here for the reviewer). To further outline the relationship between GEFs and tumor clones we have now included a Sankey Flow plot (Ext Data Fig 3f). This plot demonstrates that certain GEFs have a single relationship to an inferred clone while other GEFs represent multiple clones. In contrast, spatially inferred CNV (siCNV) delivered broadly homogenous groupings, and we therefore moved to siCNV as a preferred strategy to delineate clone structure. For this reason, as prompted by the reviewer, we have focussed our biological differential gene expression interrogation on these clonal groupings (please see the response to comment 2). Nonetheless, we undertook further DEG analyses of GEFs 7 and 17 and found approximately n = 1800 DEGs, despite the similarity in iCNV profiles, and the DEG lists have been added to Mendeley (please also see response to reviewer 3, comment 16).

4. The observation that both benign and malignant cells can be part of the same clone is interesting but begs the question of why some of these cells remain benign with identical CNV landscapes. Can the authors speculate on this?

We certainly agree that this is interesting. Clonal status alone, and the mutational events that give rise to heritable clonal lineages at cell division do seem to be insufficient to deliver immediate phenotypic transformation. We believe our work generates interesting hypotheses regarding the environmental effect with, for example, the stromal niche or cross-talk between neighbouring clones, or indeed regarding the timing of events and how long is needed for morphological transformation to occur. Additionally, there could be other somatic events that we have not captured with ST, as mentioned by the reviewer in their first comment.

In recognition of the biological significance of this finding, we have expanded the discussion to include a brief commentary on this page 12, lines 2-11.

“It seems that clonal status alone, and the somatic events described here retained in heritable clonal lineages at cell division, are insufficient to deliver immediate phenotypic transformation. We believe our work generates interesting hypotheses regarding epigenetic determinism²⁸ and the environmental effect with, for example, the stromal niche or cross-talk between neighbouring clones. Furthermore, questions remain about the timing of events and how long is needed for morphological transformation to occur. Expression analysis of these altered benign clones revealed changes consistent with enhanced phenotypic versatility suggesting that these cells may represent an intermediate state between benign and malignant cells - metabolically active as they try to survive the mutational burden they have acquired, prior to phenotypic transformation.”

5. I have some serious concerns regarding the construction of the phylogenies from the iCNV data.

- Extended Data Fig. 1a: While the samples in the right-hand phylogeny (derived from DNA) line up in the same way as the groups in the left-hand side (derived from RNA), the tree topology from the DNA-based phylogeny allows many more configurations due to the large multifurcation in the centre. This suggests that the RNA-based iCNV method groups samples into clones that do not seem to be fully supported by the DNA, contrary to what is said in the text. Instead of the current x-axis (which is unclear to me as to what it represents), a phylogeny is conventionally displayed with the axis beginning at the root of the tree and branch lengths reflecting the number of genetic events, such as the number of inferred acquired CNVs. It would be pertinent to show this corroboration as it underpins the methodology.

We apologize to the reviewer that this was unclear; we used the results in Extended Fig 1 to test automatically obtaining clone calls from inferCNV outputs. We identified, as the reviewer has noted, that clone calling from “raw” inferCNV, while similar, does have discrepancies from the published results. This is why we developed our own tree building approach (used in Figures 2 and 3) with inferCNV pre-processing to identify clusters. Although performed manually, we consider this approach to be algorithmic as we follow a set of pre-defined criteria when constructing the trees to ensure consistency and reproducibility. We describe these in detail in the methods but have made this much clearer by noting the discrepancies between automated clone calling and published phylogenies on page 3, lines 1-2, and specifying a new method subheader “**Manual, Algorithmic Tree Building from Pre-Processed inferCNV Data**”, and numbering chronologically the relevant methods subsections (**1. Clone Tree consensus siCNV event calling, 2. Clone Trees – Branch Lengths, and 3. Clone Trees – Clone Diameters**), resulting in further changes on pages 29 and 30. Further, we have made access to the GitHub repository available (please see details in response to reviewer 3, comment 14) so that the reviewer may also view how the clustering is performed. In addition, we have made improvements to the Extended Data Fig 1 as also noted by reviewer 3, in comment “Ext Fig1a”.

- Extended Data Fig. 1b/c: in a similar way to the previous point, the iCNV tree topologies in Extended Data Fig. 1b/c do not seem to mirror the ones from the published data correctly, but it is difficult to judge if the dendrogram is based on hierarchical clustering rather than a proper phylogeny. In b, one would expect the liver mets to coalesce with clone A before coalescing with clone I (etc.). Similar discrepancies are present in panel c.

The reviewer's observation highlights where the trees suggested by the inferCNV package fail to capture relationships identified in DNA-based tree-construction. The dendrogram presented is based on hierarchical clustering performed by inferCNV. Hierarchical clustering in itself captures similarity between clusters through a distance metric, and does not reflect the sequential gain of genetic alterations that characterise cancer evolution. Therefore it is not surprising that the unprocessed inferCNV output performed sub-optimally. This motivated the development of our new approach detailed in response to the point above.

- The way in which the phylogenies in Figures 2 and 3 are constructed from the iCNV data is unclear to me and seems erroneous. For instance, all clones in Fig. 2 besides clone A share "6q_loss_2", but this apparent shared event is completely absent from the tree topology. There are [sic]

Thank you for the chance to elaborate on this. We followed examples in Gundem et al. (<https://doi.org/10.1038/nature14347>), and Woodcock et al. (<https://doi.org/10.1038/s41467-020-18843-5>), where we selected a handful of informative copy-number changes to annotate the pattern detected in the clones. As mentioned above, we hope that our manual, algorithmic approach is made much clearer by changes in the manuscript on lines page 29, lines 7, 8, and 16 as well as page 30, line 1. As mentioned in subsection "1. Clone Tree consensus siCNV event calling", "Both HMM siCNVs, and manual interpretation of denoised outputs were used to identify putative subclonal CNVs. These were then merged in a final consensus set for building clone trees." Regarding the specific CNVs underpinning the trees, we had previously included this information with accompanying detail in Supplementary Table 1 and 2. On prompting by this reviewer we have revised the tables to detail the events within Figures 2 and 3, within the clones, and including unobserved common ancestors, and in the branches.

Here is a representative screenshot of the revised Supplementary Table 1.

	A	B	C	D	E	F	G	H	I	J	K	L	M
1	Clone A	Clone B	Clone C	Clone D	Clone E	Clone G	Clone F	Clone H	Clone I	Clone J	Clone K	ClonesGan	ClonesGanCl
2	2p_Gain_1	1q_Loss_2	1q_Loss_2	1p_Gain_1	3p_Loss	1p_Loss_1	3p_Loss	1p_Loss_1	1p_Loss_1	1p_Loss_1	1p_Loss_1	3p_Loss	3p_Loss
3	2p_Gain_2	6q_Loss_2	2q_Loss	1q_Loss_2	5q_Loss	3p_Loss	5q_Loss	2q_Loss	2q_Loss	2q_Loss	1q_Loss_1	5q_Loss	5q_Loss
4	3p_Gain_1	10p_Loss	3p_Loss	2q_Loss	6p_Gain	5q_Loss	6q_Loss_2	3p_Loss	3p_Loss	3p_Loss	2q_Loss	6q_Loss_2	6q_Loss_2
5	3p_Gain_2	16p_Gain	4p_Gain_1	3p_Loss	6q_Loss_2	6q_Loss_2	7p_Gain	5q_Loss	4p_Gain_1	4p_Gain_1	3p_Loss	7p_Gain	7p_Gain
6	4p_Gain_2	17q_Gain	4q_Loss	4p_Gain_1	7p_Gain	7p_Gain	7p_Loss	6p_Loss	4q_Gain_1	4q_Gain_1	4p_Gain_1	7p_Loss	7p_Loss
7	4q_Gain_1	19q_Gain	6q_Loss_2	4q_Loss	7p_Loss	7p_Loss	8p_Loss	6q_Loss_2	6p_Loss	6p_Loss	4q_Gain_1	18p_Loss	8p_Loss
8	5p_Gain_2		7p_Gain	5q_Loss	8p_Loss	8p_Loss	8q_Gain	7p_Loss	6q_Loss_2	6q_Loss_2	5p_Gain_1	18q_Gain	8q_Gain
9	6p_Loss		8p_Loss	6q_Loss_2	8q_Gain	8q_Gain	9q_Gain	8p_Loss	7p_Gain	7p_Gain	6p_Loss	10p_Loss	10p_Loss
10	8q_Gain		8q_Gain	7p_Gain	9q_Gain	10p_Loss	10p_Loss	8q_Gain	7p_Loss	7p_Loss	6q_Loss_2	16q_Loss	16q_Loss_1
11	9p_Gain		10p_Loss	Whole7_a	10p_Loss	15p_Loss	12q_Gain	10p_Loss	8p_Loss	8p_Loss	8p_Loss	7p_Gain	
12	9q_Loss		10q_Loss	8p_Loss	15p_Loss	16q_Loss	16q_Loss	10q_Loss	8q_Gain	8q_Gain	7p_Loss		
13	10q_gain		11p_Gain	8q_Gain	16q_Loss	17q_Gain		12p_Gain	10q_Loss	10p_Loss	8p_Loss		
14	11p_Loss_1		11p_Gain	10p_Loss	21p_Gain	19q_Gain_1		12q_Gain	11q_Gain	10q_Loss	8q_Gain		
15	11_Gain_2		11q_Loss	10q_Loss		21p_Gain		13q_Loss	12p_Loss	11q_Gain	9q_Gain		
16	12q_Gain_1		12q_Gain	11p_Gain_1		ChrX_Gain		15p_Loss	12p_Loss	12p_Loss	10q_Loss		
17	14p_Gain		13q_Loss	11p_Gain_2				16p_Gain	12p_Gain	12p_Loss	10q_gain		
18	19p_Loss		15p_Loss	11q_Loss				16p_Gain	12q_Gain	12p_Gain	11p_Loss_1		
19	21p_Gain		16p_Gain	12q_Gain_2				16q_Loss	13p_Loss	12q_Gain	11p_Loss_2		
20	22p_Loss		16p_Gain	13q_Loss				17q_Gain	13q_Loss	13q_Loss	11q_Gain_1		
21			16q_Loss	15p_Loss				19p_Gain	13q_Gain	15p_Loss	12p_Loss_1		
22			17n_Gain	16n_Gain_1				19n_Gain	15n_Loss	16n_Gain	17n_Loss_2		

Regarding the specific example offered by the reviewer (“6q_loss_2”) we agree there are a small number of prolific changes that are present across a broad range of clones, the most striking of which is actually 16q_loss_1. Interestingly this is present in all tumour clones with the exception of the “low grade” ISUP Grade Group 1 clones in H1_2 (Ext Data Fig 5d). We wish to emphasise again that spatial proximity is an important component of our clonal selection algorithm. It is for this reason that the clone tree in Figure 2b has dotted lines separating the components of the clone tree that are overtly spatially disparate: H1_4, H1_5, H2_5 separate from H2_1 and from H1_2). Though unlikely where multiple common features exist, it seems possible for one or two mutational events to occur sporadically in different parts of the prostate. We are not sufficiently confident, at this stage, to advocate intra-prostatic seeding/metastases. We do believe, however, that this is an exciting future avenue of research.

- The methods section does not list a specific algorithm for tree building, which leads me to believe they were constructed manually. To sort out the discrepancies, it would be imperative to algorithmically build trees. Given the assumptions of shared CNVs likely denoting shared ancestry and CNVs generally not being lost once acquired, a maximum parsimony framework seems appropriate. If the data supports more than one phylogeny, it would be good to provide alternative solutions.

We apologize for the lack of clarity. Although performed manually, this approach is algorithmic in the sense that we follow a set of pre-defined criteria (please see methods subsections; Manual, Algorithmic Tree Building from Pre-Processed inferCNV Data) when constructing the trees to ensure consistency and reproducibility, as described in the above response. We have made further edits to make this much more clear.

Additionally, using the R package phangorn (<https://github.com/KlausVigo/phangorn>), we have performed a computational phylogenetic analysis of the clones in Figure 3, using inferCNV Hidden Markov Model clone-level inferences (n = 3324 genes) as the input to computationally generate a

tree from iCNV gene level data. We present the following Maximum Parsimony Reconstruction as requested by the reviewer.

The tree agrees with our manual-algorithmic approach (as observed in Figure 3, notably confirming that altered benign clone C is more similar to tumor clones E, F and G than benign clone A).

- Within the phylogenies, it would be good to annotate branches with the CNVs that occurred on them.

We thank the reviewer for their suggestion. In early iterations of these panels, we did actually include this information, displayed in the clones, branches and common ancestors (please see the images below).

However, after seeking advice, we realized that this made the trees hard to interpret. We opted for the simplified versions currently in the figures, however on prompting by this reviewer we now include revised clone specific tables as Supplementary Tables 1 and 2.

In summary, we recognise the reviewers overarching concerns in this comment and hope that we have addressed each of the details of these concerns in turn. Specifically, with the novel emphasis, made possible by our approach, on spatial identification of clones as related groups of cells (as opposed to clones being simply related mutations, an approach commonly taken in bulk-sequenced studies) we believe the systematic manual approach we describe delivers the most informative presentation of clonal lineage.

6. To see CNVs observed in different clones are due to shared ancestry rather than multiple, independent acquisitions of the same CNV, could the authors assess whether it is the same allele that is amplified/lost across all clones with those CNVs from SNPs in the transcriptomic data? In case there is a CNV violating the topology of the phylogeny, this might be proof of an independent acquisition.

Thank you for this interesting question. We have now undertaken an allele-specific SNV analysis and are pleased to report that this does add granular detail as hypothesised by the reviewer.

We have made the assumption that spots bearing the same collection of CNVs are more likely to be clonally or ancestrally related than to have acquired the same combination of CNVs by chance. But we acknowledge that single common CNVs could have arisen sporadically. We have therefore conducted additional single-nucleotide variant (SNV) analyses as recommended by the reviewer. It is important to note that the Visium RNAseq read lengths are limited to approximately 150-200bp at the 3' end of each gene and we are therefore limited to identification of SNVs in a region of RNA corresponding to a short length DNA.

We analyzed the Visium library for the clones in section H2_1, using `cb_sniffer` (https://github.com/sridnona/cb_sniffer) as published by Petti et al., 2019 (doi.org/10.1038/s41467-019-11591-1). `Cb_sniffer` was designed for use with 10x Genomics Chromium single-cell RNA sequencing on 3' libraries, sequenced at a depth of 200,000 reads per cell. As we've noted, Visium spots comprise 5-15 cells, and the libraries in section H2_1 were sequenced at a depth of 50,000 reads/spot.

We identified all variants, within any gene with an inferCNV Hidden-Markov Model-predicted alteration (5.4 million Variants, from $n = 3,324$ genes, with an iCNV in any clone), and called these variants using `cb_sniffer`. This output a total of $n = 13,447,918$ reads mapping the SNV loci, which corresponded to $n = 573,781$ unique candidate snv loci detected in any spot. Of these, $n = 51,945$ SNVs had "complete data" (at least 1 read in 1 clone spot for each clone, and we calculated clonal variant allele fractions (clonal VAF) for each variant, within each clone. We focused on genes altered in Chromosome 8, given that we could identify a clone with no predicted iCNV alteration (A), and shared CNVs in the other clones (8q amp). Filtering for candidate variants with at least 10 reads per clone, resulted in a dataset of $n = 10,529$ SNVs. Further filtering for candidate SNV loci in Chr8 resulted in $n = 925$ candidates. Of these 96.2% ($n = 890$) had no differences in clonalVAF values between clones A and B (eg, all detected reads were both ref or alt allele for all clones). Given the sparse data, it is difficult to globally characterize, however, we highlight two examples that are relevant to the reviewer's question: chr8 (143580183 and 99892049).

We identified a candidate SNV locus from the gene `EEF1D` (Chr8: 143580183) with high quality data (reads covering the SNV loci were detected in >90% of spots). Gene `EEF1D` was predicted to be

diploid by iCNV in benign clone A, whereas clones B-G were all detected to have an amplification. Clone A had a clonal VAF of 0.52, suggesting an even distribution of transcripts expressing both the alt and ref alleles. Clones B-G also had >90% spots having a detection of the candidate SNV locus, but had clonal VAF values of 0.27-0.36. This suggests at least single copy amplification in EEF1D that had the ref allele, resulting in a 2(ref):1(alt) ratio of transcripts. We include here a table summarizing the data for the reviewers:

Gene EEF1D, chr8:143580183						
Clone	Ref_count	Alt_count	Total_CB	ClonalVAF	SpotPercentage	iCNV
A	463	501	964	0.52	90.2	Diploid
B	201	115	316	0.36	94.6	AMP
C	1668	931	2599	0.36	99.1	AMP
D	210	97	307	0.32	94.0	AMP
E	1217	722	1939	0.37	99.6	AMP
F	1149	426	1575	0.27	100.0	AMP
G	1066	462	1528	0.3	97.4	AMP

We also identified another candidate SNV locus found in COX6C (chr8: 99892049) which diploid in clone A, but with predicted amplifications in clones B-G. In contrast to the previous example, the clonal VAF values increased to 0.63-0.72, suggesting at least a single copy gain of the gene containing the alt allele.

Gene COX6C, chr8:99892049						
Clone	Ref_count	Alt_count	Total_CB	ClonalVAF	SpotPercentage	iCNV
A	234	270	504	0.54	79.56	Diploid
B	75	126	201	0.63	90.91	AMP
C	1009	1885	2894	0.65	100	AMP
D	73	125	198	0.63	76.12	AMP
E	612	1489	2101	0.71	96.46	AMP
F	313	988	1301	0.76	100	AMP
G	436	1138	1574	0.72	95.22	AMP

We additionally performed a principal component analysis of the clonalVAF values for all clones A-G: clones B and D segregated separately, from clones A, C, E, F and G. Next, when subsetting the clonal VAF data for clones A, E, F and G alone, resulted in Clone A grouping separately from Clones C, E, F and G.

We have added the following additional sentences on page 7, lines 16-20, as well as additional text in the methods (page 32, lines 21-23 and page 33, lines 1-9) regarding these SNV analyses.

“In recognition that other somatic mutations could add value in discriminating clonal groupings, we undertook an analysis of transcribed (exonic) single-nucleotide variants (SNV) using `cb_sniffer`²¹. Analyses of the ratios of clonal variant allele fractions of both specific events with high coverage SNVs (exemplified by chr8:143580183 & 99892049) [Extended Data Fig. 7] support shared ancestry [Figure 3b].”

In conclusion, while SNVs derived from spatial transcriptomics data are inherently sparse, the results support our phylogenetic clone trees.

7. In Fig. 4 c/d, there seem to be many CNVs in stromal cells, some of which seem to be share with the squamous cell cancer, such as loss of 8q (in clones A, B and D), gain of 1q (A and B) and loss of 19q (A, D, possibly B). Intuitively, given that

all these cells seem to share some CNVs, it could indicate all derive from a single cell that carried those CNVs. However, this seems to suggest a rather large clonal expansion traversing cell types. Is it likely these represent artefactual calls?

We agree with this interesting observation in skin and have undertaken further FISH analysis to exclude the possibility of artefactual calls.

We obtained gene-level, inferred CNV profiles as obtained by Hidden-Markov Models from inferCNV, for each clone A, B, C and D from the SCC sample in Figures 4 c/d. From these data, we identified a predicted deletion in EHD2 (Chr 19q) in all 4 clones, a predicted deletion in MYC (chromosome 8q) in clones A, B, and D, and a predicted amplification in CKS1B (chr 1q) in clones A, B, and C. These have been added as full panels to Extended Data Figure 12 (EHD2), and the other two have been added to Mendeley.

We performed DNA FISH against these genes. We identified that iCNV correctly predicted ground truth CNV status in 91% (n = 11/12) regions. FISH results table as follows:

	CKS1B iCNV and FISH match	MYC: iCNV and FISH match	EHD2 iCNV and FISH match
Clone A	Both Amp	Both Del	Both Del
Clone B	Both Amp	Both Del	Both Del
Clone C	Both Amp	Both Diploid	Both Del*
Clone D	iCNV Diploid, FISH amp	Both Del	Both Del*

*: Both Diploid and EHD2 deleted cells were detected by FISH

In conclusion, we do not believe these to be due to artefactual calls in the iCNV data. However, we recognize that there may be some uncertainty in clones B and D (labeled as stroma). The histology in this specimen was annotated at multi-spot level: with the majority tissue histology being assigned to the clone label. The FISH results indicate that while the specific iCNVs are being detected in these clones, not all cells within the regions of interest are altered, and that admixed clonal lineages may be detected by our spot level analysis.

We have added the following sentences regarding the FISH validation to the main manuscript on page 10, lines 2-5, and added accompanying methods text regarding the details of the two new FISH probes.

“Additional validation of siCNV signals were confirmed by DNA FISH against 3 probes: chr1q gain (CKS1B), chr8q loss (MYC), and chr19q loss (EHD2) from consecutive sections of the SCC sample. We

identified that siCNV correctly predicted CNV status in 91% (n = 11/12) spatial clonal regions (Fig 4d, Extended Data Fig. 12, Mendeleev)”

Some minor points:

8. The introductory paragraph lacks references.

We apologise for omitting these and have now included the following citations in support of our assertion in the second sentence of the introduction: Milholland et al <https://www.nature.com/articles/ncomms15183>; and for the third sentence of the introduction: Grossman et al (<https://doi.org/10.1016/j.stem.2021.02.005>),

Chen et al (Cancer Res. 2002 Nov 15;62(22):6470-4) and

Alvarado et al (DOI: 10.1158/0008-5472.CAN-05-0399).

9. I believe using subheadings would increase the readability of the manuscript.

Thank you for this suggestion (also highlighted by reviewer 3 in comment 5). We agree with both reviewers and have therefore added appropriate headings through the manuscript.

10. Fig. 2a: Why do the cells assigned to clone J appear to have a CN profile much less noisy than cells belonging to other clones? Is the denoising specific to each clone? This seems to be the case for clone C in Fig. 4c as well

Denoising is not specific to each clone, we have run inferCNV with default noise parameters as should be clarified by our shared GitHub repo (please see details in response to reviewer 3, comment 14), and from the original inferCNV documentation (<https://github.com/broadinstitute/inferCNV/wiki/De-noising-Filters>). We ran Seurat metrics to visualize the normalized counts and unique features for each clone. These results indicate that clone J has a much higher number of reads and unique features as compared to the other clones, indicating that increased read depth results in decreased noise in iCNV outputs. We would also like to note that Clone J is primarily located in one specific section (H2_5), a high grade cancer area on the lower right of the prostate. Due to higher cell density of high grade cancer we commonly see higher UMI counts with Visium in these areas. This specific section performed very well experimentally resulting in decreased noise after normalization.

11. On page 5, line 12: 16q is listed to be both gained and lost, which I think is an error.

We thank the reviewer. The lines read as follows: “Using this approach, we observed a common ancestral clone (clone H, Fig. 2b) containing truncal events including CN loss on chr 6q and 16q, and CN gain on 12q and 16q”. We have reproduced below an image from Fig. 2b, with annotations highlighting the events in question in Clone H. Thus, we do observe a gain and loss on two separate 16q regions and can confirm that this is not an error.

12. I cannot find any clinical information on the patients, such as the age of the men whose prostates were sampled. Are the large clones in prostate due to advanced age of the men, at which point such benign clonal expansions might be expected?

Prostate patient 1 was 82 years old, and prostate patient 2 was 63 years old. Both had reported Gleason Scores of 4+3 (ISUP Grade Group 3) at initial biopsy, and the prostatectomy pathology is as indicated in the paper (patient 1 = ISUP Grade Group 4; patient 2 = ISUP Grade Group 3). We have added these details to the manuscript materials and methods (page 21, lines 3-6).

Regarding uncontrolled clonal expansion in older men, we are not aware of this eventuating in prostate cancer; indeed in our experience, the appearances of both prostates are consistent with the multifocal heterogeneity typically seen in men undergoing this surgery.

Thank you very much for your engagement with our manuscript and extremely constructive comments.

Referee #2 (Remarks to the Author):

The broad goals of this study is to elucidate the development of cancer from benign tissue, improving our understanding of disease progression and enabling earlier diagnosis of cancer. Specifically, the authors have used spatial transcriptomics to

infer genome-wide copy-number variations (CNV) in situ. Using the 10X Genomics Visium system, the authors comprehensively analyzed a full width axial section of two prostate specimens in addition to several other tumor types. The authors convincingly demonstrate their ability to resolve in situ transcriptomic data that represents highly localized subclonal populations within the prostate. Using their spatially defined transcriptomic data, the authors infer somatic copy number alterations as others have done with single cell data. The authors then used these inferred CNAs (iCNAs) to construct detailed phylogenies of tumor development with profound spatial resolution.

One of the key findings of the manuscript is the discovery of 'benign' subpopulations of prostate cancer cells that nonetheless carry deleterious oncogenic iCNAs. Figure 3 demonstrates this finding for MYC amplification and PTEN loss in both 'benign' and tumor tissues. Additional strengths of this manuscript include the robust computational validation of iCNAs comparing with WGS and FISH data, and the demonstration of the complementary roles of transcriptomics, histopathology and iCNAs in resolving unique aspects of tumor heterogeneity. The use of multiple tumor types to support utility of the iCNA method in detecting alterations in benign tissue is also noteworthy. The study is well done and represents an important advance, although previous studies have documented oncogenic mutations in benign tissue. (E.g. PNAS 1996 doi: 10.1073/pnas.93.24.14025, Science 2015 doi: 10.1126/science.aaa6806, and Science 2019 doi: 10.1126/science.aaw0726.

Thank you to this reviewer for highlighting so effectively the key findings in our manuscript. We agree that the finding of “benign” clones with siCNVs normally presumed to be oncogenic is a key finding and, given this unexpected observation, our robust validation with both WGS and FISH is important. We would like to highlight for the reviewer that we have further validated these findings with additional FISH analysis of the skin samples (Figure 4 c/d) (please also see response to reviewer 1, comment 7), as well as attempted targeted scDNAseq from one of our prostate specimens. We acknowledge the reviewer’s assistance in pointing out previous studies documenting mutations in benign tissue. We have now cited these in the paper (page 5, line 2) and note that these former studies, mainly based on bulk-sequencing of skin, point to the need for higher resolution studies able to interrogate the clonal relationship between these benign regions and cancer, a need specifically flagged in Yizhak et al’s commentary (10.1126/science.aaw0726). We feel that our siCNV method has delivered on this need.

There are a few questions that need the attention of the authors as follows:

1. One of the most striking findings of this study is that copy number alterations are evident in “benign” tissue as defined by two pathologists. Utilization of clinical stage markers e.g. AMACR/HMW cytokeratin staining could be helpful in further validating the ‘benign’ nature of these regions.

We agree that it is critically important to validate the ‘benign’ nature of these regions. Indeed p63 and AMACR stains were used in this process and we have now included these images in the Mendeley data attached to our paper. We have also highlighted this by adding a sentence in our methods, section “Pathologist Workflow – Spot-level annotation for prostate patient 1”. While every effort to determine the malignant potential of all tissue spots was made, including examining close

sections with basal markers, spots which the pathologists could not confidently call or agree upon were excluded from the analysis.

2. The authors' speculation on page 7 line 21-23 that low-grade prostate cancer is fundamentally distinct from higher grade cancer, as opposed to being in an evolutionary continuum is overly strong considering that they present evidence from only two patients at a single point in time. Additionally, a more nuanced discussion of the translational relevance e.g. targeted treatment for subclones (p9 line 11-14), and contextualization in light of how this analysis might be implemented when most men undergo TURPs rather than open resection, are warranted.

The finding that low grade cancer in our study lacks most of the siCNV events is very revealing but we accept that the global application of these findings will need further validation in greater numbers. We also accept that the speculation that this explains the lack of progression of low grade cancer compared to higher grade disease would be strengthened by future studies sampling clones repeatedly over time. Nonetheless, we do believe there is precedent for drawing chronological implications from snapshot studies (e.g. DOI: [10.1038/nature14347](https://doi.org/10.1038/nature14347); <http://doi.org/10.1038/ng.3221>; DOI: [10.1038/ncomms7605](https://doi.org/10.1038/ncomms7605)) but accept the linear sampling and long-term clinical outcomes are needed to provide definitive proof on this.

In terms of the translational relevance of these findings, we have nuanced our discussion as requested by the reviewer including the following statement on page 11, lines 4-6: "Such targeted approaches could include a more intelligent rationale for focal therapy or, for systemic therapy, could be particularly valuable if such clones could be identified by 'liquid biopsy'". We have also made a small adjustment to our concluding comment. Most men who are treated surgically for prostate cancer undergo radical prostatectomy, although, as the reviewer points out, TURP, HoLEP or one of the other bladder outflow modalities are more commonly used to treat benign enlargement of the prostate. These operations are often useful sources of benign material but can be less reliable in sampling malignant tissue. We feel that our findings do have profound implications for any partial gland treatment (e.g. HIFU, VTP, electroporation) as we have demonstrated that treating the histologically transformed 'high grade' lesion alone could miss areas of 'genotypic transformation' in histological 'benign' areas of the prostate.

Thank you to the reviewer for prompting these important additions.

Referee #3 (Remarks to the Author):

I have reviewed the manuscript presented by Erickson and colleagues in which they present the application of inferred CNV profiles at 55um spatial resolution in various normal, benign and tumorous tissues. The authors present results of an algorithm which infers the CNV profiles from ST/Visium data, which is demonstrated on a synthetic in silico dataset,

demonstrate its application to decipher heterogeneity in nearly a full planar section of a resected prostate, followed by demonstrating it on a number of other tissues.

The application of spatial resolved transcriptomics (SRT) to decipher spatial genomic copy numbers accurately is exciting and much needed in the field. The results presented in the manuscript suggest that the authors can indeed do this.

However, I feel that the focus of the paper lacks clarity and this in turns questions the novelty of tis paper. On the one hand, the paper reads like a methods paper, but the details of the computational methods are lacking so it becomes difficult to compare their approach to e.g. Kueckelhaus et al 2020 (<https://doi.org/10.1101/2020.10.20.346544>) who have also inferred CNVs from SRT data. The authors also do not compare their tool to a similar published tool called STARCH (<https://doi.org/10.1088/1478-3975/abbe99>). On the other hand, they describe the clonal CNV heterogeneity in multiple tumour types (including an impressive full slice of prostate tissue), however do not go into details of the biology nor extensively validate their ST/Visium findings using non-spatial single cell sequencing nor provide extensive comparisons between the inferred CNV and transcriptional landscape.

Never-the-less, the paper is a good start to what could be a very good piece of work. I hope that with the following points I can describe some of my specific concerns.

Thank you for positive comments and helpful critique of our paper. We have undertaken a number of further analyses to develop the paper as suggested here and hope that the reviewer agrees with us that these additions build on the 'good start' described above. We also want to apologise to this reviewer for not ensuring access to the computational methods of our manuscript. These were included in a GitHub page (please see details in response to comment 14) as well as a linked Mendeley resource which we provided in our cover letter alongside our submission but we believe that it was not possible for this reviewer to access these resources. Please accept our apologies for this.

Original reviewer comments are in black with our responses in blue. Where we mention pages & lines we are specifically referencing the tracked changes version of the manuscript.

Major

[1] Focus of the paper. Is this a methods paper, or a biology paper? The title and abstract point to the biology of the prostate cross section, but the start and end of the paper focus on the methodological aspect. Given that the concept of calling CNVs from ST/Visium data has already been described and demonstrated by others, I would recommend focusing more on the biological results.

Thank you to the reviewer for highlighting the biologically interesting aspects of our paper and for this critical feedback regarding the presentation of the manuscript in terms of the balance between biology and methodology. Although we have left the title unchanged, we have made substantial additions to the manuscript in both results (page 7, lines 21-22, and page 8, lines 1 - 13) and discussion (page 12, lines 3-12) to further emphasize the biological results as suggested by the

referee. This was also highlighted by Reviewer 1 and we have undertaken further biological analyses as outlined in response to Reviewer 1 (Comments 2, 4, 6, 7) and below (comment 4.1 & 16).

[2] Novelty of inferring CNVs. Inference of spatial CNV profiles has already been described by Kueckelhaus et al 2020 (<https://doi.org/10.1101/2020.10.20.346544>) where they used inferCNV to infer CNVs, Lu et al 2021 (<https://doi.org/10.1186/s13058-021-01451-6>) where they use LCM and bulk WGS+RNAseq. I believe that the only published computational tool that infers CNVs from ST/Visium data is STARCH, presented by Elyanow and colleagues (<https://doi.org/10.1088/1478-3975/abbe99>) in 2021 with initial commits to their GitHub repo made in May 2020. Given the concept of applying scRNAseq CNV inferences tools to ST data has already been described and that a method for inferring CNVs from ST data exists, I believe that the authors should try to go to the same lengths as Elyanow and colleagues in demonstration and benchmarking of their computational approach. Elyanow also present pretty much the same 1D-to-2D synthetic in silico tissue benchmark. It would suggest that the comparison include performance of inferCNV as described by Kueckelhaus et al, similarly using CopyKat (Gao et al <https://doi.org/10.1038/s41587-020-00795-2>), and also perhaps considering also the simulated spatial data from scRNAseq as demonstrated in STARCH.

Thank you for flagging these relevant papers. We agree that the in silico / synthetic approach used by Elyanow et al (Phys Biol, 2021) is a helpful way of validating spatial inferred CNVs and comparing approaches. Our Extended Data Figure 2 includes a similar validation of our siCNV approach acting as an in silico sanity check or 'benchmark' as the reviewer suggests. We want to emphasise that, similar to the preprinted work from Kueckelhaus et al, we have taken the single-cell inferCNV tool (Patel et al) and applied it to spatial data to derive our siCNV approach (see Extended Data fig 2, page 7, line 29 through page 8, line 4); and response to comment 14 below). In summary, while we have developed a novel algorithm to cluster inferCNV outputs and permit spatial visualisation of the resultant clones, we have not developed a new CNV inference method. Given that this is the case, we did not consider it necessary to re-benchmark inferCNV. However, in our selection of this tool we did look at other methods, including STARCH (Elyanow et al) and CopyKate (Gao et al) before selection of inferCNV. This included running STARCH alongside inferCNV during our in-silico work-up, but not CopyKat which requires human gene names/ENSMBLID- annotated matrices for input while our synthetic data approach uses an artificial genome.

As shown, STARCH was only able to identify one of the two synthesised clones. By contrast, inferCNV faithfully recapitulated both 'ground truth' clones (Extended Data Fig. 2b,d,e).

We also ran CopyKat and STARCH on section H2_1 (from Figure 3) and the results broadly corroborate our siCNV findings:

Having selected inferCNV, we have gone to quite some length to validate the siCNV findings, benchmarking against *in situ* visualisation of copy number (FISH) as well as DNA sequencing (WGS), and with the further approaches recommended by the reviewer in their next point.

[3] Validation of the inferred CNV profiles. While the authors do indeed validate the MYC and PTEN CNAs via FISH, and validate the CNV calling on matched scRNAseq+scDNAseq data, they do not extensively validate their results from spatial data. I believe that validation via single cell DNA sequencing of at least one heterogeneous section of the prostate tissue should be provided. If this is not possible, then the authors should try to source a similarly heterogeneous piece of tissue to profile using Visium and scDNAseq.

We identified two commercially available single cell DNaseq technologies: 10x Chromium Genome and Exome from 10x Genomics and Tapestry from MissionBio. 10x Genomics, however, have discontinued selling the 10x Chromium Genome and Exome reagents as of June 2020 (<https://www.10xgenomics.com/products/linked-reads>). We identified two samples with publicly available data, and enquired if 10x Genomics would have the specimens retained for spatial transcriptomics experiments: however, while the fresh frozen samples were retained, the QC of the specimens indicated that they were not suitable for Visium due to degraded RNA (https://pages.10xgenomics.com/rs/446-PBO-704/images/10x_AN026_SCCNV_Assessing_Tumor%20Heterogeneity_digital.pdf, Figure 3) and poor morphology.

We then turned to the single cell/nuclei Tapestri platform from MissionBio (<https://missionbio.com/products/platform/>). Given that most studies applying this technology have analyzed dissociated cells (specifically from hematological malignancies) we explored multiple ways to extract nuclei from frozen tissue sections. We first tried to work with neighbouring sections from prostate patient 1, section H2_1 (featured in Figure 3) however, unfortunately, these were cut-through to the point where no relevant clonal heterogeneity remained. As suggested by the reviewer, we therefore selected another suitable heterogenous section from prostate patient 1: section H1_4, featured in Figure 2. We assayed this using the MissionBio Tumor Hotspot Panel (THP) using extracted nuclei, which harbors 234 amplicons, ranging in size from approx 175-275 bp, covering 59 tumor genes. After running the default Tapestri Pipeline on the resultant FASTQ files, 13.07% of DNA read pairs were assigned to cells, resulting in a data set of n= 5,436 cells. Next, we performed quality control on the amplicons: after quality controlling for low performing amplicons, amplicons with low uniformity, amplicons with $> 2 \times$ standard deviation Gini score, and amplicons with a median count less than 0, a total of n = 144 amplicons, remained from n = 39 genes (after filtering, range: 1-12 amplicons per gene: MissionBio recommends a minimum 15 genes per amplicon to robustly call CNV regions).

Tapestri has documentation for calling ploidy to identify copy number variation, but requires a set of known diploid cells to be set as a diploid reference. In consultation with MissionBio technical personnel, we were unable to identify any SNVs that could be used to distinguish a diploid population. We thus applied unsupervised learning techniques (method='graph-community', k=10) to the normalized count data from all cells, resulting in n = 69 clusters. These are broadly grouped into 2 large groups when visualized by UMAP (pictured below).

Upon visual inspection of the normalized count data for the groups, we were able to further merge these into n = 8 groups harboring similar normalized count profiles. Under the assumption that diploid cells would have the most genotypic homogeneity, we identified one large group, comprising n = 4,096 cells, to be a possible diploid population. As assessed visually by heatmap, two other groups harbored nearly identical normalized count profiles as the possible wild type group, but these groups had elevated counts in the THP_v2_EGFR_55221702 amplicon (n = 297 cells) and

THP_v2_ERBB3_56478798 amplicon (n = 223 cells) respectively, and were thus filtered as outliers based on elevated counts. Lastly, an outlier group of < 0.8% cells (n = 11) were filtered. This resulted in a final data set for analysis of 4 groups with a total of n = 809 cells.

Of the amplicon data, none of the genes of interest within Tumor Hotspot Panel (THP) met the 15 amplicon per gene criteria to robustly call CNVs. Nonetheless, we identified that PTEN (n = 5 amplicons) was called as diploid in all groups 1 - 4, which is consistent with our iCNV findings for this section H1_4 (Fig 2, Clones E-K), as well as PTEN FISH results (previously unreported) from section H1_4, which we now include below:

[Legend: Representative images taken from different regions of section H1_4 (red probe = PTEN; green probe = Chr10 centromere control). PTEN was diploid, across the different histologies from section H1_4]

In order to further address the reviewer's request to spatially validate our findings at DNA level, we also performed further DNA FISH experiments against additional targets in the squamous cell carcinoma specimen (Figure 4; please also see response to Reviewer 1, comment 7). We reproduce the summary table here:

	CKS1B iCNV and FISH match	MYC: iCNV and FISH match	EHD2 iCNV and FISH match
Clone A	Both Amp	Both Del	Both Del
Clone B	Both Amp	Both Del	Both Del
Clone C	Both Amp	Both Diploid	Both Del*
Clone D	iCNV Diploid, FISH amp	Both Del	Both Del*

*: Both Diploid and EHD2 deleted cells were detected by FISH

We now include additional text in the results (page 10, lines 2-5) as follows: “Additional validation of siCNV signals were confirmed by DNA FISH against 3 probes: chr1 gain (CKS1B), chr8 loss (MYC), and chr19 loss (EHD2) from consecutive sections of the SCC sample. We identified that siCNV correctly predicted CNV status in 91% (n = 11/12) spatial clonal regions (Fig 4d, Extended Data Fig. 12, Mendeley)”

In summary, we believe these several additional validations have been useful and have provided helpful additions to our manuscript. However, our work to address the reviewer's comment have further highlighted the shortcomings in applying the latest available single cell technologies, which inherently lose spatial tissue context through dissociation, to address questions of histological and genomic heterogeneity.

[4] Integration of scRNAseq.

[4.1] The authors seems to be missing in depth description of the consistency and differences of gene expression profiles with the inferred CNV patterns. One would assume that these should be similar, but the extent of this is not described.

Thank you for highlighting this and we agree that differential gene expression analyses between siCNV clones are missing. We have therefore undertaken a DEG analysis of CNV derived clones from section H2_1 (please see response to reviewer 1, comments 2 and 3). We have included a new figure on this (Extended Data Figure 8) and provided additional comment in the manuscript results (page 7, line 21-22 and page 8, line 1-13) and discussion (page 12, line 8-11).

Having established the clonal sub-groups in this heterogeneous section of prostate tissue, we used differential expression analysis to investigate potential functional alterations unique to these cellular groups. Focussing on Clone C, altered benign cells, we observed an upregulation of Myc activity (Extended Data Fig. 8, panel c) as well as pathways responsible for phenotypic versatility²² (Extended Data Fig 8, panel b) when compared to diploid benign cells (Clone A). Furthermore, there was a down-regulation of conventional androgen receptor (AR) target genes (e.g. KLK2, KLK3, FKBP5, NKX3-1) raising the hypothesis of a reduced (or altered) dependence on AR regulation in these cells²³. We also investigated the distinction between Clone C and clones containing histological transformed cells (Clones E-G). We saw reduced MSMB and increased GDF15 expression in both groups (Extended Data Fig 8, panel a, d), which are normally thought to be pathognomonic of malignant transformed cells^{20,24}. When analysing differentially expressed genes only found in altered benign cells, we observed an enrichment for genes associated with oxidative phosphorylation and mitochondrial energy metabolism as well as protein stabilisation (Supplementary table 5), consistent with cells trying to cope with extrinsic and intrinsic stress.”

[4.2] The “normal” tissue for the CNV inferences should ideally be the cell type that is the cell of origin of the tumour, however this basic analysis is ignored.

Thank you for this comment. We think the reviewer is specifically referring to Figures 2 and Figure 3. We used the consensus pathology defined, histologically benign spots, composed of copy-number neutral populations, as the reference sets for these analyses. We want to emphasise that the selection of benign references focuses exclusively on epithelial cells, to ensure cell type consistency. To make this clear, in line with the reviewer’s comment, we have edited the methods subsection “Selection of Benign References”, on page 27, line 15, to add the words “luminal epithelial” as follows: “We first performed 15 an unsupervised analysis of only the benign luminal epithelial

reference cells". For the SCC sample in Figure 4, we used patient matched, scRNAseq data from normal skin, as detailed in the first two sentences of section "SpatialInferCNV Parameters (Fig. 4)" on (page 30, lines 6-8).

[4.3] The authors do not consider that spots could contain a mixture of tumour and normal cells, e.g. TILs. This is a concept that a number of co-authors should be familiar with. It would be nice to see if the authors can also consider the tumour cell content of spots in their model using supervised decomposition methods (for samples with matched Visium and scRNAseq data) or de novo deconvolution methods (if such tools are accurate). Alternatively, the authors could try to segment DAPI stained slides and perhaps use this to model the number of cells within a spot to assist with deconvolution. At the very least this should be demonstrated on one of the heterogeneous prostate cancer slide H1_4.

We do not have access to, and are not aware of, any ground truth scRNAseq dataset with single cells annotated for tumor or benign from primary prostate cancer. The definition of tumor, in primary prostate cancer, is spatially defined based on morphology (<https://www.nature.com/articles/3800054>; <https://pubmed.ncbi.nlm.nih.gov/5948714/>), and the presence or absence of basal cells within glands (absence = tumor).

Karthaus et al. [Science, 2020; doi: 10.1126/science.aay0267] generated scRNAseq data from patients with primary prostate cancers. In their work, they attempted to distinguish benign cells from tumor, and were unable to do so using scRNAseq alone, and used inferCNV instead [Fig S16-18]. These results suggest that it is extremely hard to distinguish malignant primary prostate cells from non-malignant cells using expression from scRNAseq alone. These results are also highlighted in the Sankey Flow plots of GEF to Clone, and SeuratClusters to Clone, added in our response to reviewer 1, comment 3 and below to comment 16. We are not aware of any ground truth datasets which accurately distinguish tumor from benign, in primary prostate cancer, using supervised decomposition methods.

Nonetheless, we have found datasets [Henry et al; doi: 10.1016/j.celrep.2018.11.086], where the authors manually sorted cells within non-tumorous primary prostates for scRNAseq. The authors were able to successfully sort basal cells, luminal cells, and stromal cells. We therefore applied Stereoscope [Andersson et al; doi.org/10.1038/s42003-020-01247-y], a supervised decomposition method, using the input data from Henry et al., to section H1_4 (image below). We would like to note that approximately 80% of UMIs from Henry et al. mapped to annotations, and we have therefore refined the set to only those with a confirmed annotation. Here are the Stereoscope results, which we have also included in Mendeley.

Histology

Annotated

- Benign
- Chronic inflammation
- GG2
- GG4 Cribriform
- Stroma

Basal Epithelia

Club Epithelia

Endothelia

Fibroblast

Hillock Epithelia

Luminal Epithelia

Leukocyte

Neuroendocrine

Smooth Muscle

Luminal epithelial cell expression is primarily elevated in spots annotated by pathologists as Benign, GG2, and GG4 (all of which are luminal epithelial cell populations). Fibroblast and smooth muscle expression is primarily elevated in spots annotated as Stroma. Interestingly, neuroendocrine (NE) expression is elevated primarily in the GG4 regions, potentially suggesting potential neuro-endocrine differentiation. Basal cell expression is sparse/diffuse: basal cells line luminal epithelial glands, and loss of this layer is a key distinguishing factor in whether a prostate gland is benign or tumor: we do not observe significant basal cell expression. This is not entirely unexpected, given that as noted by the reviewer, Visium spots comprise a mixed population of cells, and basal cells are proportionately less present than luminal epithelial cells. Endothelial cells are typically observed in prostate histology as part of blood vessels, and there were no annotations of distinct blood vessels in our data. Leukocyte expression was detected across the tissue, whereas only $n = 8$ spots of “chronic inflammation” were detected. Finally, Hillock and Club cell expression was primarily detected in regions of benign and GG2, with some expression being detected in GG4.

Minor

[5] The manuscript would benefit from subheadings that separate out results sub sections and the discussion

Thank you for this suggestion, which was also identified by reviewer 1 in comment 9. We agree with both reviewers that subheadings would make it easier for the reader. We have therefore added appropriate headings through the manuscript.

[6] The use of the term “inferCNV” in the manuscript is inappropriate given that there is another method called inferCNV... even more so considering that the authors also refer specifically to this tool in their manuscript.

We apologise for our lack of clarity on this. As stated in our methods and main text (page 29, line 7 through page 30, line 4) we have taken the inferCNV code designed for scRNAseq and adapted this to the spatial context. We have a GitHub repository outlining the use of inferCNV as a key dependency (Patel et al.), but apologize that access was not ensured for the reviewer, and now have made the repository available (please see details in response to comment 14). We now call this siCNV (spatially-inferred CNV). To help make this as clear as possible we have now ensured that whenever “inferCNV” is stated the citation is included.

[7] CNV as a term is synonymous with polymorphic variants, but the term CNA (CN-alteration or CN-aberration) is more common for cancer genomics. Alternatively, the authors can use the term “somatic CNVs”.

This is interesting and we certainly see where the reviewer is coming from. We do agree that it is important to differentiate, where possible, between population polymorphisms and individual

somatic mutations. And that, therefore, the distinction between SNV/CNV and SNA/CNA can be helpful. We also note that this distinction is often ignored in the field. We are happy to take guidance on this from the reviewer / editorial team. We have left “inferCNV” unchanged in order not to confuse given that this is a published algorithm by that name.

[8] The choice of T for the GEF analysis was not rationalised. Was this an informed choice based on a factorisation metric such as silhouette or FN? Or was this based of other statistics such as AIC, BIC, or the “elbow method”?

The factor analysis uses Bayesian shrinkage to avoid overfitting the expression factors. Notably, when extraneous factors are included, their inferred baseline expression levels will be very low. Thus, extraneous factors do not worsen model fit but may make results less interpretable by, for example, introducing noise in visualizations. To accommodate for this fact, we initially overspecified the number of expression factors and then reran the analysis with the number of factors appropriate for our data. This approach avoids underfitting while maximizing the expressiveness and interpretability of the final model.

[9] Similar to point [8], the authors do not actually show that the GEF modelling is converging at 5000 iterations. It is very likely that it is, but this should be stated.

Convergence was assessed by tracking the loss (negative unnormalized log-posterior). Optimization was stopped when the loss had plateaued. This is exemplified by the loss and root mean square error (RMSE) plots below:

We have revised the manuscript methods accordingly (Page 23, Line 16-17).

[10] The colour pallet for clusters is hard to follow at times. E.g. in figure 2 and 3, the authors use a large variety of red colours for the various CNV clusters – these are very difficult to see in subsequent panels, e.g. I cannot tell the different between the colour for clone E and F in Figure 3d.

Thank you for flagging this. We have already wrestled with this in our first iterations and are keen to make the figures as accessible as possible. There is a tension here between our desire to use biologically meaningful colours (reds for cancer / severely altered clones, blue/greens for benign / indolent clones) versus the need to make colours easily distinguishable. To help with this we have now used a design-ready colour pallet from the cartography/mapping industry (www.colorbrewer2.org) and completely revised figures 2 and 3. We additionally have revised the colors in Extended Data Figure 5 to match the changes in revised Figure 2. We hope that the reviewer now finds the clones (and histology) more distinguishable. This was extremely helpful feedback.

[11] The results of the prostate tumour sample could implicate a poly-clonal seeding of tumors. Did the authors check to see if this patient carries a predisposition allele to prostate tumors or cancer?

Thank you for this suggestion. We have undertaken a bulk WGS of the two histologically benign sections each, from both prostate patients 1 and 2 in our manuscript to interrogate for germline predisposition. We sequenced the samples on an Illumina (NovaSeq) to a depth of 43X and 32X coverage respectively. The resultant FASTQ files were then processed using the Sarek pipeline [Garcia et al.; doi:10.12688/f1000research.16665.2]. The SNPs and small indels were called using GATK HaplotypeCaller with the GATK bundle for GRCh38 and annotated using snpEff. This resulted in identification of $n = 4,915,432$ variants in patient 1, and $n = 4,897,897$ variants in patient 2. We then compared the SNPs to lists of known prostate cancer predisposition alleles [Aly et al.; doi: 10.1016/j.eururo.2011.01.017, Sipkey et al; doi:10.1038/s41598-020-74172-z]. We noted that some of these risk variants are the reference variant and when we performed variant calling on WGS these were not called. Out of the called variants we identified 10 and 19 of the 36 Stockholm-1 risk alleles in patient 1 and patient 2, respectively with at least one copy of the risk allele. Using the list from Sipkey et al. where they selected SNPs that were associated with prostate cancer at a genome-wide significance level ($p < 5 \times 10^{-8}$) and had the effect size of $OR > 1.1$ for risk SNPs and $OR < 0.9$ for protective SNPs. Out of the 41 risk and 14 protective variants we found 11 risk variants and 6 protection variants in patient 1 out of which 3 risk and 1 protective variant were present on both

alleles. The corresponding numbers for patient 2 were 11 risk and 4 protective variants out of which 3 risk and 1 protective variant were present on both alleles.

We have added this as text to Mendeley, and those interested can be provided variant details upon a materials transfer agreement and request from the authors following GDPR guidelines.

[12] The authors could provide more insights into application of their method to their own large datasets, e.g. <https://doi.org/10.1038/s41551-020-0578-x>.

As mentioned in previous comments, we have not developed a new CNV inference method, but a novel approach to derive spatial cell groups from inferCNV outputs. We also would like to note that the spatial transcriptomics data for the reference highlighted by the reviewer, was generated by the previous “1k spot array” ST technology (He et al; doi.org/10.1038/s41551-020-0578-x). We used data from this technology to globally profile CNVs, at organ scale, in Figure 1 to highlight spatial distribution of CNVs. But, as highlighted in the manuscript (page 4, lines 11-12), and the Extended Data figure 2 d/e, we noted a significant difference of the 1k array’s ability to spatially resolve clonal events and thus moved to using Visium ST for the bulk of our analysis.

However, at the reviewer’s suggestion, we have analyzed three specimens from one patient from this study, patient BC23209, and provide both the iCNV and global CNV events.

[13] Can authors say something about ERG on chr21 (seems amplified in clone C)?

We have checked the inferCNV gene level Hidden-Markov Model (HMM) predictions (17_HMM_predHMMi6.hmm_mode-samples.pred_cnv_genes.dat) as requested. There were reads covering ERG detected in Clone C. While the HMM for Clone C predicted an amplification spanning

large parts of Chr 21, as it was absent in the HMM table output, ERG itself was specifically not inferred to be amplified by HMM. We have produced a table showing a modified version of the relevant region for Clone C, from the table, the genomic information from ERG inserted in between the nearest Genes with HMM predicted amplifications on Chromosome 21.

CNV State	Gene Name	ENSMBL ID	Chromosome	start	end
Amplified	ETS2	ENSG00000157557	chr21	37365573	37526358
Diploid	ERG	ENSG00000157554	chr21	38380027	38661780
Amplified	PSMG1	ENSG00000183527	chr21	38805183	38824955

Read counts from ERG were detected and included in the count matrices input into inferCNV.

[14] The method for inferring CNVs doesn't seem well described. Or am I mistaken and the authors only use the existing "inferCNV" tool to inferCNVs?

Our sincere apologies for not making the GitHub repository for our siCNV method more readily accessible. The reviewer is indeed correct in that we used the inferCNV tool to inferCNVs. We included a private link and password in our cover letter to the editor but appreciate this must not have been working for this reviewer. We invite the reviewer to visit: <https://github.com/aerickso/SpatialInferCNV> and ask them use the following details:

Username: forericksonetalsubmission1

Password: 14MloggingIntoGithub

As GitHub requires two-factor authentication, we provide the related gmail account information.

gmail: foericksonetalsubmission1@gmail.com

password: 14MloggingIntoGithub

[15] Github repo link doesn't work.

Please see above. Apologies again.

[16] GEF CNV profiles. In Ext Fig 3 the authors shows that the GEF harbour what looks like different CNV clones. However, I find this analysis lacking. It only compares the authors approach of GEFs and not more established methods such as clustering via Leiden/Louvain+SNN. Also, the authors could further look into sub-clustering of gene expression of e.g. GEF10 to see if these can indeed be separated at gene expression level. Likewise, the authors do not elaborate on how different GEF17 and GEF7 are w.r.t. transcriptional profile given their similarity at the CNV level.

We understand the reviewers desire for us to consider other approaches to gene expression clustering and have therefore performed clustering via Louvain+SNN using Seurat and STUtility on Visium data from prostate patient 1. After normalisation, dimensionality reduction was performed using principal component analysis and the expression-based clustering was performed with resolution parameter set to 0.8. A two-dimensional UMAP embedding was then constructed from the previously established top principal components. Using a Sankey Flow diagram we also highlight that it is not feasible to separate malignant prostate cells from non-malignant cells by solely analyzing gene expression levels.

Spatial transcriptome decomposition (STD) into gene expression factors (GEFs) is in principle similar to commonly used non-negative matrix factorization (NMF). Much like NMF, the output after decomposing the gene expression data is two matrices (gene x GEF and spot x GEF). GEFs serve as a lower-dimensional representation of the data but a single GEF will thus never fully represent a single spot. Since spots contain a mixture of cells, as previously highlighted by the reviewer, it becomes inherently difficult to properly assign an identity through clustering of gene expression, and identifying transcriptomic patterns better describes the underlying biological phenomena. In order to assess GEF to clone concordance each spot was assigned the GEF with the highest proportion.

We do wish to note that we intentionally moved away from STD as well as more established clustering via Leiden/Louvain+SNN to clone calling due to the lack of spatial conformation (as well as CNV clone differences). We reasoned that clonal 'clusters' needed to both display genomic homogeneity AND be situated nearby spatially. It was only when we moved to a CNV-based approach that we were able to achieve this.

Nonetheless, we have also now performed a sub-clustering using the dimensionally reduced factors data generated by spatial transcriptome decomposition. The clustering was performed with conventional Louvain+SNN, using only the spots previously annotated as GEF10. The resolution was set to 0.3 which generated 3 clusters, these along with corresponding sections are visualized below using umap. As seen in the Sankey Flow diagram below, despite sub-clustering of GEF10 improving GEF to clone concordance, GEF to clone heterogeneity remained. We have added a description of this result to Supplemental Figure 3 legend.

We additionally perform DEG analysis on GEFs 7 and 17. We identified $n = \text{XXX}$ genes, that were differentially expressed despite similarity in CNV profiles as noted by the reviewer. The DEG lists have been provided in Mendeley.

Line specific issues [Page 1, Line 2 = P1L2]

[P2L2] Page 2 Line 2 (P2L2). Should this rather be “polymorphisms” that are inherited rather than “alterations”?

Thank you. This has been changed as suggested.

[P2L3] How sure are we that somatic mutations are only in a small fraction of cells? Wouldn't a somatic mutation in the early stages of development be present in many cells? Uniparental disomy? Perhaps the wording should be re-evaluated by the authors.

Thank you for this helpful challenge. This is a fair point. We have reworded it to say instead: “...while post-developmental somatic mutations are usually only present in a small fraction of cells”

[P2L5] Somatic alterations can also happen in normal tissue.

We readily acknowledge this, indeed it's an essential finding of our study. This sentence currently reads: "In order to obtain spatial information of these rarer non-heritable genetic events occurring in cancer...". Our intention is not to suggest that these can only occur in cancer but that they have been commonly looked for in cancer using LCM and sc analyses. However, we recognise that the mention of cancer at this stage is probably redundant and have therefore removed it from this sentence.

[P2L6] Perhaps add some references?

We have added some representative examples of studies employing this approach.

[P2L11] Is spatial transcriptomics a "genome-wide" methodology? By extensions, I would find it hard to say that a gene-expression microarray is also "genome-wide" – genomes contain mainly more features than genes, including regulatory elements, repeats, centromeres, telomeres, etc.

We acknowledge that greater precision could be helpful here and have changed to "Genome-wide analysis of gene expression".

[P2L14] Missing space in between "infer" and "CNV"

Thank you. We have corrected this.

[P2L16] This is hard to follow. "inferCNV" is not actually introduced as a new method. Also see pints [P2L14] and [6].

As above we apologise for not making this clearer. We have reworded this sentence here as: "we sought corroboration that inferred CNV data (using inferCNV⁶) could mirror DNA-based phylogenies" to make this clear. We have a GitHub repository outlining the use of inferCNV as a key dependency (Patel et al.), but apologize that access was not ensured for the reviewer, and now have made the repository available (please see details in response to your comment 14).

[P2L18] The use of "successful" is questionable without statistical support. While the reconstructions seem OK, it is not perfect, e.g. Ext fig 1e has clone "E" in the wrong clade. Support with statistics such as RF score, Matching Split Distance, etc. The R package TreeDist (<https://github.com/ms609/TreeDist/>) implements a few useful metrics. This should also be compared to some of the other existing tools – see point [2].

We thank the reviewer for this comment. Please see our response Reviewer 1, Comment 5. We used the results in Extended Fig 1 to test automatically obtaining clone calls from iCNV outputs. We

identified, as this reviewer has noted, that clone calling from “raw” inferCNV has discrepancies from the published results. This is why we developed our own tree building approach (used in Figures 2 and 3) with inferCNV pre-processing to identify clusters. In accordance with this reviewer's comment, and in line with our response to Reviewer 1, Comment 5, we have revised “successfully recapitulate” to “attempted to recapitulate...”.

As noted in the Extended Fig 1 legend, we calculated entanglement for dendrograms in Ext Figure 1a, and found the entanglement to be 0.11. At the prompting of the reviewer we converted these dendrograms into phylogram objects using the dendextend R package, and used the TreeDist R package and calculated RF score (25.4), and the Matching Split Distance (70). We cannot run these metrics for Ext Fig 1 b/c, as the published trees were manually constructed, and we thus do not have access to digital phylogenetic tree structures/dendrograms to run the TreeDist functions.

[P3L5-6] The authors should mention that the samples was resected from a prostate tumour patient.

This information is available in the next sentence: “The specimen was obtained by open radical prostatectomy and an axial section was taken from the mid-gland”. We have added “...from a patient with prostate cancer...”

[P3L11] Erroneous space in between 21 and 000.

Thank you. Corrected.

[P3L14] “GEFs” is not a well established term, so perhaps a citation would be useful.

Thank you. We have added this (Berglund et al, DOI: [10.1038/s41467-018-04724-5](https://doi.org/10.1038/s41467-018-04724-5)).

[P3L19-20] The authors refer to these interesting regions in 3 different ways in the text. On P3L19-20 they are regions with “increased iCNV activity”, on P4L3 they are “seven key regions”, and on P4L20 they are described as “regions of interest”. This should be consistent.

Thank you. We have added “...of siCNV activity” to the two subsequent places in the text, flagged by the reviewer.

[P3L21] I believed the authors want to say that “iCNVs could be used to distinguish this regions,...”.

Thank you. We have added "...of siCNV activity" as stated above to ensure that the reader is clear that the sections / regions of interest are those highlighted in the previous sentence. In the sentence mentioned here, we are trying to make a concluding point at the end of the paragraph and so the words "...at organ scale..." are intended to set the context.

[P4L5] Erroneous space in between 30 and 000.

Thank you. Corrected.

[P4L7-8] Would it flow better if the authors describe the pathology annotation of spots when the Visium experiment is first described, e.g. at the end of P4L3.

Thank you for this suggestion. We have moved this sentence earlier as indicated.

[P4L8] Perhaps "investigating clonal relationships" could be more precisely termed as clonal evolution patterns?

Thank you. This now reads: "We then investigated clonal evolution patterns across the investigated tissue using iCNVs"

[P4L19-20] Perhaps this could be better illustrated using a Sankey Flow plot of GEFs to CNV clusters.

This is a good idea and we have now included this in Extended Data Fig. 3 and reproduce it here for the reviewer.

[P4L21] Do you mean a “phylogenetic tree”? Did the authors “construct” or “compute” this tree?

Thanks for querying this. We want to emphasise that this was a manual construction using our algorithm described in our methods as is often employed in these approaches (Cooper et al doi: [10.1038/ng.3221](https://doi.org/10.1038/ng.3221).; Wedge et al, doi: [10.1038/s41588-018-0086-z](https://doi.org/10.1038/s41588-018-0086-z); Woodcock et al, [10.1038/s41467-020-18843-5](https://doi.org/10.1038/s41467-020-18843-5)). We have amended the sentence as suggested “We constructed a phylogenetic tree to describe sequential clonal events...” Please also see response to Reviewer 1, Comment 5.

[P6L8-9] Can the authors relate these CNVs to oncogenes and tumour suppressor genes that have been described (e.g. PTEN and MYC).

Thanks for requesting clarity on this. We can confirm that the 8q24 region that is amplified in Clone C does indeed include Myc, and the 10p loss covers PTEN as well. This was the reason for selecting these two targets for validatory FISH analysis as outlined in the following paragraph of the main text. To make this clearer we have added at the point highlighted by the reviewer: “most notably in chr 8 and 10, which has been well-described in aggressive prostate cancer including oncogene *MYC* and tumour suppressor gene *PTEN*”^{12–14}

[P7L14] Should “low grade cancer” rather be “benign tumour”, or “low grade tumour”? In some models, such as the Vogelstein CRC progression model, it is only a late loss of TP53 that is the “carcinogenic” transformation from the adenoma (which would already be harbouring a loss of chr18 with SMAD2/4).

This is interesting. In prostate cancer we observe a spectrum of localised cancer from Gleason Grade Group (or ISUP group) 1 to 5 where we consider Grade Group 1 to be “low-risk” or “low-grade”,

Grade Group 2 and 3 to be “intermediate” and Grade Group 4 or 5 to be “high grade” or “high risk” (see EAU, AUA & NCCN guidelines; e.g. <https://uroweb.org/guideline/prostate-cancer/>). We were intrigued to note that the “Low grade” cancer here displayed a markedly different siCNV profile than the higher grade cancer. The mention of colorectal cancer in this setting is also interesting. Of course, CRC has traditionally been considered the archetypal two hit model of cancer progression. But then prostate cancer does seem far more complex than this and, indeed, this model has required many further more nuanced iterations even in prostate cancer (<https://www.nature.com/articles/s10038-021-00930-0>; DOI: [10.1016/j.cell.2017.01.018](https://doi.org/10.1016/j.cell.2017.01.018)).

[P10L5] Is this consistent with P7L14?

We feel it is really important to make the distinction between low grade cancer (Grade Group 1) which we interrogate in P7L14 and the ‘altered benign’ clonal group which is clonally related to and a precursor to more aggressive disease (Figure 3). Please note that low grade prostate cancer is an end-state in itself and does not progress to more aggressive disease (e.g. Ross et al, doi: [10.1073/pnas.0801318105](https://doi.org/10.1073/pnas.0801318105); also added to main text page 9, line 7). It is now increasingly considered almost to be a different disease. For perhaps the first time we are able to suggest why.

[P10L6-8] Haven’t there been studies of normal oesophageal tissue which show the same thing?

Yes, the reviewer is correct, a recent paper from Phil Jones’ group at the Sanger has shown this (doi: [10.1126/science.aau3879](https://doi.org/10.1126/science.aau3879)). We have cited this in our manuscript (page 5 line 2).

We wish to highlight for the reviewer that in the sentence concluding “....truly early events, occurring in tissue regions currently unknown to and therefore ignored by pathologists...” (previous P10L6-8) the word “unknown” refers to the fact that these regions are “unknown to...pathologists” because such regions are not captured by histology. We hope therefore that the reviewer is content to let this sentence stand.

Figures

[Fig1f] Is it worrying that the tumour associated GEFs are not distinct from the “normal” GEFs?

As outlined above (and Rev 1.3), we believe that our siCNV approach generates a more discrete partition of clonal groupings. We also agree that the lack of clear distinction of ‘tumour-associated’ GEFs from ‘normal’ GEFs is noteworthy and were initially puzzled by this. However, with our

subsequent findings that certain benign clonal groupings contained many of the CN features of cancer clones this is, perhaps, less surprising.

[Fig2a etc] Scale is not consistent with “CNV state” in Extd Fig2.

We thank for the author for their comment. We have revised Figures 2, 3, 4, and Extended Data Figures 3, 4, 5, 10 and 11 to include a box displaying “siCNV diploid”.

[Fig4g] missing prostate.

We agree that it would be useful to add prostate into this panel and have therefore included an additional “Venn” circle corresponding to Prostate Patient 1. An embedded picture of revised Figure 4 is also included in response to Reviewer 1, comment 7.

[Ext Fig1a] the dendrogram can be flipped to align better between iCNV and WGS-Ginkgo. E.g. the green and blue parts of the dendrogram for iCNV can be flipped with no change in interpretation.

We appreciate the reviewer’s comment. We plotted a tanglegram using the dendextend R package (<https://cran.r-project.org/web/packages/dendextend/vignettes/dendextend.html#tanglegram>). At the reviewer’s suggestion, we have modified the visual layout of the dendrograms, as follows and revised Extended Data Figure 1a, and have reproduced the revised subpanel as follows.

[Ext Fig2b] Why is the normal state set at "1" and not "2" for diploid? Does 0 indicate loss of 1 copy or 2 copy?

We appreciate the reviewer's comment. The current model is not diploid in character but rather just has a single strand of genes with different lengths and expressivity. Thus, the "CNV state" rather indicates how many copies of a gene that a cell/spot has or is presumed to have (when inferred). Here, 1 is the normal state (since 1 copy means the profile of said gene is unaltered), 0 means that there's a loss of the gene (you can only lose something once, hence why there are no lower states), and 2 and above means there's been a duplication of the gene with the value representing the total number of copies in the genome.

Regarding inferCNV, we would like to refer the reviewer to the documentation:

<https://github.com/broadinstitute/inferCNV/wiki/inferCNV-HMM-based-CNV-Prediction-Methods>.

In short, inferCNV's Hidden-Markov Model functions can either predict "3 states" (gain/loss/diploid), or "6 states". We also would like to note that STARCH also predicts a similar 3 state model (gain/loss/diploid). We interpreted any degree of amplification from inferCNV outputs to be a gain, or any degree of loss to be a loss. Further work, beyond the scope of this study, would need to be done to fully validate the ground truth degree of copy-number variation (LOH vs full deletions, or 1 copy vs 2+ copy gains) against the algorithm's predictive accuracy, which to our knowledge, has not been reported.

[Ext Fig2d,e] Perhaps using a seriation algorithm would make the 2 panels more comparable.

We appreciate the reviewer's comment. Given that inferCNV does not have native functionality to alter plots, we imported the denoised inferCNV matrix, processed them according to the following (<https://github.com/broadinstitute/infercnv/issues/206#issuecomment-823084179>), and plotted the following in the ComplexHeatmap package. We have included and compared the two approaches as suggested by the reviewer, and we are inclined to state that the seriation algorithm does not make the two panels appear to be more comparable. Nevertheless, we are open to change the panels if requested.

[Fig 3e] Low resolution. Does the centromere control of chr8 duplicate in clone C, but get lost in clone G? Is this expected? Figure 3A doesn't seem to imply that there should be centromere amplification in chr8.

We thank the reviewer for their attention to detail. Figure 3e, from Clone C, contains two cells, for which the cell boundaries have now been annotated. We also have added arrows to clone G, denoting the locations of the centromere controls: they are not lost in clone G. We reproduce the revised panels here, and have edited Figure 3 (and legends) accordingly.

Thank you to this reviewer for their extremely helpful comments and amazing attention to detail. Addressing these comments has enabled us to substantially improve our manuscript, both in the main text/figures and in the array of extended data and supplementary material that we can now make available. Thank you.

Reviewer Reports on the First Revision:

Referees' comments:

Referee #1 (Remarks to the Author):

I thank the authors for the revision, in particular for the added biological analyses and interpretation on expression differences and the further confirmation of CNVs in skin using FISH. However, they have not adequately addressed some of my concerns.

1. I thank the authors for their considerations and further analysis. While it is possible to identify SNVs in RNA data (often relying on having identified the somatic SNVs previously in DNA), somatic SNVs in the extremely limited regions covered by 10X sequencing cannot be adequately referred to as a 'landscape'. Furthermore, the novel SNV analysis performed here likely identified germline and not somatic SNVs (see response to point 6). Therefore, the data in the paper still far from warrants the use of 'landscape of clonal somatic mutations' in the title and I still request changing it to 'landscape of clonal copy number alterations' or equivalent. Of course, events involving copy-number neutral loss of heterozygosity also constitute important genomic events in the evolution of cancer but are not detectable using the inferCNV method. While this is a minor criticism and by no means needs to be incorporated in the title, it is important to note this limitation.

2. Thank you very much for this extended analysis. The added biological detail is good and adds to the narrative.

3. Thank you for the additional explanation.

4. I think the added elements in the discussion section are a good and thoughtful addition. In line with my response to points 1 and 6, I would be careful with using "somatic events described here" (1.3 p.12) and suggest changing it to more explicitly convey that these are exclusively copy number alterations.

5. The current manual approach for the construction of phylogenetic trees still lacks the necessary rigor and reproducibility. While I do not doubt the central biological observation of a benign clone carrying CNVs also found in the tumor, but the phylogenies require precision and robustness to be publishable.

Only a heading and some numberings were added to the Methods section, not any additional detail pertaining to the building of the phylogenies themselves. This is not sufficient to reproduce the trees.

- For example, the authors state in their response that 'spatial proximity is an important component of our clonal selection algorithm' leading to a dotted line in their phylogeny. This is vital for reproducibility, but this is not stated at all in the methods. Does this mean every spatial area gets their own phylogeny without the possibility of a connection between them?

- Far from the 'one or two mutational events to occur sporadically', clones C/D seem to share 8 CNVs with clones H/I/J/K at least according to Supplementary Table 1 (10q_Loss, 12q_Gain_2, 15p_Loss, 2q_Loss, 3p_Loss, 6q_loss_2, 8p_Loss, 8q_Gain), yet they are portrayed as completely independent. Likewise in Supplementary Table 2, it is difficult to reconcile clone D being completely unrelated to clone F, while they share 9 CNVs.

- The column naming in Supplementary Tables 1 and 2 is confusing. What is the difference between 'Second_Unobserved_Ancessor_to_Clone_K' and clone K itself?

Given the concordance the authors present in their response, a maximum parsimony approach from an algorithm that is actually reproducible, such as phangorn, should be used throughout the paper.

Another point coming from reviewing the phylogenies and images concerns clonal mixing. Given that a Visium spot contains multiple cells, can you be sure some of these smaller clones (such as H in Fig. 2 and B and D in Fig. 3) are not in actuality clusters of spots with cells from multiple different genomic clones? In essence, whether spots belonging to clone B simply contain a mixture of cells coming from e.g. A and C. I have read the response to reviewer 3 on mixed cell populations, but that only deals with transcriptomic heterogeneity, not with mixing of different genomic subclones. This should be addressed.

6. I commend the authors on the efforts of calling single nucleotide variants in their data and am pleased to note their two example base substitutions back up the inferred CNVs. However, one fatal flaw is that these SNVs are portrayed to be somatic, while they are most likely inherited. There are a few reasons for this:

- The method section does not detail an approach to filter out germline variants, so I suspect they are retained. In fact, it seems the method section states explicitly that only variant sites reported in the 1000Genomes project were used as a basis for this analysis, so these are likely SNP sites.
- The VAF of both these mutations approximates 0.5 in the diploid normal clones. It is very unlikely that clone A represents a true single clonal outgrowth. Whole-genome sequencing of laser-capture microdissections of prostate (200-500 cells) revealed that these small populations do contain clones but are not fully monoclonal (median VAFs between 0.15 and 0.3) [1]. Hence, it is much more likely that clone A represents a polyclonal population of diploid cells, which I think aligns with the authors' view on clone A.
- Both SNVs are situated at known common SNP sites (SNP ID rs1062391 for chr8:143580183 and rs1130474 for chr8:99892049), further signaling it is likely these SNVs were inherited rather than acquired post-zygotically.

These VAF of these substitutions confirms the CNVs at these loci, but there is no set of SNV lineage markers that serves as an orthogonal validation of the CNV phylogeny (e.g., a substitution present in clones C-G but absent in the others). I suggest retaining this analysis as it confirms the inferCNV calls and rewriting the section dealing in the text to reflect that these are not somatic mutations.

[1] Grossmann, S., Hooks, Y., Wilson, L., Moore, L., O'Neill, L., Martincorena, I., ... & Campbell, P. J. (2021). Development, maturation, and maintenance of human prostate inferred from somatic mutations. *Cell Stem Cell*, 28(7), 1262-1274.

Referee #2 (Remarks to the Author):

The authors have adequately addressed my concerns and have updated the manuscript accordingly.

Referee #3 (Remarks to the Author):

I have reviewed the revised manuscript by Erickson and colleagues.

My major comments we well met with good responses. While the paper itself is much improved and exciting, I would like to draw the authors to their GitHub page, which I was not able to review in the initial submission. I would now expect that top-tier analysis papers to come with very good GitHub repos that describe how to reproduce all analyses and figures, however, there are a number of issues with the quality of the current documentation that should be addressed. There are also a small number of minor issues that should be addressed.

Major points on the GitHub repo:

1) General organisation and expectations for a software repository

1a) The GitHub page should ideally be technically oriented around the software, e.g.

<https://github.com/aerickso/cvat>. Right now, the page emphasises the analysis of some sample data, which would be ideally part of a Jupyter notebook or ReadTheDocs site.

1b) An in-depth user guide could be provided via ReadTheDocs.

1c) Reanalysis of data should be provided through some literate programming document (Jupyter notebook, knitr, etc).

1d) By now I would expect that it is common practice to provide a notebook or scripts which shows how all figures were generated in the paper. This could be distinct from the software repo.

1e) The siCNV framework used in the study should probably be a tagged release cited in the manuscript (e.g. <https://docs.github.com/en/repositories/archiving-a-github-repository/referencing-and-citing-content>).

2) Some specific issues about the repository as a home for data analysis script that can be used to reproduce the results of the study. This might be

2a) The scripts should describe how all major results can be reproduced, and not just the siCNV part

2b) In the current README.md of the GitHub repo, some sections have no descriptive text. All sections should be accompanied by at least 1-2 sentences of text. Also, some figures are shown without context or captions (e.g. the first 4 images)

2c) There is no description of how to access the data from this study (or appropriate test data, should the authors want this to be a general user guide for their siCNV framework).

2d) There is no obvious way of finding out how the "purest benigns" or "Benign|GG2|PIN|GG4" were identified.

2e) There is no description or code block for the "clone tree building" section.

2g) The analysis scripts/notebooks used to reproduce results presented in the study should probably be a tagged release cited in the manuscript if they are hosted via another GitHub repository (e.g. <https://docs.github.com/en/repositories/archiving-a-github-repository/referencing-and-citing-content>).

Minor comments:

3) The author's response to my minor point 12 suggests that the siCNV clustering approach has limitations. The figure in the author's response to my previous point 12 shows the clustering of the siCNV identified 3 clones. On closer inspection I feel that there is something not quite right with the clustering, e.g. the CNV profile of the top part of the first cluster looks remarkably similar to the bottom half of the first – yet there two parts are in clearly separate clades. Compared to the siCNV clustering presenting the manuscript (which looks convincing), that reviewer only figure is worrying. Can the authors comment on why this happened?

4) A brand new paper describes how the slide-RNA-seq approach can be adapted to slide-DNA-seq (<https://www.nature.com/articles/s41586-021-04217-4>). The authors could consider citing this paper in their discussion.

Line specific comments from the tracked version of the resubmitted manuscript (406046_1_related_ms_3807532_r4km0w.pdf). Please note that some points are questions.

5) P1L6: there are still occurrences of using a white space as a thousand separator (e.g. P1L6 "120 000"). Please also check the rest of the manuscript.

- 6) P2L12: subheading should be bold.
- 7) P2L16: remove "_" in "infer_CNVs".
- 8) P2L18: add a comma between "modality generating"?
- 9) P3L2: "robustly could" or "could robustly"?
- 10) P4L3: should "iCNV" be "siCNV" or "CNV"?
- 11) P7L21: "clonal sub-groups" or "sub-clonal groups"?
- 12) P22L16: "Data processing" should be more specific. Perhaps "Spatial transcriptomics data processing"?
- 13) P24L16: link to GATK toolkit is incomplete – it should be <https://github.com/broadinstitute/gatk>, and not <https://github.com/broadinstitute>.
- 14) P26L18: change "InferCNV – Data Pre-processing" to "Data pre-processing for inferring spatial CNVs" (to avoid the InferCNV confusion).
- 15) P29LL5: I cannot easily interpret what is meant by "if a clone in a given section had ≤ 10 1k or Visium spots". Did you mean "< 1k [UMIs/genes] and <10 Visium Spots"?

Figures

- 16) Figure 1 panel a – still refers to iCNVs.
- 17) Figure 2 legend has "sCNV" instead of "siCNV".

Author Rebuttals to First Revision:

Editor comments:

Your manuscript entitled "The spatial landscape of clonal somatic mutations in benign and malignant tissue" has now been seen again by 3 referees. You will see from their comments below that while they find your work strengthened, some important points are still raised. We are interested in the possibility of publishing your study in Nature, but would like to consider your response to these concerns in the form of a revised manuscript before we make a final decision on publication.

We therefore invite you to revise and resubmit your manuscript, taking into account the points raised by the reviewers. Please highlight all changes in the manuscript text file. We would specifically point out the importance of a well-organised code repository, as per Reviewer #3 (ultimately, at publication, we would also ask you to put the code into a permanent repository and provide a doi in the manuscript, but Github is fine for now). We also ask that you re-think the title of the paper; not just with regards to the points made by reviewer #1, but also in terms of the over-used word 'landscape'. A starting point could be "Spatially resolved clonal copy number alterations in benign and malignant tissue", but do feel free to propose something else.

We hope to receive your revised paper within four weeks. If you cannot send it within this time, we will still be happy to reconsider your paper at a later date as long as nothing similar has been accepted for publication at Nature or published elsewhere in the meantime.

Thank you for this further feedback and the additional reviewers comments. As requested, we have particularly focussed on a substantially re-worked GitHub repository and have now provided step-by-step scripts for the construction of the novel analyses in the main figures as requested by Reviewer 3. We have also amended the title of the paper in line with your comments above (and also Reviewer 1) to: "*Spatial atlas of clonal copy number alterations in co-existing benign and malignant tissue*". We believe that because, alongside the biology, we are also providing a spot-by-spot map of several tissues that "atlas" is a more appropriate term than "landscape". We also wish to make the point that the analyses address neighbouring regions of benign and malignant tissue - hence "co-existing". An additional important piece of work requested by reviewer 1 is a further justification for our manual phylogenetic tree construction and a fuller explanation in the methods section to enable

reproducibility. The top line is that, in our experience, manual approaches are the norm in this area and we have provided many citations to support this. However, we accept that automated approaches do also bring value here and have extended our use of maximum parsimony networks/phangorn as requested by Reviewer 1 and have added an additional Extended Data Figure 14. Encouragingly the automated networks are supportive of our tree construction. Thank you again for considering our paper.

All references to page and lines are from the manuscript file containing tracked changes.

Referees' comments:

Referee #1 (Remarks to the Author):

I thank the authors for the revision, in particular for the added biological analyses and interpretation on expression differences and the further confirmation of CNVs in skin using FISH. However, they have not adequately addressed some of my concerns.

Thank you for your further comments which we have addressed in turn.

1. I thank the authors for their considerations and further analysis. While it is possible to identify SNVs in RNA data (often relying on having identified the somatic SNVs previously in DNA), somatic SNVs in the extremely limited regions covered by 10X sequencing cannot be adequately referred to as a 'landscape'. Furthermore, the novel SNV analysis performed here likely identified germline and not somatic SNVs (see response to point 6). Therefore, the data in the paper still far from warrants the use of 'landscape of clonal somatic mutations' in the title and I still request changing it to 'landscape of clonal copy number alterations' or equivalent. Of course, events involving copy-number neutral loss of heterozygosity also constitute important genomic events in the evolution of cancer but are not detectable using the inferCNV method. While this is a minor criticism and by no means needs to be incorporated in the title, it is important to note this limitation.

We again thank the reviewer. We have revised the title to *Spatial atlas of clonal copy number alterations in co-existing benign and malignant tissue*. As outlined in our comments to the editor above, we accept that landscape is perhaps an overstatement and have modified this to "atlas" and then replaced "clonal somatic mutations" to "clonal copy number alterations" as per your helpful suggestion. We also note your point about copy-neutral LOH and have acknowledged this limitation with changes on (page 7, line 15) as requested. Nonetheless we have gone some way to address this in our allele-specific SNV analysis of the data in Figure 3 (Extended Data 7) and wish to note that while we have not applied this approach to every siCNV event we have provided a strategy for interrogating allele specificity if required.

2. Thank you very much for this extended analysis. The added biological detail is good and adds to the narrative.

Thank you.

3. Thank you for the additional explanation.

Thank you.

4. I think the added elements in the discussion section are a good and thoughtful addition. In line with my response to points 1 and 6, I would be careful with using “somatic events described here” (1.3 p.12) and suggest changing it to more explicitly convey that these are exclusively copy number alterations.

We thank the reviewer for their suggestion. We have revised the use of this term in the title as requested and also in the text of the manuscript at the point indicated above and also in the legend for Figure 2.

5. The current manual approach for the construction of phylogenetic trees still lacks the necessary rigor and reproducibility. While I do not doubt the central biological observation of a benign clone carrying CNVs also found in the tumor, but the phylogenies require precision and robustness to be publishable.

Only a heading and some numberings were added to the Methods section, not any additional detail pertaining to the building of the phylogenies themselves. This is not sufficient to reproduce the trees.

We apologise for putting so much of our explanation to this point in the “response to reviewers” document last time but not implementing sufficient changes to the manuscript methods section itself. We have revised the manuscript methods to ensure that the trees are fully reproducible.

We have included the revised section here for clarity, and have included these changes in the manuscript on page 29, lines 7-20.

Clone Tree consensus siCNV event calling:

Both HMM siCNVs (from files: infercnv.17_HMM_predHMMi6.hmm_mode-samples.png and 17_HMM_predHMMi6.hmm_mode-samples.pred_cnv_regions.dat), and manual interpretation of denoised outputs (from the file: infercnv.21_denoised.png) were used to identify putative subclonal CNVs. These were then merged in a final consensus set, where events were listed for each clone for building clone trees (Supplementary Table 1, 2). Briefly, trees were constructed by identifying where CNVs were shared across clusters identified above as, under the assumption that a CNV cannot be reversed once it occurs, this indicates the cells in those clusters share a common ancestry. We therefore used this logic to identify ancestral relationships between clusters and build the clone trees. As our clone trees identify clones as related groups of cells (as opposed to clones being simply related mutations, an approach commonly taken in bulk-sequenced studies), where clones were present in subtrees that were not spatially proximate, we marked this uncertainty with dotted lines between common ancestors.

- For example, the authors state in their response that ‘spatial proximity is an important component of our clonal selection algorithm’ leading to a dotted line in their phylogeny. This is vital for reproducibility, but this is not stated at all in the methods. Does this mean every spatial area gets their own phylogeny without the possibility of a connection between them?

We have provided further clarity on this point in our methods page 29, lines 7-20. We wish to emphasise an important overarching principle here which is that we do not believe that we have sufficient evidence to advocate intra-prostatic seeding / intra-organ metastases in this study. Of course, this would be a very exciting phenomenon if present. We certainly considered this possibility when we observed many common events shared between the two main tumour sites on opposite sides of the prostate in Figure 2. However, the alternative explanation, which is probably more generally acceptable, is that this represents convergent evolution¹⁻³ with similar events (such as well known prostate cancer alterations in chromosome 8), arising sporadically in different places due to common selective pressures. We therefore chose to take the more cautious / conservative approach in our tree building. This means that we require spatial proximity, in line with our understanding of three-dimensional branching morphogenesis (Extended Data Figure 9) to call clonal lineage.

We believe that the trees reported are consistent with spatial clones visualised either on the ST 2D map or in an unobserved third dimension (tissue levels above/below the observed sections) with branching glandular structures giving rise to ancestral progeny through the branching, morphogenetic development of the prostate cancer glands⁴. In the absence of spatial proximity, placing all clones on the same phylogenetic tree, would imply direct seeding. While we recognize the high degree of interest there might be in any evidence for intraprostatic seeding and/or intra-organ metastasis, we do not believe we have sufficient evidence to draw such conclusions.

- Far from the ‘one or two mutational events to occur sporadically’, clones C/D seem to share 8 CNVs with clones H/I/J/K at least according to Supplementary Table 1 (10q_Loss, 12q_Gain_2, 15p_Loss, 2q_Loss, 3p_Loss, 6q_loss_2, 8p_Loss, 8q_Gain), yet they are portrayed as completely independent. Likewise in Supplementary Table 2, it is difficult to reconcile clone D being completely unrelated to clone F, while they share 9 CNVs.

For Figure 2, and distant clone relatedness, please see our response to the comment above. For Supplementary Table 2 and maximum parsimony please see our comment below.

- The column naming in Supplementary Tables 1 and 2 is confusing. What is the difference between 'Second_Unobserved_Ancestor_to_Clone_K' and clone K itself?

We apologise for the inconsistency in our naming convention which arose when converting the previous tables from the initial submission to the revisions. While the clone-specific columns were correct, our previous reporting of the changes occurring in each branch generated duplicates. We have now revised Supplementary Tables 1 and 2 to report only clone-specific, and unobserved common ancestor data. The common ancestors have now been renamed with the following convention: (Fig 2):

Common_Ancestor_1_Ancestor_to_A_B, Common_Ancestor_2_Ancestor_to_C_D, etc.

Given the concordance the authors present in their response, a maximum parsimony approach from an algorithm that is actually reproducible, such as phangorn, should be used throughout the paper.

We recognise this reviewer's preference for automated algorithms. As requested we have run a further Maximum Parsimony Reconstruction using phangorn on Figure 2, using gene level Hidden Markov Model outputs from clones F/G/H/I/J/K alone and the results (included below alongside Figure 2 and our new Extended Data Figure 14) do seem to recapitulate the tree topology for these clones from our manual annotation approach. We wish to highlight that, for the reasons outlined above, we have focussed on the large tumour focus on the right side of the prostate. We did also run a separate maximum parsimony reconstruction including C and D and these clones did associate with the I/J/K group. The most likely explanation for this are the CN losses in Chr 6 and 16 and the gain/loss in Chr 8. As explained above, this does not take into account the possibility that these mutations have arisen by convergent evolution¹⁻³, rather than a common ancestral lineage. We would like to note that the analysis implemented here through phangorn was not originally designed for tumor evolution but to resolve speciation discrimination⁵.

Manual reconstruction methods are not new for reporting cancer clonal phylogenies, indeed they are probably the most common approach in this setting and have been extensively used in the field⁶⁻¹⁰. We therefore suggest to stay with our original manual algorithm for tree construction as we believe it to be the most faithful and conservative method for representing the data.

Note, we also have included a new Extended Data Figure 14, which includes an additional validation of the manual clone trees by maximum parsimony reconstructions (from phangorn) as requested and have made changes to the manuscript with a new subheading titled "Maximum Parsimony Clone Trees" on page 30, lines 11-18.

Supplementary Figure X. Maximum parsimony reconstructions of prostate cancer clone trees. (a) Maximum parsimony tree for clones F-K from spatially proximate tumor bearing sections from sections H1_4, H1_5, and H2_5 from prostate cancer patient 1 (Figure 2). **(b)** Maximum parsimony tree for prostate cancer and benign epithelial clones A-G from from sections H2_1 from prostate cancer patient 1 (Figure 3). Input data to construct both trees were derived from gene-level siCNV hidden markov model data.

Another point coming from reviewing the phylogenies and images concerns clonal mixing. Given that a Visium spot contains multiple cells, can you be sure some of these smaller clones (such as H in Fig. 2 and B and D in Fig. 3) are not in actuality clusters of spots with cells from multiple different genomic clones? In essence, whether spots belonging to clone B simply contain a mixture of cells coming from e.g. A and C. I have read the response to reviewer 3 on mixed cell populations, but that only deals with transcriptomic heterogeneity, not with mixing of different genomic subclones. This should be addressed.

The reviewer is correct in that there could be mixed clonal populations. We are reporting CNVs within the limit of the resolution that we are able to resolve. We recognize this limitation and have added a statement in the subsection “Organ-wide clonal landscape in prostate cancer” on page 5, lines 15-17. As the reviewer notes we have already discussed the question of mixed cell populations and outlined our strategy for minimising transcriptomic heterogeneity. We await with interest the upcoming “subcellular” spatial transcriptomic

platforms (such as from 10x Genomics) which will deliver a further order of magnitude of resolution and provide intriguing answers to the question raised by the reviewer.

6. I commend the authors on the efforts of calling single nucleotide variants in their data and am pleased to note their two example base substitutions back up the inferred CNVs. However, one fatal flaw is that these SNVs are portrayed to be somatic, while they are most likely inherited. There are a few reasons for this:

We thank the reviewer for their commendation regarding SNV corroboration and for raising this point regarding germline variants. We have provided some more detail on this below and made some further changes to the manuscript in the light of this criticism.

- The method section does not detail an approach to filter out germline variants, so I suspect they are retained. In fact, it seems the method section states explicitly that only variant sites reported in the 1000Genomes project were used as a basis for this analysis, so these are likely SNP sites.

The reviewer is correct that the variant sites in the 1000 genomes project were used. We do not have access to matched bloods to perform true germline SNP profiling. We accept the criticism regarding a lack of clarity on somatic versus inherited SNVs and have removed the word “somatic” from relevant parts of the manuscript, particularly on page 7, line 15 and from title.

- The VAF of both these mutations approximates 0.5 in the diploid normal clones. It is very unlikely that clone A represents a true single clonal outgrowth. Whole-genome sequencing of laser-capture microdissections of prostate (200-500 cells) revealed that these small populations do contain clones but are not fully monoclonal (median VAFs between 0.15 and 0.3) [1]. Hence, it is much more likely that clone A represents a polyclonal population of diploid cells, which I think aligns with the authors’ view on clone A.

We agree with the reviewer that clone A likely represents a polyclonal population of diploid cells as would be expected during the branching morphogenesis responsible for glandular epithelial outgrowth. In order to reflect this supposition more clearly we have made changes to the legend of Figure 3.

- Both SNVs are situated at known common SNP sites (SNP ID rs1062391 for chr8:143580183 and rs1130474 for chr8:99892049), further signaling it is likely these SNVs were inherited rather than acquired post-zygotically.

These VAF of these substitutions confirms the CNVs at these loci, but there is no set of SNV lineage markers that serves as an orthogonal validation of the CNV phylogeny (e.g., a substitution present in clones C-G but absent in the others). I suggest retaining this analysis as it confirms the inferCNV calls and rewriting the section dealing in the text to reflect that these are not somatic mutations.

[1] Grossmann, S., Hooks, Y., Wilson, L., Moore, L., O'Neill, L., Martincorena, I., ... & Campbell, P. J. (2021). Development, maturation, and maintenance of human prostate inferred from somatic mutations. *Cell Stem Cell*, 28(7), 1262-1274.

We agree with the reviewer's suggestion to retain the analysis, and have edited the section to remove the word "somatic", on page 7, line 15 as well as having removed it from the title.

Referee #2 (Remarks to the Author):

The authors have adequately addressed my concerns and have updated the manuscript accordingly.

Thank you.

Referee #3 (Remarks to the Author):

I have reviewed the revised manuscript by Erickson and colleagues.

My major comments were well met with good responses. While the paper itself is much improved and exciting, I would like to draw the authors to their GitHub page, which I was not able to review in the initial submission. I would now expect that top-tier analysis papers to come with very good GitHub repos that describe how to reproduce all analyses and figures, however, there are a number of issues with the quality of the current documentation that should be addressed. There are also a small number of minor issues that should be addressed.

We thank the reviewer for their comments and the opportunity to improve the reproducibility and documentation of our siCNV framework. Please see the following responses below.

- We have changed the github landing page to document installation and general documentation regarding the package functions
- We have added knitr/rmarkdown scripts, which detail how to reproduce all main figures (major results)
- We have updated the Mendeley with the count matrices and input files for the main figure scripts

We invite the reviewer to visit: <https://github.com/aerickso/SpatialInferCNV> and ask them use the following details:

Username: forericksonetalsubmission1

Password: 14MloggingIntoGithub

As GitHub requires two-factor authentication, we provide the related gmail account information.

gmail: foericksonetalsubmission1@gmail.com
password: 14MloggingIntoGithub

Major points on the GitHub repo:

1) General organisation and expectations for a software repository

1a) The GitHub page should ideally be technically oriented around the software, e.g. <https://github.com/aerickso/cvat>. Right now, the page emphasises the analysis of some sample data, which would be ideally part of a Jupyter notebook or ReadTheDocs site.

At the reviewer's suggestion, the landing page for the GitHub page now includes documentation for installation and sources for data to reproduce the analyses in the manuscript. As the package is written in R, we have added rmarkdown notebooks to reproduce all of the novel analyses in the main figures and results. We wish to note, that the documentation for the GEF analyses (Figure 1c/1f), and Seurat clustering (Figure 4b) have already been published previously and can be found at <https://www.nature.com/articles/s41467-018-04724-5> and https://satijalab.org/seurat/articles/spatial_vignette.html. We have undertaken internal review (unit testing) on the package functions, and figure scripts from a colleague who had never used the package before and have incorporated their edits into the GitHub repository. Taken together, 10 completely new rmarkdown scripts for figures were added, as well as revisions to two previous markdown scripts, the completion of documentation for 11 package functions, a complete revision of the landing page to the GitHub, and a significant restructuring in terms of data (removed from GitHub and added to Mendeley) were performed in the process of this revision.

1b) An in-depth user guide could be provided via ReadTheDocs.

We have added rmarkdown notebooks to the GitHub page to reproduce all of the novel analyses in the main figures and results from start to finish.

1c) Reanalysis of data should be provided through some literate programming document (Jupyter notebook, knitr, etc).

We have added rmarkdown notebooks to the GitHub page to reproduce all of the novel analyses in the main figures and results from start to finish.

1d) By now I would expect that it is common practice to provide a notebook or scripts which shows how all figures were generated in the paper. This could be distinct from the software repo.

We have added rmarkdown notebooks to the GitHub page to reproduce all of the novel analyses in the main figures and results from start to finish.

1e) The siCNV framework used in the study should probably be a tagged release cited in the manuscript (e.g. <https://docs.github.com/en/repositories/archiving-a-github-repository/referencing-and-citing-content>).

We thank the reviewer for their suggestion. We fully agree with the reviewer and editor: at the editor's suggestion, at publication we will provide a tagged release (and put the code into a permanent repository and provide a doi in the manuscript).

2) Some specific issues about the repository as a home for data analysis script that can be used to reproduce the results of the study. This might be

2a) The scripts should describe how all major results can be reproduced, and not just the siCNV part

We have added rmarkdown notebooks to the GitHub page to reproduce all of the novel analyses in the main figures and results from start to finish. We wish to note, that the documentation for the GEF analyses (Figure 1c/1f), and Seurat clustering (Figure 4b) have already been published previously and can be found at <https://www.nature.com/articles/s41467-018-04724-5> and https://satijalab.org/seurat/articles/spatial_vignette.html.

2b) In the current README.md of the GitHub repo, some sections have no descriptive text. All sections should be accompanied by at least 1-2 sentences of text. Also, some figures are shown without context or captions (e.g. the first 4 images)

We have revised the GitHub significantly, and have included figure-specific scripts to reproduce the main figure content on the GitHub page. We have improved the documentation by adding text to each section.

2c) There is no description of how to access the data from this study (or appropriate test data, should the authors want this to be a general user guide for their siCNV framework).

We have added all annotation files, input files, and count files to the GitHub page.

2d) There is no obvious way of finding out how the “purest benigns” or “Benign|GG2|PIN|GG4” were identified.

Regarding “Benign|GG2|PIN|GG4”, these are manual annotations, by expert pathologists, then “harmonized” through consensus, of individual Visium spots, as performed by the LoupeBrowser. This is described, in detail, in the methods section “Pathologist Workflow – Spot-level annotation for prostate patient 1.” Regarding “purest benigns”, we provide documentation how to identify the “purest benigns” used in Figures 2 and 3 at the following script

https://github.com/aerickso/SpatialInferCNV/blob/main/FigureScripts/BenignRefs_ForFigs2and3/BenignRefs.md. We provide the annotations, and the purest benigns in .csv files in the Mendeley repository.

2e) There is no description or code block for the “clone tree building” section.

We have revised the GitHub, and please see our response above to reviewer 1, comment 5. The clone trees are manually constructed, and provide methods text for clone tree building in the manuscript on pages 29/30, lines 7-20/1-10.

2g) The analysis scripts/notebooks used to reproduce results presented in the study should probably be a tagged release cited in the manuscript if they are hosted via another GitHub repository (e.g. <https://docs.github.com/en/repositories/archiving-a-github-repository/referencing-and-citing-content>).

We fully agree with the reviewer and editor: at the editor's suggestion, at publication we will provide a tagged release (and put the code into a permanent repository and provide a doi in the manuscript).

Minor comments:

3) The author's response to my minor point 12 suggests that the siCNV clustering approach has limitations. The figure in the author's response to my previous point 12 shows the clustering of the siCNV identified 3 clones. On closer inspection I feel that there is something not quite right with the clustering, e.g. the CNV profile of the top part of the first cluster looks remarkably similar to the bottom half of the first – yet there two parts are in clearly separate clades. Compared to the siCNV clustering presenting the manuscript (which looks convincing), that reviewer only figure is worrying. Can the authors comment on why this happened?

We thank the reviewer for their comment. For clarity we first reproduce here a portion of the previous response:

“Previous response:

At the reviewer’s suggestion, we have analyzed three specimens from one patient from this study, patient BC23209, and provide both the iCNV and global CNV events.”

We apologize for the lack of clarity: siCNV data provided in this response come from three specimens (C1, D1; D2), from this patient. Please note that C2 and D1 are consecutive sections of the same specimen. We unhelpfully cropped the following label from the previous siCNV image.

The colors correspond to the respective sections: and we wish to emphasize that this was a forced/supervised analysis per section (not, an unsupervised analysis).

To provide further clarity, we provide here an unsupervised analysis of all three sections.

The regions highlighted by the reviewer do now cluster under the same clade as expected. These regions represented putative common cancer clone traversing the boundaries of the histological sections. We hope the clarification showing both unsupervised (individually) and supervised (combined) analysis provided strengthen the reviewer’s confidence in siCNV clustering.

4) A brand new paper describes how the slide-RNA-seq approach can be adapted to slide-DNA-seq (<https://www.nature.com/articles/s41586-021-04217-4>). The authors could consider citing this paper in their discussion.

We thank the reviewer for highlighting this paper. We agree it is interesting and complements our work. We actually included this paper as a citation (#28).

Line specific comments from the tracked version of the resubmitted manuscript (406046_1_related_ms_3807532_r4km0w.pdf). Please note that some points are questions.

5) P1L6: there are still occurrences of using a white space as a thousand separator (e.g. P1L6 “120 000”). Please also check the rest of the manuscript.

Thank you for flagging this. We have revised accordingly on page 1, line 5, as well as the rest of the manuscript.

6) P2L12: subheading should be bold.

Thank you. We have revised this (page 2, line 11).

7) P2L16: remove “_” in “infer_CNVs”.

Thank you. We have revised this (page 2, Line 15).

8) P2L18: add a comma between “modality generating”?

This has been added on page 2, line 16.

9) P3L2: “robustly could” or “could robustly”?

We have changed the order as per the reviewers’ suggestion on page 2, line 22 - page 3, line 1.

10) P4L3: should “iCNV” be “siCNV” or “CNV”?

We have changed the iCNV to CNV on page 4, line 1.

11) P7L21: “clonal sub-groups” or “sub-clonal groups”?

We have kept this as “clonal sub-groups” as we are referring to sub-groups of epithelial cells based on clonality.

12) P22L16: “Data processing” should be more specific. Perhaps “Spatial transcriptomics data processing”?

We have revised this (page 22, line 16).

13) P24L16: link to GATK toolkit is incomplete – it should be <https://github.com/broadinstitute/gatk>, and not <https://github.com/broadinstitute>.

We have revised this (page 24, line 16).

14) P26L18: change “InferCNV – Data Pre-processing” to “Data pre-processing for inferring spatial CNVs” (to avoid the InferCNV confusion).

We have revised this (page 26, line 18).

15) P29LL5: I cannot easily interpret what is meant by “if a clone in a given section had ≤ 10 1k or Visium spots”. Did you mean “ $< 1k$ [UMIs/genes] and < 10 Visium Spots”?

We thank the reviewer for their question. 1k was used here as a reference to the Spatial transcriptomic 1k array or ST version 1 which has a total 1000 spots¹¹. We have edited page 29, line 5-6 for clarity.

Figures

16) Figure 1 panel a – still refers to iCNVs.

We have revised the figure 1 panel a and also corrected iCNVs present on page 7, line 10 as well as page 10 line 8.

17) Figure 2 legend has “sCNV” instead of “siCNV”.

We have revised the Figure 2 legend.

1. Gao, Y. *et al.* Single-cell sequencing deciphers a convergent evolution of copy number alterations from primary to circulating tumor cells. *Genome Res.* **27**, 1312–1322 (2017).
2. Brocks, D. *et al.* Intratumor DNA methylation heterogeneity reflects clonal evolution in aggressive prostate cancer. *Cell Rep.* **8**, 798–806 (2014).
3. Pienta, K. J., Hammarlund, E. U., Axelrod, R., Amend, S. R. & Brown, J. S. Convergent Evolution, Evolving Evolvability, and the Origins of Lethal Cancer. *Mol. Cancer Res.* **18**, 801–810 (2020).
4. Grossmann, S. *et al.* Development, maturation, and maintenance of human prostate inferred from somatic mutations. *Cell Stem Cell* (2021) doi:10.1016/j.stem.2021.02.005.
5. Schliep, K. P. phangorn: phylogenetic analysis in R. *Bioinformatics* **27**, 592–593 (2011).
6. Gundem, G. *et al.* The evolutionary history of lethal metastatic prostate cancer. *Nature* **520**, 353–357 (2015).
7. Hong, M. K. H. *et al.* Tracking the origins and drivers of subclonal metastatic expansion in prostate cancer. *Nat. Commun.* **6**, 6605 (2015).
8. Cooper, C. S. *et al.* Analysis of the genetic phylogeny of multifocal prostate cancer identifies multiple independent clonal expansions in neoplastic and morphologically normal prostate tissue. *Nat. Genet.* **47**, 367–372 (2015).
9. Woodcock, D. J. *et al.* Prostate cancer evolution from multilineage primary to single

lineage metastases with implications for liquid biopsy. *Nat. Commun.* **11**, 5070 (2020).

10. Martincorena, I. *et al.* Somatic mutant clones colonize the human esophagus with age. *Science* **362**, 911–917 (2018).

11. Ståhl, P. L. *et al.* Visualization and analysis of gene expression in tissue sections by spatial transcriptomics. *Science* **353**, 78–82 (2016).

Reviewer Reports on the Second Revision:

Referees' comments:

Referee #1 (Remarks to the Author):

I thank the authors for their revisions. Just to explain my point of view in the previous round of comments, I want to remark that as a scientific community, we should always strive for absolute reproducibility of our analyses and conclusions. Rather than a preference of mine, it is essential to be crystal clear about the way in which the data has generated a result, such as a phylogeny. This is especially true when the single cell RNA data has several layers of noise, which are absent from conventional (whole-genome) DNA sequencing data. In this respect, it is irrelevant what previous publications have done – if I cannot reproduce the analyses from the data, I cannot support the results.

With the addition of the maximum parsimony trees and a much clearer supplementary table listing the copy number events, the reproducibility has been much improved and my concerns have been assuaged.

A few textual remarks:

- The authors state they have noted the limitation of copy-neutral LOH on page 7, line 15, but I cannot find the stated limitation. This should be fixed.
- Page 7, line 16: please change "single nucleotide variants (SNV)" to "single nucleotide polymorphisms (SNP)" just to make abundantly clear what is meant here.

I congratulate the authors on their manuscript and thank them for the fruitful rounds of discussion.

Referee #3 (Remarks to the Author):

Dear Authors,

The revised work addresses my concerns with regards to the GitHub software repository - I was able to install the package using the instructions for conda. I am excited to see this work closer to publication, however, some minor concerns remain:

- 1) The conda recipe for siCNV states the packages, but the versions of the tools are not defined. The authors should revise the YML file to define the version for all tools (e.g. "r-base=4.0.1"). This is absolutely required to ensure reproducibility (e.g. results from Seurat v3 may differ from Seurat v4).
- 2) Reviewer 1 correctly pointed out that two exemplary SNVs reported in the study (Page7-Line15) are indeed polymorphisms and not somatic. I appreciate that the authors have omitted the word somatic on Page7-Line15, but there are still some elements that could be improved. Specifically:
 - 2a) P7-L18. Since it is clear that the 2 exemplary positions on chr8 are germline, the authors should refer to them as "SNPs" or "polymorphisms" instead of "SNVs" to avoid confusion.
 - 2b) P7-L19. Since it is clear that the 2 exemplary positions on chr8 are germline, the conclusion that they "support shared ancestry" is no longer appropriate in this context. Analysis of germline mutations in this context can only support that the clones originated from the same patient (which is a trivial conclusion). The authors would need to find a somatic mutation to illustrate that the clones originated from the same primary tumor clone, which might not be easily possible. The authors should either find a somatic mutation that supports their original conclusion, or revise

their conclusion (to e.g. "supports the accuracy of the copy number calling by inferCNV, which does not explicitly model allelic imbalance in calculating copy number states").

If my points are well addressed and that reviewer 1 has no further concerns on point 2, then I would be fine with the editor making a final decision without another round of review from me. In that case, thank you for entertaining my criticisms and well done.

Author Rebuttals to Second Revision:

Dear Michelle,

We have revised the manuscript as requested by the reviewers (see below) as well as the editorial comments both in the manuscript. We also have addressed all of the comments and requested changes in the editorial summary.

Thank you again for the opportunity to publish our work with you.

Sincerely

Joakim Lundeberg and Alastair Lamb (on behalf of all authors)

Referees' comments:

Referee #1 (Remarks to the Author):

I thank the authors for their revisions. Just to explain my point of view in the previous round of comments, I want to remark that as a scientific community, we should always strive for absolute reproducibility of our analyses and conclusions. Rather than a preference of mine, it is essential to be crystal clear about the way in which the data has generated a result, such as a phylogeny. This is especially true when the single cell RNA data has several layers of noise, which are absent from conventional (whole-genome) DNA sequencing data. In this respect, it is irrelevant what previous publications have done – if I cannot reproduce the analyses from the data, I cannot support the results.

We thank the reviewer for this explanation and fully agree.

With the addition of the maximum parsimony trees and a much clearer supplementary table listing the copy number events, the reproducibility has been much improved and my concerns have been assuaged.

A few textual remarks:

- The authors state they have noted the limitation of copy-neutral LOH on page 7, line 15, but I cannot find the stated limitation. This should be fixed.

Additional lines that clarify these limitations have been added to the beginning of the paragraph. It now reads “We recognize that a limitation of using siCNV is that this approach does not capture mutations such as single-nucleotide variants (SNV) or other copy neutral events, which could add value in discriminating clonal groupings. We therefore undertook an analysis of transcribed (exonic) single-nucleotide polymorphism (SNP) using...”

- Page 7, line 16: please change “single nucleotide variants (SNV)” to “single nucleotide polymorphisms (SNP)” just to make abundantly clear what is meant here.

Thanks for pointing this out. We have revised this accordingly.

I congratulate the authors on their manuscript and thank them for the fruitful rounds of discussion.

Referee #3 (Remarks to the Author):

Dear Authors,

The revised work addresses my concerns with regards to the GitHub software repository - I was able to install the package using the instructions for conda. I am excited to see this work closer to publication, however, some minor concerns remain:

1) The conda recipe for siCNV states the packages, but the versions of the tools are not defined. The authors should revise the YML file to define the version for all tools (e.g. "r-base=4.0.1"). This is absolutely required to ensure reproducibility (e.g. results from Seurat v3 may differ from Seurat v4).

The YML file has been updated and now includes version numbers for all tools.

2) Reviewer 1 correctly pointed out that two exemplary SNVs reported in the study (Page7-Line15) are indeed polymorphisms and not somatic. I appreciate that the authors have omitted the word somatic on Page7-Line15, but there are still some elements that could be improved. Specifically:

2a) P7-L18. Since it is clear that the 2 exemplary positions on chr8 are germline, the authors should refer to them as "SNPs" or "polymorphisms" instead of "SNVs" to avoid confusion.

We agree and these are now referred to as SNPs instead of SNVs.

2b) P7-L19. Since it is clear that the 2 exemplary positions on chr8 are germline, the conclusion that they "support shared ancestry" is no longer appropriate in this context. Analysis of germline mutations in this context can only support that the clones originated from the same patient (which is a trivial conclusion). The authors would need to find a somatic mutation to illustrate that the clones originated from the same primary tumor clone, which might not be easily possible. The authors should either find a somatic mutation that supports their original conclusion, or revise their conclusion (to e.g. "supports the accuracy of the copy number calling by inferCNV, which does not explicitly model allelic imbalance in calculating copy number states.").

We thank the reviewer for pointing this out. While the reviewer is correct in that they no longer "support shared ancestry", they still support copy number events on the same allele which in turn is consistent with shared ancestry. The conclusion has been revised to convey this and now reads "Analyses of the ratios of clonal variant allele fractions of both specific events with high coverage SNPs (exemplified by chr8:143580183 & 99892049) [Extended Data Fig. 7] support copy number events on the same allele, consistent with shared ancestry [Figure 3b]."

If my points are well addressed and that reviewer 1 has no further concerns on point 2, then I would be fine with the editor making a final decision without another round of review from me. In that case, thank you for entertaining my criticisms and well done.

Editor comments:

1. Please reduce the article title to 75 characters (with spaces) or less. We suggest the following alternative title: "Spatially resolved clonal copy number alterations in benign and malignant tissue" (this is still slightly too long, but fits our template and will work)

We have revised the title to the suggested alternative title.

2. Please provide the manuscript file as a word document.

Both the track-changes version and edited version have been uploaded as word documents.

3. Please add references to the abstract/summary paragraph.

We have added two references to the abstract.

4. Please ensure that the text size in all figures is at least 5 pt Arial.

We have checked and edited the text of all figures to be at least 5 pt DINCon. We have used DINCon throughout all the figures and also provided the fonts in the form of .otf files.

5. Please separate the main the methods reference into two continuously numbered, but separate lists, following directly the discussion and the methods section, respectively.

The reference list has been separated as requested.

6. Please remove the main figures from the article file and re-supply them individually in an acceptable format such as EPS, AI, PS, PDF, PPT, CDR, PSD or XLS (for graphs).

The main figures have been removed from the article file and re-submitted as AI files.

7. Please remove the Extended data figures from the article file and re-supply in EPS, JPEG or TIF format.

The Extended data figures have been removed from the extended data file and re-supplied as TIF files.

8. Figure 1 and 4 are too tall in height when resized to 18 cm width, please reduce to 17 cm or less

All main figures have been supplied as AI files inside artboards that are 18 cm in width and 17 cm in height.

9. Please reduce subheadings to 40 characters (with spaces) or less

All main subheadings have been reduced to 40 characters or less.

10. Please provide an SI Guide.

A SI Guide with titles and text summary for each file has been submitted.

11. Can the Supplementary Note "Processing and visualization of non-prostate samples" be included in the Methods section?

This has now been included in the *Processing and visualization of non-prostate samples* section.

12. Are any permissions required for the human silhouette and illustrations in Figure 1?

No permissions required.

13. Please split data and code availability statement into two separate sections.

These have been split and complemented with additional availability statements.

14. For the deposited data in EGA, please clarify which access restrictions apply.

The specific access restrictions that apply have been added.

15. For the github link, please make sure to now include the updated link. Please also (in addition to the github repo) archive the code in a permanent repository and provide a doi here.

The github repository has now been made public (<https://github.com/aerickso/SpatialInferCNV>). A permanent repository is available using the following doi: <https://doi.org/10.6084/m9.figshare.19666317.v1>

16. We notice that part of your data have been deposited on Mendeley for which registration is required. Is it possible to deposit these data somewhere where such a registration step can be avoided?

The data deposited on Mendeley was previously a draft and required registration. It has now been published with its own doi and can be accessed without registration using the following doi: <https://doi.org/10.17632/svw96g68dv.1>

17. Please move the Ethics Declaration section to the beginning of the Methods section.

The Ethics Declaration section has been moved to the beginning of the Methods section.